# Development and validation of the pandemic fatigue scale

Lau Lilleholt [1,2,3] ✉, Ingo Zettler [1,2], Cornelia Betsch [4,5] & Robert Böhm [1,2,6]

The existence and nature of pandemic fatigue–defined as a gradually emerging subjective state of weariness and exhaustion from, and a general demotivation towards, following recommended health-protective behaviors, including keeping oneself informed during a pandemic–has been debated. Herein, we introduce the Pandemic Fatigue Scale and show how pandemic fatigue evolved during the COVID-19 pandemic, using data from one panel survey and two repeated cross-sectional surveys in Denmark and Germany (overall $N = 34{,}582$). We map the correlates of pandemic fatigue and show that pandemic fatigue is negatively related to people's self-reported adherence to recommended health-protective behaviors. Manipulating the (de)motivational aspect of pandemic fatigue in a preregistered online experiment ($N = 1584$), we further show that pandemic fatigue negatively affects people's intention to adhere to recommended health-protective behaviors. Combined, these findings provide evidence not only for the existence of pandemic fatigue, but also its psychological and behavioral associations.

Throughout the COVID-19 pandemic, governments and health authorities recommended and mandated various health-protective behaviors, such as mask wearing and physical distancing. While being effective in constraining the pandemic[1–5], health-protective behaviors have economic and psychological costs[6,7]. Correspondingly, several countries witnessed a gradual decline in public adherence to health-protective behaviors over the course of the pandemic[8–10].

According to the World Health Organization (WHO), one potential explanation for such a decline is (the rise of) pandemic fatigue[11]. As a latent phenomenon not directly observable, pandemic fatigue has been proposed to express itself behaviorally "through an increasing number of people not sufficiently following recommendations and restrictions, [and] decreasing their effort to keep themselves informed about the pandemic" (p. 7)[11]. Whether the observed decline in public adherence to health-protective behaviors can be attributed to pandemic fatigue has been debated, however. On the one hand, some researchers have questioned the existence of pandemic fatigue, pointing to a lack of scientific evidence to support the claim that pandemic fatigue is responsible for the decline in public adherence to health-protective behaviors[12–14]. On the other hand, some politicians, the WHO, and other researchers have–on the basis of behavioral observations–argued that pandemic fatigue is a real and important phenomenon[8,10,11,15–18].

Arguably, much of this debate is fueled by the fact that pandemic fatigue has largely been derived from an observed decline in public adherence to health-protective behaviors, which could be explained by other factors[8–10,18], such as changes in the perceived risk of COVID-19 or plummeting trust in governments' abilities to handle the pandemic[13,14]. Although some studies aligning pandemic fatigue with behavioral observations have sought to control for such alternative explanations[8,9], they still only provide indirect evidence for the existence of pandemic fatigue. In studies aiming to assess pandemic fatigue differently, by contrast, the construct has been defined rather vaguely, conceptualized rather roughly, and measured using single items or non-validated scales[15,19–22], making it difficult to draw firm conclusions about the existence and nature of pandemic fatigue.

[1]Department of Psychology, University of Copenhagen, Øster Farimagsgade 2A, 1353 Copenhagen, Denmark. [2]Copenhagen Center for Social Data Science (SODAS), University of Copenhagen, Øster Farimagsgade 5, 1353 Copenhagen, Denmark. [3]Centre for the Experimental-Philosophical Study of Discrimination, Aarhus University, Bartholins Allé 7, 8000 Aarhus, Denmark. [4]Institute for Planetary Health Behaviour, University of Erfurt, Nordhäuser Str. 63, 99089 Erfurt, Germany. [5]Health Communication, Bernhard-Nocht-Institute for Tropical Medicine, Bernhard-Nocht-Strasse 74, 20359 Hamburg, Germany. [6]Faculty of Psychology, University of Vienna, Universitätsstrasse 7, 1010 Vienna, Austria. ✉e-mail: llj@psy.ku.dk

Tackling these issues, we herein present a theoretically informed conceptualization of pandemic fatigue and provide empirical evidence for its existence and nature.

Conceptualizing pandemic fatigue, we first consider the nature of fatigue more generally and then clarify what makes pandemic fatigue unique and conceptually different from related constructs[23]. In general, fatigue is a complex phenomenon with no commonly accepted definition[24–26]. Broadly speaking, fatigue has either been conceptualized as a state of weariness, exhaustion, and reduced motivation to perform various activities (i.e., a subjective feeling) or as the inability to sustain physical and/or mental operations over time caused by a depletion of physical and/or mental resources (i.e., a performance decrement)[24–28]. Whereas both conceptualizations contribute to the understanding of fatigue, a growing number of scholars have argued for an increased focus on fatigue as a subjective feeling, given that physical and/or mental endurance is chiefly limited by people's motivation to exert effort[27,29–32]. That is, while people's performance is likely to deteriorate over time as their physical or mental resources get depleted, they rarely reach a point at which they can no longer sustain the physical or mental operations needed to perform the activity at hand[27]. In contrast, people typically stop an activity because they feel exhausted and find it difficult to motivate themselves[27]. In line with the increasing focus on fatigue as a subjective feeling[27,29–32], we conceptualize pandemic fatigue as a subjective state rather than as a physical or mental breakdown of one's ability to continuously adhere to recommended health-protective behaviors and/or to stay informed about the pandemic. More specifically, drawing on the notion of pandemic fatigue put forward by the WHO[11], we define pandemic fatigue as a gradually emerging subjective state of weariness and exhaustion from, and as a general demotivation towards, following recommended health-protective behaviors, including keeping oneself informed about the pandemic. Correspondingly, pandemic fatigue is different from general fatigue, which may arise for various reasons and may affect people's engagement in many different activities. Notably, the introduced definition of pandemic fatigue highlights information seeking as a health-protective behavior. This is crucial as it (i) acknowledges that one needs to keep oneself informed about the current situation and guidelines; and (ii) recognizes that feeling exhausted from and demotivated towards keeping oneself informed is as integral to the experience of pandemic fatigue as feeling exhausted from and demotivated towards adhering to other health-protective behaviors (e.g., physical distancing). With regard to the information seeking aspect, it is important to note that people will tend to seek less information over the course of a pandemic due to information saturation and habituation, irrespective of whether or not they are experiencing pandemic fatigue. The information seeking aspect of pandemic fatigue thus refers to a decline in people's tendency to seek information beyond what might be expected naturally.

It is important to dissociate pandemic fatigue from both amotivation (or demotivation) and burnout. Amotivation may be defined as "a state in which one either is not motivated to behave, or one behaves in a way that is not mediated by intentionality" (p. 190)[33]. According to Self-Determination Theory, amotivation can take two forms[33]. First, people may feel amotivated if they believe that their actions will not yield a desired outcome (e.g., believing that physical distancing will not slow down a pandemic) or if they perceive themselves as incapable of attaining a desired outcome (e.g., finding it impossible to keep a safe distance to others)[33]. Second, people may feel amotivated when a behavior has no meaning or value for them[33]. That is, people may feel amotivated when the perceived intrinsic and/or extrinsic utility of doing something is low. For instance, people's motivation for wearing a mask may be undermined if the perceived cost of wearing a mask outweighs its perceived intrinsic and extrinsic benefits. While these two forms of amotivation are likely to play a role in shaping people's experience of pandemic fatigue, there is more to pandemic fatigue

than feeling amotivated. In particular, people will also feel worn out and exhausted from having adhered to various health-protective behaviors for a prolonged time period. It is thus possible to differentiate between pandemic fatigue and being amotivated: Someone who doubts the effectiveness of physical distancing measures and for this reason does not adhere to them is not experiencing pandemic fatigue, but rather feels amotivated. In contrast, someone who, after several weeks of adhering to physical distancing measures, feels exhausted and no longer adheres to the measures is not just amotivated, but rather experiencing pandemic fatigue.

Another construct related to pandemic fatigue is burnout. Defined as a prolonged psychological response to chronic emotional and interpersonal stressors on the job, burnout is characterized by feelings of cynicism, exhaustion, and inefficacy[34,35]. What differentiates burnout from pandemic fatigue is not only (some of) the symptoms, but also the source of the symptoms. Whereas burnout develops as a consequence of a persistent imbalance between one's job resources and demands and/or diverging personal and organizational values and visions[34], pandemic fatigue emerges as a consequence of continuously having to adhere to various health-protective behaviors which impose individual costs that can be at odds with one's basic needs, such as the need for autonomy and relatedness (i.e., feeling socially connected)[33].

To test the existence and conceptualization of a new construct it is crucial to have a sound measurement tool[23], and to provide evidence for the construct validity of the proposed measurement tool, including its content, convergent, and criterion-oriented validity[36,37]. Given our conceptualization of pandemic fatigue, this entails (i) developing a measure that assesses all relevant aspects of people's experience of pandemic fatigue (i.e., content validity), (ii) demonstrating that pandemic fatigue develops over time both within and between individuals, (iii) showing that it is meaningfully associated with other constructs (i.e., convergent validity), and (iv) providing evidence for its connection to people's tendency to adhere to recommended health-protective behaviors (i.e., criterion-oriented validity).

In this work, we accordingly develop a brief measure of pandemic fatigue and then use it to explore the development of pandemic fatigue over time, investigate its relation to other constructs relevant for people's adherence to various health-protective behaviors (e.g., institutional trust[38]), and examine its relation to people's tendency to adhere to four health-protective behaviors (namely, physical distancing, hygienic practices, mask wearing, and information seeking) in a series of repeated cross-sectional surveys conducted in Denmark and Germany as well as a corresponding Danish panel survey (overall $N = 34,582$). Following this, we provide evidence for the impact of the (de)motivational aspect of pandemic fatigue on people's intention to adhere to recommended health-protective behaviors in a preregistered online experiment ($N = 1584$). Taken together, our findings suggest that pandemic fatigue is a multifaceted construct that waxes and wanes over the course of a pandemic, and that is consistently related to people's tendency to adhere to recommended health-protective behaviors.

## Results

All presented analyses were conducted in R 4.2.2[39]. To help interpret our findings, we report standardized effect sizes. For any comparison of group means we report Cohen's d, for which values of $\geq 0.20$, $\geq 0.50$, and $\geq 0.80$ can be interpreted as small, medium, and large effect sizes, respectively[40]. For all regression-based analyses we report Cohen's $f^2$, for which values of $\geq 0.02$, $\geq 0.15$, and $\geq 0.35$ can be interpreted as small, medium, and large effect sizes, respectively[40]. For all mixed-model regression analyses we provide an estimate of Cohen's $f^2$ based on either the marginal $R^2$ (i.e., the proportion of the total variance attributable to the fixed effects portion of the model) or the conditional $R^2$ (i.e., the proportion of the total variance attributable to both the fixed and random effects portion of the model)[41,42]. For individual fixed effect

**Table 1 | Standardized loadings, communalities, uniqueness, and complexity for the six items retained based on Pearson product-moment correlations**

| Item | IF | BF | Communalities | Uniqueness | Complexity |
|---|---|---|---|---|---|
| 1. I am tired of all the COVID-19 discussions in TV shows, newspapers, and radio programs, etc. | 0.85 | −0.04 | 0.68 | 0.32 | 1.00 |
| 2. I am sick of hearing about COVID-19. | 0.88 | 0.01 | 0.79 | 0.21 | 1.00 |
| 3. When friends or family members talk about COVID-19, I try to change the subject because I do not want to talk about it anymore. | 0.50 | 0.22 | 0.44 | 0.56 | 1.40 |
| 4. I feel strained from following all of the behavioral regulations and recommendations around COVID-19. | 0.02 | 0.83 | 0.70 | 0.30 | 1.00 |
| 5. I am tired of restraining myself to save those who are most vulnerable to COVID-19. | 0.09 | 0.58 | 0.41 | 0.59 | 1.10 |
| 6. I am losing my spirit to fight against COVID-19. | −0.06 | 0.71 | 0.45 | 0.55 | 1.00 |
| Eigenvalues | 1.83 | 1.64 | | | |
| Proportion of variance | 0.30 | 0.27 | | | |

Response scale: 1 = strongly disagree, 2 = disagree, 3 = somewhat disagree, 4 = neutral/neither disagree nor agree, 5 = somewhat agree, 6 = agree, 7 = strongly agree.
*IF* information fatigue, *BF* behavioral fatigue.

predictors, we report marginal Cohen's $f^2$ based on the marginal $R^2$, whereas for full models we report both marginal and conditional Cohen's $f^2$ based on the marginal and conditional $R^2$, respectively.

**Development and validation of the pandemic fatigue scale (PFS)**
Via item generation and selection processes, exploratory and confirmatory factor analyses, internal consistency analyses, and measurement invariance testing, we developed and validated a six-item pandemic fatigue scale (PFS; Table 1). The scale measures pandemic fatigue as a second-order latent construct with two subfactors: 'information fatigue' (feeling exhausted from and demotivated towards keeping oneself informed about the pandemic) and 'behavioral fatigue' (feeling exhausted from and demotivated towards following recommended health-protective behaviors). The PFS has excellent psychometric properties and is partially invariant across Denmark and Germany (for more information, see Methods).

**The development of pandemic fatigue over time**
We observe an increase of pandemic fatigue over time, using ordinary least square regression analysis for the Danish ($\beta_{standardized} = 0.02$, t(15,983) = 2.11, $p_{two-tailed} = 0.035$, Cohen's $f^2_{model} < 0.001$, 95% CI [0.00, 0.04]) and German repeated cross-sectional data ($\beta_{standardized} = 0.24$, $t(17,944) = 21.28$, $p_{two-tailed} < 0.001$, Cohen's $f^2_{model} = 0.025$, 95% CI [0.22, 0.26]), and mixed-model regression analysis with random intercepts and slopes for the Danish panel data ($\beta_{standardized} = 0.13$, $t(430.84) = 8.70$, $p_{two-tailed} < 0.001$, marginal/conditional Cohen's $f^2_{model} = 0.009/3.671$, 95% CI [0.10, 0.16]). As shown in Fig. 1A, the development of pandemic fatigue in both Denmark and Germany did not follow a linear trend, but rather a concave pattern in which pandemic fatigue increased from October 2020 to March 2021, then—in Germany only—stagnated, and subsequently decreased—in Denmark and Germany—until September 2021. Including a quadratic term significantly improved the fit of the ordinary least square regression models for both the Danish ($F(1, 15,982) = 229.33$, $p < 0.001$, Cohen's $f^2_{model} = 0.015$) and German repeated cross-sectional data ($F(1, 17,943) = 66.55$, $p < 0.001$, Cohen's $f^2_{model} = 0.029$), and of the mixed-model regression for the Danish panel data ($X^2 (4) = 248.56$, $p < 0.001$, marginal/conditional Cohen's $f^2_{model} = 0.024/4.285$). Controlling for time-dependent contextual factors in terms of new COVID-19 cases and deaths per million, the COVID-19 reproduction rate, and policy stringency (Fig. 1B), we obtain a similar pattern of results with one exception: Only the quadratic term for time remained significant in the mixed-model regression for the Danish panel data (Fig. S1). Overall, this pattern of results corroborates the notion of pandemic fatigue as a gradually emerging subjective state that evolves both within (Danish panel data) and between (Danish and German repeated cross-sectional data) people.

**Correlates of pandemic fatigue**
Next, we investigated the relation between pandemic fatigue and other constructs relevant for people's adherence to recommended health-protective behaviors. With respect to the convergent validity of the PFS one would expect pandemic fatigue to be negatively associated with factors that have been shown to correlate positively with people's tendency to adhere to recommended health-protective behaviors (e.g., institutional trust[38]) as well as positively associated with factors that have been shown to correlate negatively with people's tendency to adhere to recommended health-protective behaviors (e.g., negative affect[43]). Given our exploratory approach, we primarily focus on results that are stable across models and countries when presenting and interpreting our findings. Pairwise correlations for all variables considered in the Danish and German repeated cross-sectional surveys are presented in Figs. S2–S3.

**Sociodemographics and personality dimensions.** As shown in Fig. 2, results from several ordinary least square regression analyses based on the Danish and German repeated cross-sectional surveys revealed that younger people (Cohens $f^2_{predictor – Denmark/Germany} = 0.030/0.069$), women (Cohens $f^2_{predictor – Denmark/Germany} = 0.002/<0.001$), and those with a job (Cohens $f^2_{predictor – Denmark/Germany} = 0.001/0.001$) experienced more pandemic fatigue. In Denmark, the results further indicate that people with more than ten years of education experienced less pandemic fatigue (Cohens $f^2_{predictor} = 0.002$). With regard to basic personality dimensions, which were assessed in the Danish but not in the German repeated cross-sectional survey, we find that people high in emotionality (Cohens $f^2_{predictor} < 0.001$) and extraversion (Cohens $f^2_{predictor} = 0.004$) experienced more pandemic fatigue, whereas people high in honesty-humility (Cohens $f^2_{predictor} = 0.002$), agreeableness vs. anger (Cohens $f^2_{predictor} < 0.001$), conscientiousness (Cohens $f^2_{predictor} < 0.001$), and openness to experience (Cohens $f^2_{predictor} = 0.015$) experienced less pandemic fatigue.

Turning to the Danish panel survey (Fig. 3), we also find that people high in extraversion experienced more pandemic fatigue (marginal Cohens $f^2_{predictor} = 0.004$), whereas older people (marginal Cohens $f^2_{predictor} = 0.021$) and people high in openness to experience (marginal Cohens $f^2_{predictor} = 0.019$) experienced less pandemic fatigue. The negative relation between pandemic fatigue and age observed in the Danish panel survey turned significant only when personality dimensions were not controlled for.

**Perceptions and emotions.** Based on the Danish and German repeated cross-sectional surveys (Fig. 2), we find that people who worried more about potential personal and societal consequences of the pandemic (e.g., losing a loved one or going through a recession) experienced more pandemic fatigue (Cohens $f^2_{predictor – Denmark/Germany} = 0.011/0.039$).

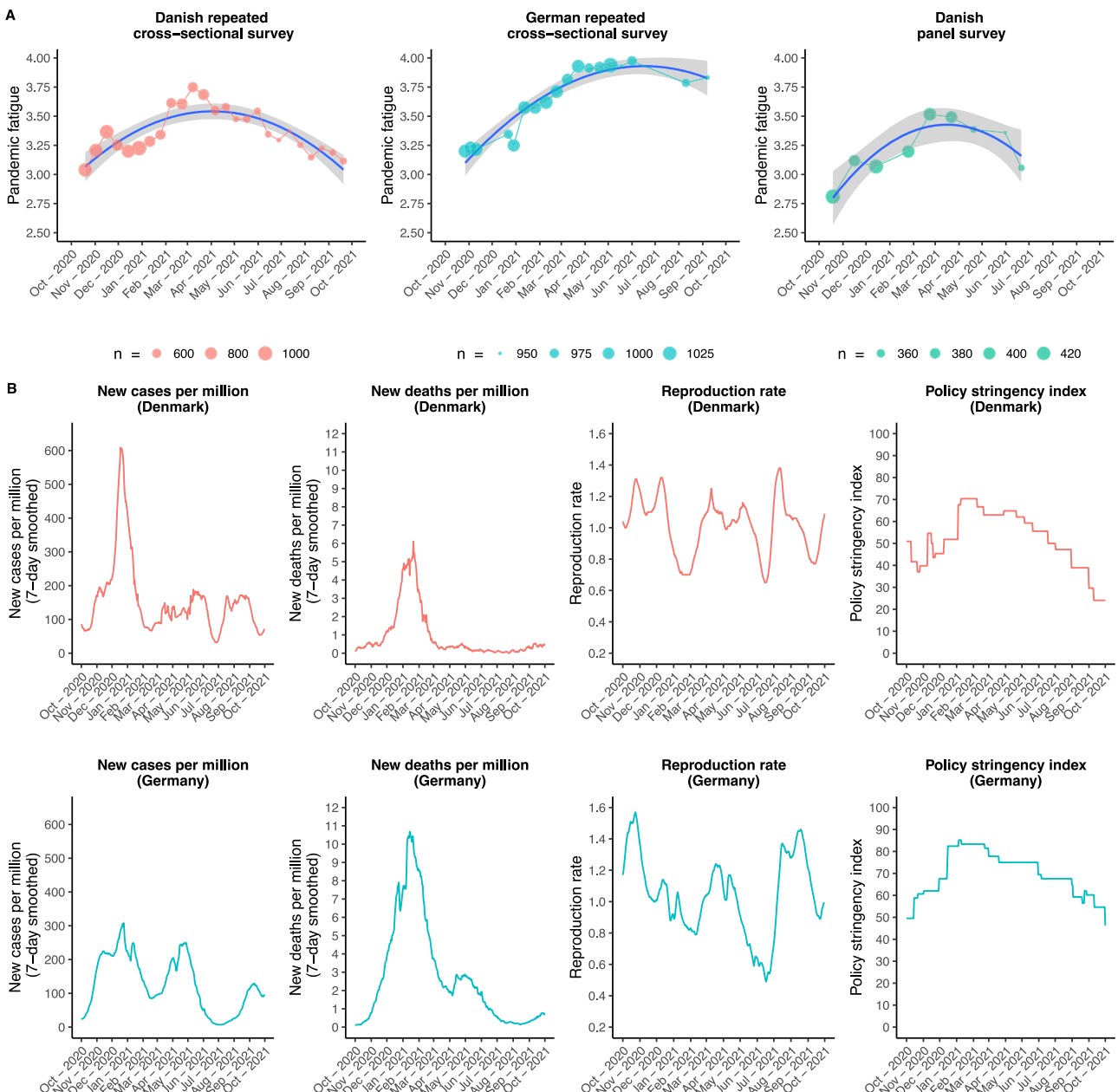

**Fig. 1 | Pandemic fatigue, new COVID-19 cases per million, new deaths per million, reproduction rate, and policy stringency index over time in Denmark and Germany. A** The mean levels of pandemic fatigue for each wave of the Danish and German repeated cross-sectional surveys and the Danish panel survey, respectively, together with polynomial ordinary least square regression lines with 95% confidence intervals. **B** The number of new COVID-19 cases per million, new death per million, reproduction rate, and policy stringency index over time in Denmark and Germany.

Conversely, people with heightened cognitive risk perception (i.e., the perceived probability and severity of getting infected with COVID-19; Cohens $f^2_{predictor\ -\ Denmark/Germany} = 0.001/<0.001$), heightened affective risk perception (i.e., the felt closeness, infectiousness, and affective response to the danger of COVID-19; Cohens $f^2_{predictor\ -\ Denmark/Germany} = 0.018/0.050$), as well as those with higher levels of institutional trust (Cohens $f^2_{predictor\ -\ Denmark/Germany} = 0.128/0.179$) experienced less pandemic fatigue. Concerning optimism, negative affect, and empathy, which were only assessed in the Danish repeated cross-sectional survey, we find that people who felt more negative emotions (e.g., boredom, stress) experienced more pandemic fatigue (Cohens $f^2_{predictor} = 0.079$), whereas people who felt optimistic about the future (Cohens $f^2_{predictor} = 0.002$) and had a strong sense of empathy towards those most vulnerable to COVID-19 (Cohens $f^2_{predictor} = 0.015$) experienced it less.

Using the person-mean centering approach[44,45] to disaggregate the within- and between-subjects effects of the time-varying perceptions and emotions considered in the Danish panel survey, we found a negative relation between pandemic fatigue and: affective risk perceptions regarding COVID-19 (marginal Cohens $f^2_{predictor\ -\ within/between} = 0.001/0.012$); institutional trust (marginal Cohens $f^2_{predictor\ -\ within/between} = 0.004/0.162$); optimism about the future (marginal Cohens $f^2_{predictor\ -\ within/between} = 0.003/0.006$); and empathy towards those most vulnerable to COVID-19 (marginal Cohens $f^2_{predictor\ -\ within/between} = 0.003/0.015$), both within and between subjects (Fig. 3). Moreover, within and between subjects, we found pandemic fatigue to be positively related to negative affect (marginal Cohens $f^2_{predictor\ -\ within/between} = 0.007/0.094$) and worries about potential personal and societal consequences of the pandemic (marginal Cohens $f^2_{predictor\ -\ within/between} = 0.001/0.019$).

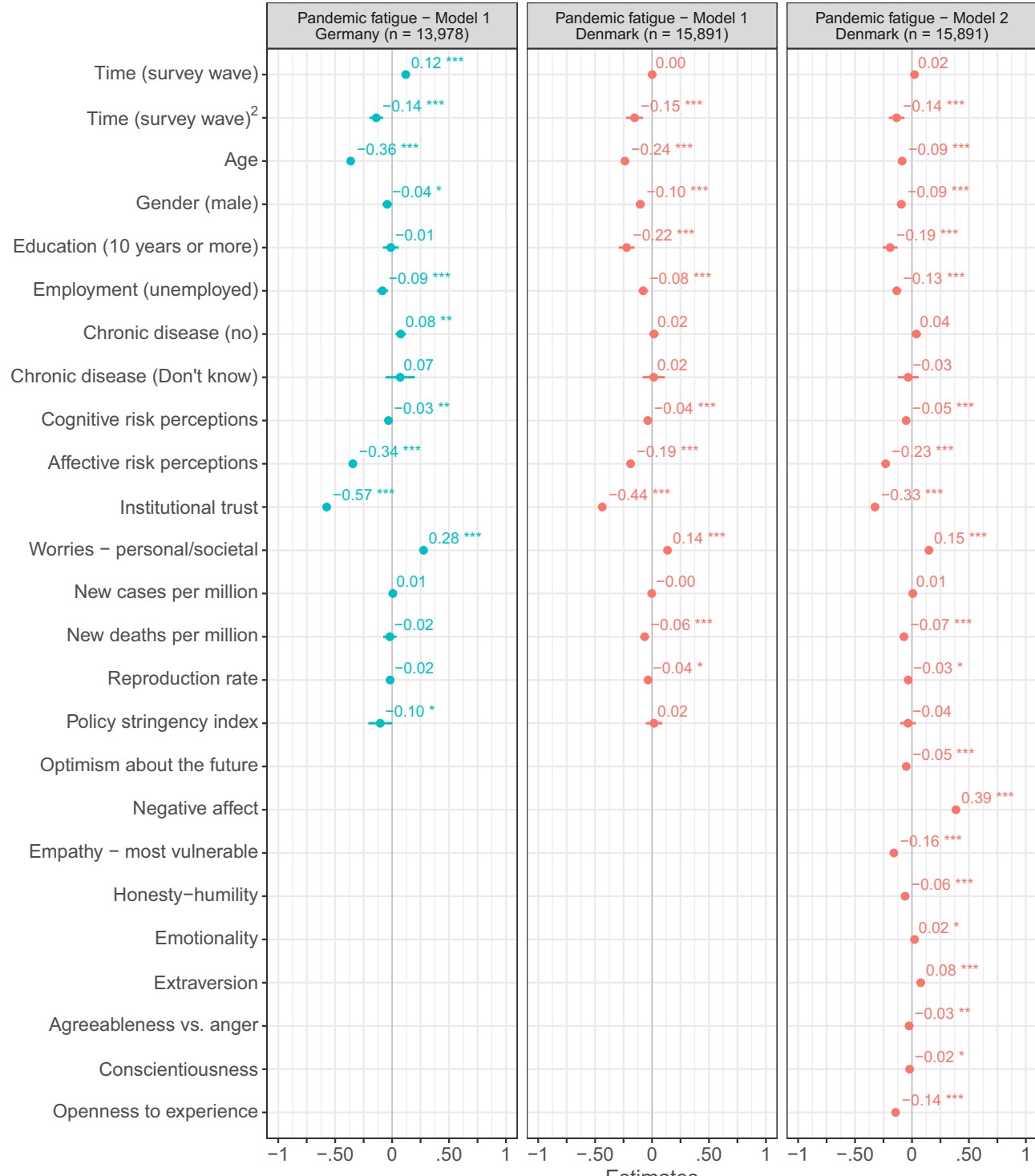

**Fig. 2 | OLS regressions predicting pandemic fatigue in Denmark and Germany.** Figure 2 shows standardized β coefficients with 95% confidence intervals based on ordinary least squares regressions with data from the Danish and German repeated cross-sectional surveys. All continuous predictors have been mean-centered and scaled by 1 standard deviation. The p-values have not been adjusted for multiple comparisons and are presented as follows: $^{***}p_{\text{two-tailed}} < 0.001$;

$^{**}p_{\text{two-tailed}} < 0.01$; $^{*}p_{\text{two-tailed}} < 0.05$. Exact p-values for all models are presented in the R-output which has been deposited on the Open Science Framework at: https://doi.org/10.17605/OSF.IO/XD463. The gender variable refers to participants self-identified gender as presented to them in the surveys. Participants who did not identify as either male or female are not included in the analyses due to an insufficient number of observations.

Taken together, pandemic fatigue was negatively associated with constructs that seem to be positively related to people's tendency to adhere to health-protective behaviors (e.g., age[46], cognitive and affective risk perceptions regarding COVID-19[47], institutional trust[38]), and positively associated with constructs that seem to be negatively related to people's tendency to adhere to health-protective behaviors (e.g., negative affect[43]). The PFS thus appears to have high convergent validity.

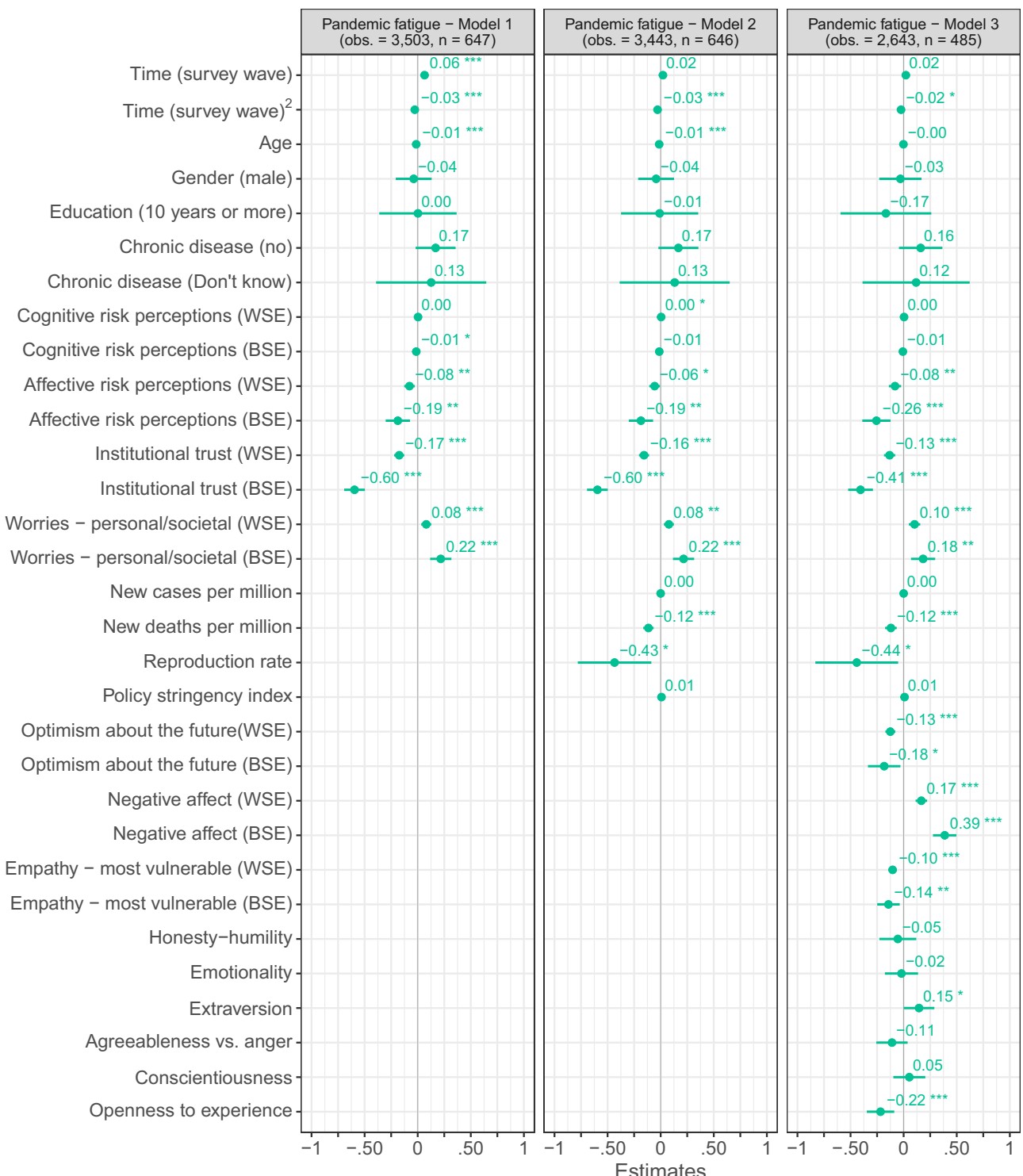

**Fig. 3 | Mixed-model regressions predicting pandemic fatigue in Denmark.**
Figure 3 shows estimated β coefficients with 95% confidence intervals based on mixed-model regressions with data from the Danish panel survey. Continuous time-invariant predictors as well as continuous time-varying contextual predictors (i.e., Time (survey wave), Time (survey wave)$^2$, new COVID-19 cases per million, new deaths per million, reproduction rate, and policy stringency index) have been mean-centered. All other time-varying predictors have been centered using the person-mean centering approach to disaggregate the within- (WSE) and between-subjects effects (BSE) of these factors[44,45]. The p-values have not been adjusted for multiple comparisons and are presented as follows: $^{***}p_{\text{two-tailed}} < 0.001$; $^{**}p_{\text{two-tailed}} < 0.01$; $^{*}p_{\text{two-tailed}} < 0.05$. Exact p-values for all models are presented in the R-output which has been deposited on the Open Science Framework at: https://doi.org/10.17605/OSF.IO/XD463. The gender variable refers to participants self-identified gender as presented to them in the surveys.

## Pandemic fatigue and recommended health-protective behaviors

Next, we examined the relation between pandemic fatigue and people's tendency to adhere to various health-protective behaviors. Based on several ordinary least square regression analyses, controlling only for time, we observe in both the Danish and German repeated cross-sectional surveys a negative relation between pandemic fatigue and people's tendency to adhere to physical distancing

measures (Denmark: $\beta_{standardized} = -0.20$, $t(15,947) = -25.83$, $p_{two\text{-}tailed} < 0.001$, Cohens $f^2_{predictor} = 0.042$, 95% CI [−0.21, −0.18]; Germany: $\beta_{standardized} = -0.23$, $t(14,552) = -39.69$, $p_{two\text{-}tailed} < 0.001$, Cohens $f^2_{predictor} = 0.108$, 95% CI [−0.25, −0.22]), uphold hygienic practices (Denmark: $\beta_{standardized} = -0.18$, $t(15,947) = -26.55$, $p_{two\text{-}tailed} < 0.001$, Cohens $f^2_{predictor} = 0.044$, 95% CI [−0.19, −0.16]; Germany: $\beta_{standardized} = -0.19$, $t(8,247) = -24.68$, $p_{two\text{-}tailed} < 0.001$, Cohens $f^2_{predictor} = 0.074$, 95% CI [−0.20, −0.17]), wear masks (Denmark: $\beta_{standardized} = -0.10$, $t(15,947) = -9.68$, $p_{two\text{-}tailed} < 0.001$, Cohens $f^2_{predictor} = 0.006$, 95% CI [−0.12, −0.08]; Germany: $\beta_{standardized} = -0.17$, $t(17,800) = -29.90$, $p_{two\text{-}tailed} < 0.001$, Cohens $f^2_{predictor} = 0.050$, 95% CI [−0.18, −0.16],), and keep themselves informed about the pandemic (Denmark: $\beta_{standardized} = -0.47$, $t(15,028) = -49.14$, $p_{two\text{-}tailed} < 0.001$, Cohens $f^2_{predictor} = 0.161$, 95% CI [−0.49, −0.45]; Germany: $\beta_{standardized} = -0.52$, $t(17,943) = -48.55$, $p_{two\text{-}tailed} < 0.001$, Cohens $f^2_{predictor} = 0.131$, 95% CI [−0.54, −0.50]).

Disaggregating the within- and between-subjects effects of pandemic fatigue, using the person-mean centering approach[44,45] across four independent mixed-model regression analyses with random intercepts and slopes for time, we observe a similar pattern of results for the Danish panel survey. Specifically, we find a negative between-subjects effect of pandemic fatigue on people's inclination to adhere to physical distancing measures ($\beta_{between\text{-}subjects} = -0.13$, $t(554.64) = -7.26$, $p_{two\text{-}tailed} < 0.001$, marginal Cohens $f^2_{predictor} = 0.043$, 95% CI [−0.16, −0.09],) and wear masks ($\beta_{between\text{-}subjects} = -0.08$, $t(645.85) = -2.61$, $p_{two\text{-}tailed} = 0.009$, marginal Cohens $f^2_{predictor} = 0.007$, 95% CI [−0.15, −0.02]), as well as a negative between- and within-subjects effect on their tendency to uphold hygienic practices ($\beta_{between\text{-}subjects} = -0.15$, $t(645.10) = -6.55$, $p_{two\text{-}tailed} < 0.001$, marginal Cohens $f^2_{predictor} = 0.049$, 95% CI [−0.19, −0.10]; $\beta_{within\text{-}subjects} = -0.04$, $t(2,873.32) = -3.45$, $p_{two\text{-}tailed} < 0.001$, marginal Cohens $f^2_{predictor} = 0.001$, 95% CI [−0.07, −0.02]) and keep themselves informed about the pandemic ($\beta_{between\text{-}subjects} = -0.47$, $t(600.29) = -14.58$, $p_{two\text{-}tailed} < 0.001$, marginal Cohens $f^2_{predictor} = 0.269$, 95% CI [−0.54, −0.41]; $\beta_{within\text{-}subjects} = -0.06$, $t(2,479.16) = -3.20$, $p_{two\text{-}tailed} = 0.001$, marginal Cohens $f^2_{predictor} = 0.002$, 95% CI [−0.10, −0.02]). Notably, in all cases the between-subjects effects of pandemic fatigue were (descriptively) larger than its within-subjects effects. This may suggest that inter-individual differences in people's mean level of pandemic fatigue over time may be more important for their tendency to adhere to health-protective behaviors than any intra-individual changes in their experience of pandemic fatigue.

Adding additional control variables to the ordinary least square regression analyses based on the Danish and German repeated cross-sectional surveys (Figs. 4–7), we find the link between pandemic fatigue and people's adherence to all four health-protective behaviors to be reduced, but still significant (all $p_{two\text{-}tailed} < 0.001$, Cohens $f^2_{predictor} = 0.002$ to 0.063). Turning to the Danish panel survey, we again observe a similar pattern of results: Adding further control variables to the mixed-model regression analyses with random intercepts and slopes for time, we find the between- and within-subjects effects of pandemic fatigue on people's adherence to all four health-protective behaviors to be weaker and in some cases even non-significant (marginal Cohens $f^2_{predictor \; - \; within/between} < 0.001/0.001$ to $= 0.004/0.139$; see Figs. S4–S7). Next to high convergent validity, the PFS thus also appears to have high criterion-oriented validity.

While we find pandemic fatigue to be related to people's tendency to adhere to various health-protective behaviors, it is not the only predictor of this tendency (Figs. 4–7). Especially age (Cohens $f^2_{predictor} = 0.002$ to 0.036), gender (Cohens $f^2_{predictor} = 0.001$ to 0.024), institutional trust (Cohens $f^2_{predictor} < 0.001$ to $= 0.042$), worries about potential personal and societal consequences of the pandemic (Cohens $f^2_{predictor} < 0.001$ to $= 0.028$), and affective risk perceptions regarding COVID-19 (Cohens $f^2_{predictor} = 0.003$ to 0.068)

predicted this tendency—in some cases even (descriptively) better than pandemic fatigue—in both the Danish and German repeated cross-sectional surveys. Similarly, for the Danish panel survey, age (marginal Cohens $f^2_{predictor} < 0.001$ to $= 0.029$), gender (marginal Cohens $f^2_{predictor} < 0.001$ to $= 0.020$), institutional trust (marginal Cohens $f^2_{predictor \; - \; within/between} < 0.001/0.001$ to $= 0.013/0.038$), and affective risk perceptions regarding COVID-19 (marginal Cohens $f^2_{predictor \; - \; within/between} < 0.001/ = 0.001$ to $= 0.004/0.094$) predicted this tendency, together with empathy towards those most vulnerable to COVID-19 (marginal Cohens $f^2_{predictor \; - \; within/between} = 0.001/<0.001$ to $= 0.010/0.045$).

Overall, these results corroborate the idea that pandemic fatigue is linked to the observed decline in public adherence to various health-protective behaviors (see Supplementary Note 1 and Fig. S8). Note that for all regression models in the preceding sections, we further report results in the Supplementary Information from corresponding models in which information and behavioral fatigue were treated as two independent factors (Figs. S9–S20).

## Pandemic fatigue and intentions to adhere to health-protective behaviors

To further substantiate the relation between pandemic fatigue and people's tendency to adhere to recommended health-protective behaviors, we conducted an online experiment in which we manipulated the (de)motivational aspect of participants' experience of pandemic fatigue and assessed its impact on participants' intention to adhere to physical distancing measures, uphold hygienic practices, wear masks, and keep themselves informed about the pandemic. A convenience sample of 1854 U.S. (Prolific[48]) participants was randomized into three conditions: control, low, and high pandemic fatigue. To manipulate the (de)motivational aspect of participants' experience of pandemic fatigue, we relied on a brief self-reflection task in which participants in the low/high pandemic fatigue condition were asked to write a few sentences about some of the things that over the last two weeks had motivated/demotivated them to adhere to the four aforementioned health-protective behaviors. In contrast, participants in the control condition were asked to write about some of the ordinary things that had happened and that somehow affected their behavior. All participants then completed the PFS before responding to four items assessing their intentions to adhere to recommended physical distancing measures, uphold hygienic practices, wear masks, and keep themselves informed about the pandemic (for more information, see Methods).

Following the preregistered analysis plan (https://aspredicted.org/ua3ca.pdf), we excluded participants who wrote fewer than 100 characters (including spaces) in the self-reflection task ($n = 245$), failed an attention check ($n = 10$), or experienced technical issues during the experiment ($n = 15$). A total of 1584 participants were included in the final analysis. To ensure that our experimental manipulation had been successful, we first compared the mean score of the PFS across conditions. As shown in Fig. 8A, results from an independent samples $t$-test showed that participants in the low pandemic fatigue condition ($M = 3.08$, SD $= 1.36$) reported lower levels of pandemic fatigue than participants in the high pandemic fatigue condition ($M = 3.55$, SD $= 1.43$; difference $= 0.47$, $t(1,017.76) = 5.43$, $p_{two\text{-}tailed \; Bonferroni\text{-}adjusted} < 0.001$, Cohen's $d = 0.34$, 95% CI [0.30, 0.64]). Results from two additional independent samples $t$-tests further revealed that participants in the control condition ($M = 3.29$, SD $= 1.45$) reported higher levels of pandemic fatigue than participants in the low pandemic fatigue condition ($M = 3.08$, SD $= 1.36$; difference $= -0.21$, $t(1,079.30) = -2.49$, $p_{two\text{-}tailed \; Bonferroni\text{-}adjusted} = 0.039$, Cohen's $d = 0.15$, 95% CI [−0.38, −0.05]) as well as lower levels than participants in the high pandemic fatigue condition ($M = 3.55$, SD $= 1.43$; difference $= 0.26$, $t(1,044.55) = 2.92$, $p_{two\text{-}tailed \; Bonferroni\text{-}adjusted} = 0.011$, Cohen's $d = 0.18$, 95% CI [0.09, 0.43]). These results suggest that our targeted

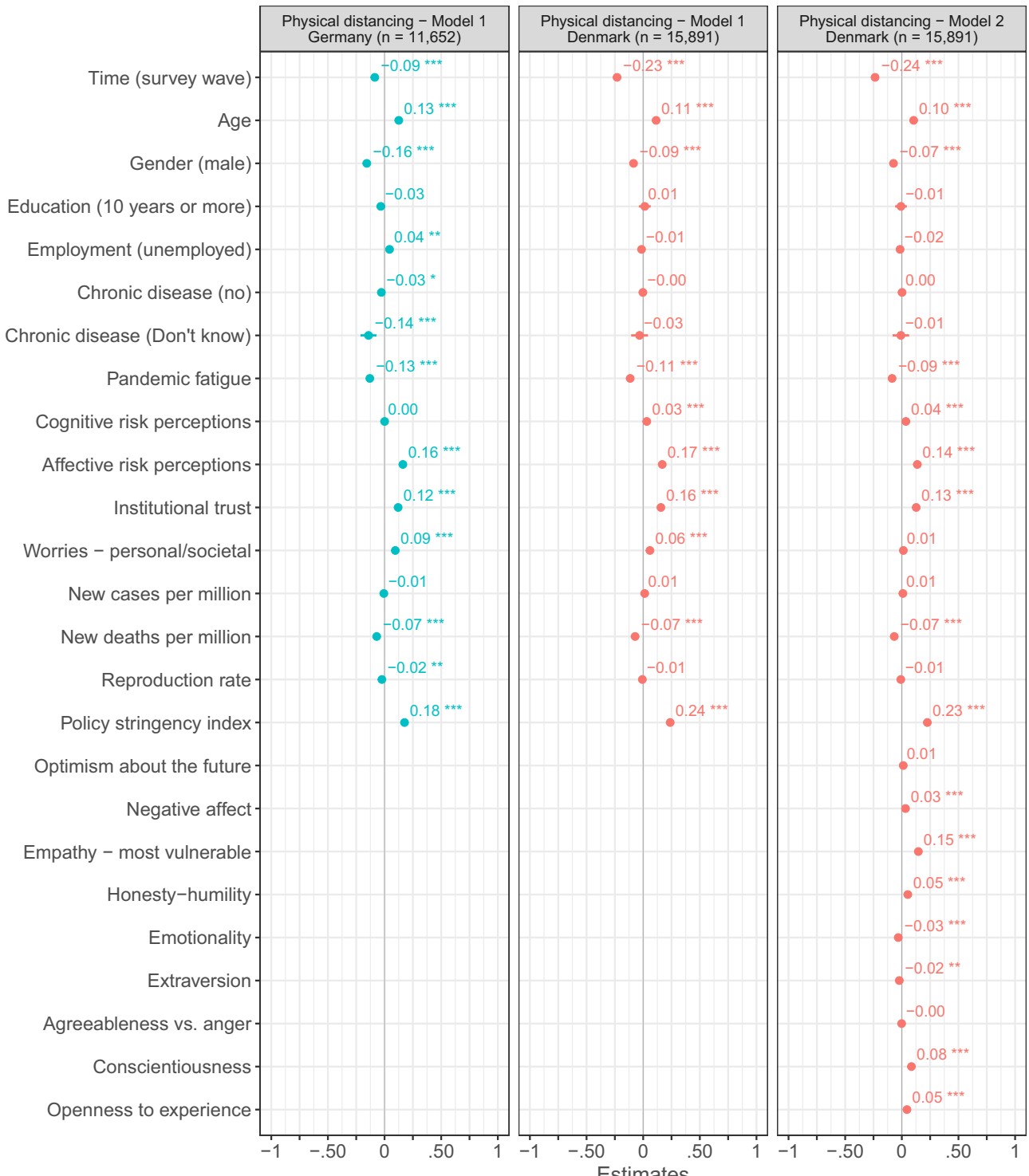

**Fig. 4 | OLS regressions predicting physical distancing in Denmark and Germany.** Figure 4 shows standardized β coefficients with 95% confidence intervals based on ordinary least squares regressions with data from the Danish and German repeated cross-sectional surveys. All continuous predictors have been mean-centered and scaled by 1 standard deviation. The p-values have not been adjusted for multiple comparisons and are presented as follows: ***$p_{\text{two-tailed}} < 0.001$; **$p_{\text{two-}}$ $_{\text{tailed}} < 0.01$; *$p_{\text{two-tailed}} < 0.05$. Exact p-values for all models are presented in the R-output which has been deposited on the Open Science Framework at: https://doi.org/10.17605/OSF.IO/XD463. The gender variable refers to participants self-identified gender as presented to them in the surveys. Participants who did not identify as either male or female are not included in the analyses due to an insufficient number of observations.

experimental manipulation of the (de)motivational aspect of pandemic fatigue was successful.

Assessing the impact of the experimental manipulation on people's intentions to adhere to recommended health-protective behaviors, we relied on an equally weighted composite score of the four outcome items (Cronbach's $\alpha = 0.76$). As illustrated in Fig. 8B, results from an independent samples $t$-test revealed that participants in the high pandemic fatigue condition ($M = 5.65$, SD = 1.18) expressed weaker intentions to adhere to the four health-protective behaviors of interest as compared to participants in the low pandemic fatigue

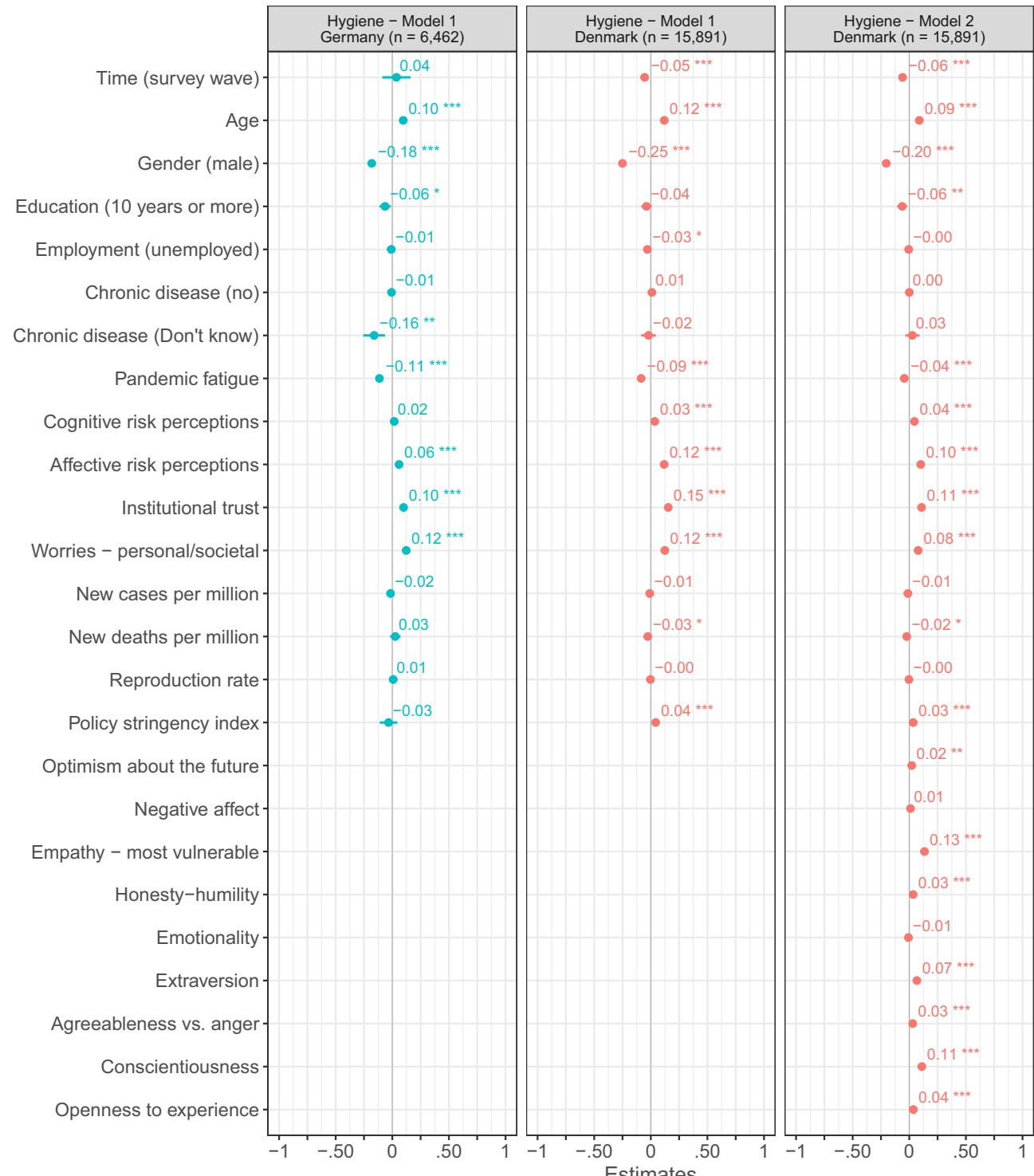

**Fig. 5 | OLS regressions predicting hygiene in Denmark and Germany.** Figure 5 shows standardized β coefficients with 95% confidence intervals based on ordinary least squares regressions with data from the Danish and German repeated cross-sectional surveys. All continuous predictors have been mean-centered and scaled by 1 standard deviation. The *p*-values have not been adjusted for multiple comparisons and are presented as follows: ***$p_{\text{two-}}$

tailed < 0.001; **$p_{\text{two-tailed}}$ < 0.01; *$p_{\text{two-tailed}}$ < 0.05. Exact *p*-values for all models are presented in the R-output which has been deposited on the Open Science Framework at: https://doi.org/10.17605/OSF.IO/XD463. The gender variable refers to participants self-identified gender as presented to them in the surveys. Participants who did not identify as either male or female are not included in the analyses due to an insufficient number of observations.

condition ($M = 5.94$, SD = 1.13; difference = 0.30, $t(1,019.86) = 4.13$, $p_{\text{two-tailed Bonferroni-adjusted}} < 0.001$, Cohen's $d = 0.26$, 95% CI [0.16, 0.44]). In addition, participants in the high pandemic fatigue condition ($M = 5.65$, SD = 1.18) expressed weaker adherence intentions than participants in the control condition ($M = 5.86$, SD = 1.13; difference = 0.21,

$t(1031.30) = 2.98$, $p_{\text{two-tailed Bonferroni-adjusted}} = 0.009$, Cohen's $d = 0.18$, 95% CI [0.07, 0.35]). There was no significant difference between the control condition ($M = 5.86$, SD = 1.13) and the low pandemic fatigue condition ($M = 5.94$, SD = 1.13; difference = 0.09, $t(1,078.10) = 1.24$, $p_{\text{two-tailed Bonferroni-adjusted}} = 0.640$, Cohen's $d = 0.08$, 95% CI [−0.05, 0.22]).

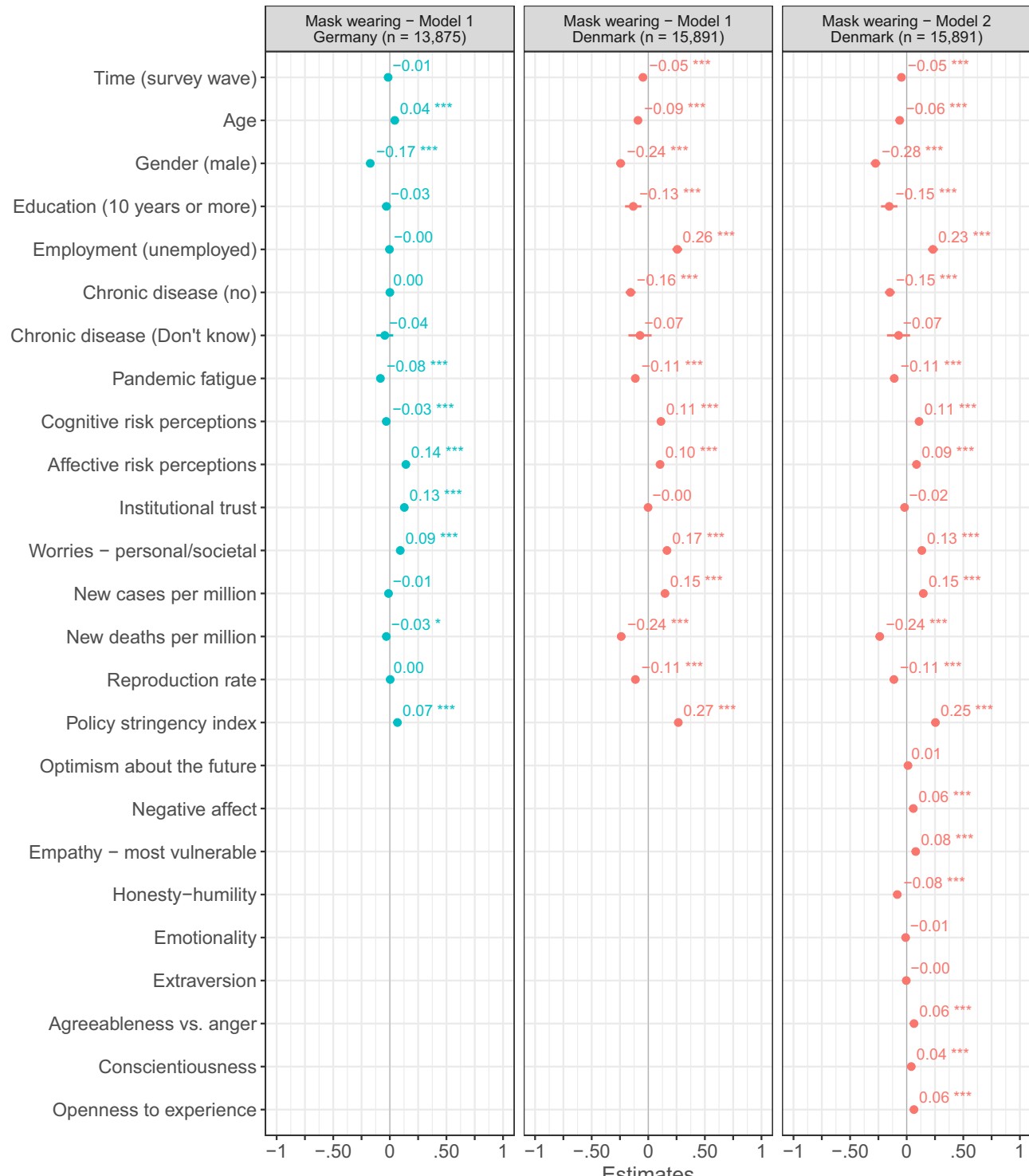

**Fig. 6 | OLS regressions predicting mask wearing in Denmark and Germany.**
Figure 6 shows standardized β coefficients with 95% confidence intervals based on ordinary least squares regressions with data from the Danish and German repeated cross-sectional surveys. All continuous predictors have been mean-centered and scaled by 1 standard deviation. The p-values have not been adjusted for multiple comparisons and are presented as follows: ***$p_{\text{two-tailed}} < 0.001$; **$p_{\text{two-tailed}} < 0.01$;

*$p_{\text{two-tailed}} < 0.05$. Exact p-values for all models are presented in the R-output which has been deposited on the Open Science Framework at: https://doi.org/10.17605/OSF.IO/XD463. The gender variable refers to participants self-identified gender as presented to them in the surveys. Participants who did not identify as either male or female are not included in the analyses due to an insufficient number of observations.

## Discussion

Across three countries, two repeated cross-sectional surveys, one panel survey, and a preregistered online experiment, we provide evidence for the existence and nature of pandemic fatigue. Three general conclusions can be drawn. First, pandemic fatigue consists of two distinct factors (information and behavioral fatigue) that vary over time, both within and between individuals. Second, while most people are likely to experience some form of pandemic fatigue over the

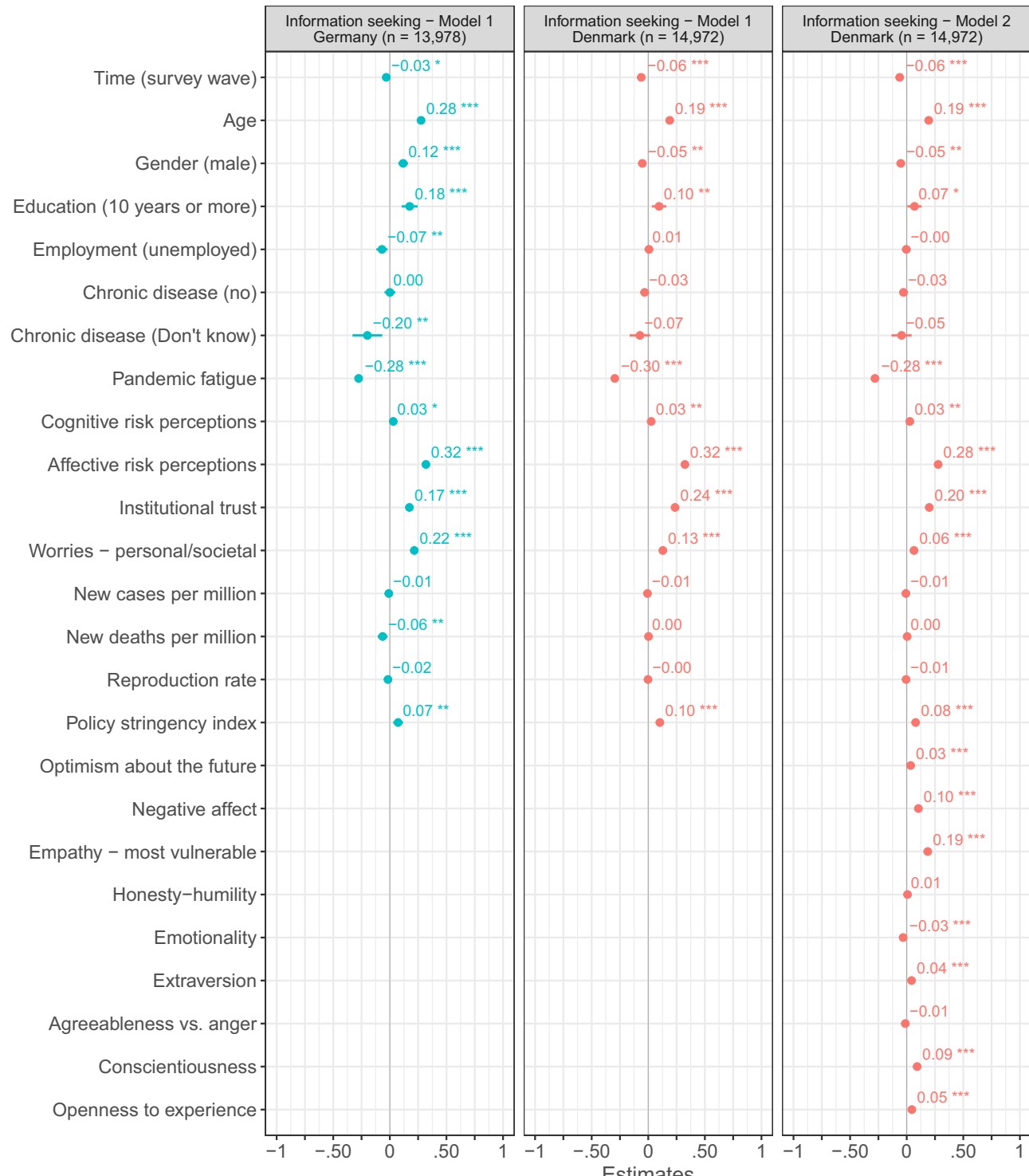

**Fig. 7 | OLS regressions predicting information seeking in Denmark and Germany.** Figure 7 shows standardized β coefficients with 95% confidence intervals based on ordinary least squares regressions with data from the Danish and German repeated cross-sectional surveys. All continuous predictors have been mean-centered and scaled by 1 standard deviation. The p-values have not been adjusted for multiple comparisons and are presented as follows: $^{***}p_{\text{two-tailed}} < 0.001$; $^{**}p_{\text{two-}}$ $_{\text{tailed}} < 0.01$; $^{*}p_{\text{two-tailed}} < 0.05$. Exact p-values for all models are presented in the R-output which has been deposited on the Open Science Framework at: https://doi.org/10.17605/OSF.IO/XD463. The gender variable refers to participants self-identified gender as presented to them in the surveys. Participants who did not identify as either male or female are not included in the analyses due to an insufficient number of observations.

course of a pandemic, not everyone is equally likely to experience it at all times. Third, pandemic fatigue is consistently related to people's self-reported tendency as well as their intention to adhere to various health-protective behaviors.

## Implications
Our findings suggest that pandemic fatigue is a real phenomenon that should not be disregarded. At the same time, our findings indicate that pandemic fatigue is one of many factors that relate to people's

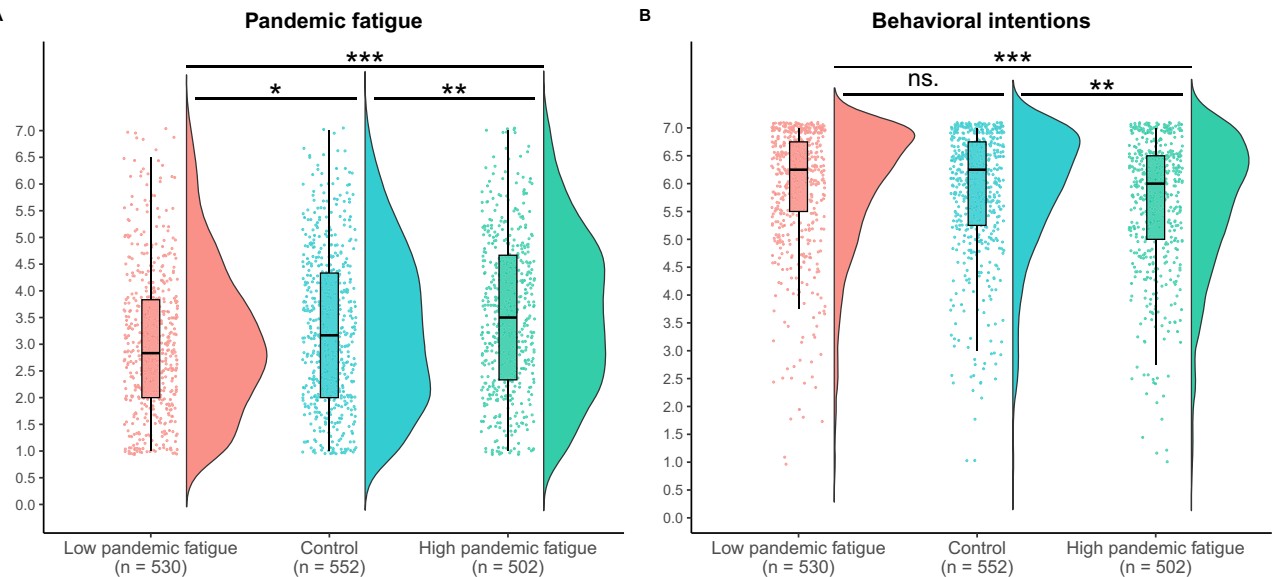

**Fig. 8 | Pandemic fatigue and behavioral intentions per condition.** Figure 8A shows raincloud plots of participants level of pandemic fatigue per experimental condition. Figure 8B shows raincloud plots of participants' intentions to comply with recommended health-protective behaviors per experimental condition. In Fig. 8A and Fig. 8B the boxplots show the 25th, 50th, and 75th percentiles of pandemic fatigue and participants' intentions to comply with recommended health-

protective behaviors, for each conditions, with whiskers extended to the most extreme data point that is no more than 1.50 times the interquartile range (i.e., Tukey style). The $p$-values have not been adjusted for multiple comparison and are presented as follows: $^{***}p_{\text{two-tailed}} < 0.001$; $^{**}p_{\text{two-tailed}} < 0.01$; $^{*}p_{\text{two-tailed}} < 0.05$; ns. $p_{\text{two-tailed}} > 0.05$. Exact $p$-values are presented in the R-output which has been deposited on the Open Science Framework at: https://doi.org/10.17605/OSF.IO/XD463.

tendency to adhere to recommended health-protective behaviors, making it crucial not to exaggerate its importance and lose sight of other (more) relevant factors. Keeping this in mind, interventions aimed at reducing pandemic fatigue could potentially still be useful, perhaps especially so if geared towards younger people who, on average, reported higher levels of pandemic fatigue. Because pandemic fatigue swiftly began to decrease in both Denmark and Germany as soon as the pandemic slowed down, however, the need for such interventions remains unclear. As an example, it might very well be that interventions aimed at reducing pandemic fatigue are largely unnecessary if each wave of the pandemic is short-lived, and people have enough time and are able to psychologically recover between waves. On the other hand, it could also be that people's experience of pandemic fatigue accumulates from one wave to the other, even if each wave of a pandemic is relatively short, making well-timed interventions aimed at reducing pandemic fatigue highly relevant. At this point we simply do not know. In order to provide clarity to this issue, future research should thus set out to critically investigate if and under what circumstances interventions aimed at reducing pandemic fatigue are (un)necessary, (in)effective, and (un)helpful.

Adding to this, our findings demonstrate that the PFS is an economic and valid measurement that may be used to monitor the development of pandemic fatigue during pandemics. Systematically monitoring pandemic fatigue within and across countries would not only provide additional insights into its nature, but also aid health authorities and policymakers in their assessment of whether interventions aimed at reducing pandemic fatigue might be necessary.

## Limitations

Some limitations of our research should be acknowledged. First, because our results exclusively rely on self-report data, it is unclear whether pandemic fatigue is related to and/or influences people's actual tendency to adhere to various health-protective behaviors. Yet, as both self-reports of past behavior[49] and behavioral intent[50,51] have been shown to correlate with actual behavior, it seems likely that our results conceptually capture the relation between pandemic fatigue and people's inclination to adhere to recommended health-protective behaviors.

Second, because our assessment of pandemic fatigue began several months into the COVID-19 pandemic, people's experience of pandemic fatigue probably had already increased as compared to their initial baseline at the onset of COVID-19. This may have limited the additional rise of pandemic fatigue, resulting in both smaller within- and between-subjects effects than one would otherwise had observed.

Third, even though we did find a robust link between pandemic fatigue and people's self-reported intention and tendency to adhere to various health-protective behaviors, this link was not particularly strong, typically yielding (very) small effect sizes. While some might argue that this renders the dawn and rise of pandemic fatigue inconsequential, it should be noted that (very) small effects can be cumulative in nature and can have important consequences in the long run and at scale[52]. Moreover, because all human behavior is driven by a multitude of factors, it is in most cases not only unrealistic but also unjustified to expect anyone of these factors to have a big impact by themselves only[53]. Indeed, all of the relations considered herein were found to be modest in nature.

Finally, while the experiment provides causal evidence for the link between the (de)motivational aspect of pandemic fatigue and people's intentions to adhere to various health-protective behaviors, it suffers from at least three limitations. First, the control condition, in which participants wrote about something ordinary, is likely to have elicited unintended feelings of (de)motivation and can therefore not be said to be perfectly neutral in terms of pandemic fatigue. Second, given the specific nature of the experimental manipulation used, we cannot rule out the possibility that the observed impact of pandemic fatigue on people's intention to adhere to recommended health-protective behaviors represents nothing more than an experimenter demand effect. Yet, it seems somewhat unlikely that this should be the case, given that experimenter demand effects tend to be fairly modest in size[54] and mostly non-existent in online survey experiments[55]. Third, for various reasons (see Methods), the experimental manipulation only targeted the (de)motivational aspect of pandemic fatigue, while not directly addressing the weariness and exhaustion related to it. Overcoming these limitations, future research might develop better and

more comprehensive manipulations of pandemic fatigue and test both their short- and long-term impact in realistic settings.

In conclusion, the existence of pandemic fatigue has been debated. Introducing a theoretical conceptualization and a corresponding measure of this elusive phenomenon, we provide evidence that not only speaks for the existence and nature of pandemic fatigue, but also broadens the understanding of the psychological and behavioral consequences of global pandemics.

## Methods

### Data sources
The present investigation relies on data from the COVID-19 Snapshot MOnitoring (COSMO) project[55]. Since March 2020, COSMO assessed citizens' knowledge, perceptions, emotions, and behavioral reactions related to COVID-19 across several countries. In Denmark[56,57] and Germany[58], a mixture of weekly, biweekly, and monthly repeated cross-sectional and (Denmark only) panel surveys were administered. More specifically, we use data from 25 waves of the Danish repeated cross-sectional survey (2020-10-19–2021-09-20), nine waves of the Danish panel survey (2020-10-19–2021-06-21), and 18 waves the German repeated cross-sectional survey (2020-10-27–2021-09-07). To control for the influence of time-dependent contextual factors (i.e., new COVID-19 cases and deaths per million, the COVID-19 reproduction rate, and policy stringency), we further rely on COVID-19 data from Our World in Data (https://ourworldindata.org)[59]. No statistical methods where used to predetermine the sample sizes for the Danish and German repeated cross-sectional surveys, nor the Danish panel survey.

### Procedure Danish repeated cross-sectional survey
In 2020, following data handling approval from the Faculty of Social Sciences of the University of Copenhagen (#514-0136/20-2000), the second author received contact information for two representative samples regarding age and gender of ~100,000 adult Danish citizens from Statistics Denmark (https://www.dst.dk/en). From these samples, random non-overlapping subsets of 5250–8500 Danes were invited via the official digital mail system in Denmark (https://www.e-boks.com/danmark/en) every other week from 2020-10-19 to 2021-09-20 to participate in the Danish repeated cross-sectional survey. The Danish repeated cross-sectional survey was set up and run in formr[60]. Participation was voluntary, and informed consent was obtained from all participants. Participants where not compensated for their participation. All participants who experienced technical issues while filling out the survey were excluded from the final dataset. The general study protocol for the Danish repeated cross-sectional survey as well as the Danish panel survey (see below; https://www.psycharchives.org/en/item/8a92091d-a1b6-42ac-ae53-7ca70ed2ccc2) received ethical approval from the Institutional Review Board at the Copenhagen Center for Social Data Science, University of Copenhagen.

A total of 15,985 respondents participated in the 25 waves of the Danish repeated cross-sectional survey considered herein without experiencing any technical issues (54.60% female, 45.18% male, 0.22% other; $M_{age} = 56.54$, $SD_{age} = 15.47$ years). Sociodemographic information for all participants in the Danish repeated cross-sectional survey is presented in Table S1. The response and completion rate for each wave of the Danish repeated cross-sectional survey considered herein is presented in Table S2. Across the 25 waves of the Danish repeated cross-sectional survey used for this investigation, some variables were assessed consistently, while others were only measured sporadically. Links to an overview of all variables assessed in the 25 waves of the Danish repeated cross-sectional survey can be found at: https://doi.org/10.17605/OSF.IO/XD463

### Procedure German repeated cross-sectional survey
The study obtained ethical clearance from the University of Erfurt Internal Review Board (#20200302/20200501), and all participants provided informed consent prior to participation. The study involved a weekly to fortnightly repeated cross-sectional survey with ~1000 non-overlapping individuals participating in each wave, using non-probability quota samples representative of the German population regarding age, gender, and federal state. The German repeated cross-sectional survey was set up and run using UNIPARK (https://www.unipark.com). Participants were compensated by the data collection company Respondi (https://www.respondi.com) for their participation. No participants were excluded from the final dataset. A total of 17,946 respondents participated in the 18 waves of the German repeated cross-sectional survey considered herein (50.69% female, 49.31% male; $M_{age} = 45.07$, $SD_{age} = 15.72$ years). The 18 waves of the German repeated cross-sectional survey used for this investigation were collected between 2020-10-27 and 2021-09-07. Sociodemographic information for all participants in the German repeated cross-sectional survey is presented in Table S1. As in the Danish repeated cross-sectional survey, some variables of the German repeated cross-sectional survey were measured consistently across all waves, while others were only assessed sporadically. An overview of all variables measured in the 18 waves of the German repeated cross-sectional survey can be found at: https://doi.org/10.23668/psycharchives.2776.

### Procedure Danish panel survey
Via the same procedure as for the Danish repeated cross-sectional survey, the second author received contact information for a representative sample regarding age and gender of ~100,000 adult Danish citizens from Statistics Denmark in 2018. From this sample, a random subset of 15,000 Danes was invited to participate in the Danish panel survey via the official digital mail system in Denmark. Like the Danish repeated cross-sectional survey, the Danish panel survey was set up and run in formr[60]. Participation was voluntary, and informed consent was obtained from all participants. Participants where compensated for their participation via a lottery in which they could win one of 30 vouchers worth 2000 DKK (approximately US $305 at the time of the study) each. A total of 2546 respondents participated in the first wave of the Danish panel survey and were thus invited to participate in the subsequent waves of the survey. Herein, we use data from waves 11–19 of the Danish panel survey which was collected between 2020-10-19 and 2021-06-21. Across these nine waves, between 341 and 438 respondents participated in each wave. All observations in which participants experienced technical issues while filling out the survey were excluded from the final dataset. Sociodemographic information for all participants in each of the nine waves of the Danish panel survey is presented in Table S3. As in both the Danish and German repeated cross-sectional surveys, some variables of the Danish panel survey were measured consistently across all waves, while others were only assessed sporadically. Links to an overview of all variables measured in the nine waves of the Danish panel survey can be found at: https://doi.org/10.17605/OSF.IO/XD463.

### Scales and measures
To best capture people's perceptions, emotions, and behavioral reactions to the COVID-19 pandemic, all COSMO surveys were specifically tailored to each country. Although there is a substantial overlap between the COSMO surveys conducted in Denmark and Germany, there are also some differences with regard to the content of the surveys as well as how certain variables were assessed. Across both countries, participants' cognitive and affective risk perceptions regarding COVID-19, their experiences of pandemic fatigue, and their chronic disease status were measured in the exact same manner. Participants' worries about potential personal and societal consequences of the pandemic, level of institutional trust, physical distancing, hygienic practices, mask wearing, information seeking, age, gender, education, and employment status (repeated cross-sectional surveys only) were also measured in both Denmark and Germany but with

slightly different items and/or response formats. Finally, respondent's feelings of optimism about the future, negative affect, and empathy towards those most vulnerable to COVID-19, as well as their personality characteristics in terms of the HEXACO dimensions were only assessed in Denmark. All variables, with the exception of sociodemographics (i.e., age, gender, education, employment, and chronic disease status), were measured with either a five- or seven-point Likert-type scale with different anchors. In both the Danish and German surveys, participants had the opportunity to answer 'Not relevant' or 'Don't know' to some items. In all cases, except for chronic disease status, we treated these responses as missing. Mean scores, standard deviations, and Cronbach's α for all scales considered herein can be found in Tables S4–S6. In Tables S7–S9 we further provide an overview of all scales and items from the Danish and German repeated cross-sectional survey, as well as the Danish panel survey used in this investigation.

### Detailed description of the development and validation of the pandemic fatigue scale (PFS)

**Item generation.** The item generation process consisted of five phases. At first, the first and last author each wrote seven or eight English items (15 items in total) that, in line with our conceptualization of pandemic fatigue, sought to capture a state of weariness and exhaustion from as well as a general demotivation towards following recommended health-protective behaviors, including keeping oneself informed about the pandemic (Phase 1). Next, the second and third author commented on the items and made suggestions on how to maximize their content validity (Phase 2). The first and last author then subsequently adapted the items in accordance with the comments and suggestions made by the second and third author (Phase 3). In accordance with the recommendations put forward by DeVellis[23], we removed any item that we (i.e., all four authors) perceived as overly redundant, lengthy, and/or difficult to read, leaving us with a final item pool of 10 items (Phase 4). Finally, the first and second author translated the items into Danish and German, respectively (Phase 5).

The final 10 items (Table S10) were administered in the 19th wave of the Danish repeated cross-sectional survey (2020-10-19–2020-10-25) in which 923 respondents participated. Notably, we only included this initial 10-item version of the PFS in one wave of the Danish repeated cross-sectional survey so as keep the length of this already extensive survey to a minimum and in turn reduce the risk of low-quality responses and survey length-related dropout in any of the subsequent waves[61,62]. Items were answered on a 7-point Likert scale ranging from 1 = "Strongly disagree" to 7 = "Strongly agree".

**Exploratory factor analysis.** For all items, no sign of severe univariate nonnormality was observed (i.e., skewness <2.0 and kurtosis <7.0)[63]. On the other hand, Mardia's multivariate tests[64] indicated that the items were multivariate nonnormal (multivariate skewness = 9.01, $p < 0.001$; multivariate kurtosis = 153.84, $p < 0.001$). To explore the factor structure of the initial 10-item PFS, we thus conducted an exploratory factor analysis using an ordinary least squares approach[65,66] because this approach, in contrast to maximum likelihood estimation, makes no multivariate distributional assumptions about the data[67]. Considering the fact that most factors are correlated[68], we opted for an oblique factor rotation, namely, oblimin[67,69]. In line with previous research suggesting that it is often reasonable to treat ordinal data as continuous[70,71], particularly when more than five response categories are used[72,73], we treated the data as continuous and conducted the exploratory factor analysis on the basis of Pearson product-moment correlations. For completeness and recognizing that treating ordinal data as continuous may introduce bias[74–76], we also report the results of an exploratory factor analysis based on polychoric correlations in the Supplementary Information (Supplementary Note 2). Notably, the exploratory factor analysis

based on polychoric correlations yield qualitative similar results to that based on Pearson product-moment correlations.

The sampling adequacy of the data was verified using the Kaiser-Meyer-Olkin test[77] and found to be acceptable (overall KMO = 0.92; all KMO values for individual items are > 0.83). Bartlett's test of sphericity[78] further indicated that the item correlations were sufficiently large for conducting an exploratory factor analysis ($X^2(45) = 4267.31$, $p < 0.001$). To determine the number of factors to extract, we considered the scree test[79], Glorfeld's modified parallel analysis[80,81], the very simple structure criterion[82], and the Velicer's minimum average partial criterion[83], which, in combination, indicated that either a one- or a two-factor solution would best reflect the data (Fig. S21 and Table S11).

Considering both a one- and a two-factor solution, the exploratory factor analysis revealed that a two-factor model fit the data better (RMSR = 0.02, RMSEA = 0.05, TLI = 0.98) than a one-factor model (RMSR = 0.07, RMSEA = 0.13; TLI = 0.84), as indicated by the RMSR being closer to zero[67], a difference in RMSEA > 0.015[84], and a TLI above 0.95[85]. The two-factor model explained 54.28% of the variance, with the first factor accounting for 19.33% of the variance and the second factor 34.95%. Assessing the items pertaining to each factor, the first factor, consisting of three items, represented what we termed 'information fatigue' (i.e., feeling exhausted from and demotivated towards keeping oneself informed about the pandemic). The second factor, consisting of seven items, largely represented what we termed 'behavioral fatigue' (i.e., feeling exhausted from and demotivated towards following recommend health-protective behaviors).

As our goal was to develop a brief pandemic fatigue scale, we reduced the number of items of the second factor (i.e., the behavioral fatigue factor) by iteratively removing one item at a time until the scale had been reduced to three items per factor. At this juncture, we simultaneously considered factor loadings, cross-loadings, and the content of each item to ensure that the final scale would have good psychometric properties and high content validity[86]. That is, we sequentially removed the item with the lowest factor loading and highest cross-loading while also considering if the content validity of the behavioral fatigue factor would be reduced by removing the item in question. The final two-factor model with three items per factor fit the data well (RMSR = 0.01, RMSEA = 0.02, TLI = 1.00), and explained 57.74% of the variance, with the information fatigue factor accounting for 30.49% of the variance and the behavioral fatigue factor 27.25%. Standardized factor loadings, communalities, uniqueness, and complexity for the final two-factor model are presented in Table 1 together with the six items retained. The correlation between the initial 10-item PFS and the final six-item PFS was very high ($r(921) = 0.96$, $p_{two-tailed} < 0.001$).

**Confirmatory factor analysis.** To validate our findings from the exploratory factor analysis, we conducted a confirmatory factor analysis with pooled data from waves 20 to 43 (16-11-2020–20-09-2021) of the Danish repeated cross-sectional survey ($n = 15,062$), and all 18 waves (27-10-2020–07-09-2021) of the German repeated cross-sectional survey considered herein ($n = 17,946$). The data showed no signs of severe univariate nonnormality (i.e., skewness <2.0 and kurtosis <7.0)[63], but was multivariate nonnormal in both Denmark (multivariate skewness = 3.04, $p < 0.001$; multivariate kurtosis = 55.46, $p < 0.001$) and Germany (multivariate skewness = 2.89, $p < 0.001$; multivariate kurtosis = 57.04, $p < 0.001$), as indicated by Mardia's tests[64]. Similar to the exploratory factor analysis, we treated the data as continuous, but estimated all models using robust maximum likelihood estimation with robust standard errors and a Satorra-Bentler scaled test statistic[87] to account for the multivariate nonnormality of the data[88]. For completeness and recognizing that treating ordinal data as continuous may introduce bias even when using robust maximum likelihood estimation[89], we additionally fitted all models treating the data as ordinal, using robust diagonally weighted least squares

estimation[88] (see Supplementary Note 3). Notably, the two estimation methods yielded qualitative similar results. To evaluate the models, we relied on robust versions[90,91] of the following fit indices and recommended cutoff values[92]: RMSEA ≤ 0.06, SRMR ≤ 0.08, TLI ≥ 0.95, and CFI ≥ 0.95.

Results indicated that a two-factor model fit the data well in both Denmark (RMSEA = 0.06, SRMR = 0.03, TLI = 0.97, CFI = 0.99) and Germany (RMSEA = 0.07, SRMR = 0.03, TLI = 0.98, CFI = 0.99). The two factors were found to be strongly correlated ($r_{\text{Denmark}} = 0.69$, $p_{\text{two-tailed}} < 0.001$; $r_{\text{Germany}} = 0.78$, $p_{\text{two-tailed}} < 0.001$), however, pointing to the possibility that a one-factor model would fit the data better. To explore this possibility, we fitted a one-factor model. In both Denmark (RMSEA = 0.18, SRMR = 0.08, TLI = 0.80, CFI = 0.88) and Germany (RMSEA = 0.16, SRMR = 0.06, TLI = 0.86, CFI = 0.92) a one-factor model did not fit the data well. Considering the bad fit of the one-factor model and the high factor intercorrelation of the two-factor model, we decided to model pandemic fatigue as a second-order latent construct with information and behavioral fatigue as first-order sub-factors. While the second-order model is statistically equivalent to the two-factor model—and thus fits the data equally well−, it has two advantages: It allows for the combination of the information and behavioral fatigue factors into an overall and parsimonious measure of pandemic fatigue, while at the same time making it possible to explore the relations of these two factors with other variables separately. The fully standardized factor loadings and (residual) variances for both the two-factor and second-order models are presented in Fig. 9. Finally, to test the robustness of the second-order model across different waves of the Danish and German repeated cross-sectional surveys, we re-fitted this model for each survey wave−except wave 19th of the Danish repeated cross-sectional which was used for the exploratory factor analysis−using both robust maximum likelihood estimation and robust diagonally weighted least squares estimation. By and large, the results from this analysis (across 84 models) suggest that the proposed second-order model of pandemic fatigue is robust across waves in both the Danish (RMSEA = 0.03 to 0.15, SRMR = 0.02 to 0.05, TLI = 0.93 to 1, CFI = 0.96 to 1) and German (RMSEA = 0.03 to 0.11, SRMR = 0.02 to 0.04, TLI = 0.95 to 1, CFI = 0.98 to 1) repeated cross-sectional surveys (see Tables S12–S15).

**Internal consistency.** In both Denmark and Germany, the internal consistency of the full PFS (Cronbach's $\alpha = 0.83/0.86$, McDonald's $\omega = 0.82/0.88$) as well as of the information ($\alpha = 0.83/0.84$, $\omega = 0.83/0.84$) and behavioral fatigue ($\alpha = 0.73/0.77$, $\omega = 0.73/0.77$) subscales was acceptable.

**Measurement invariance testing.** To ensure that the PFS measured pandemic fatigue similarly across Denmark and Germany, we tested for measurement invariance by fitting and comparing the fit of several multi-group confirmatory factor analyses with different levels of equality constraints using robust maximum likelihood estimation with robust standard errors and a Satorra-Bentler scaled test statistic[87]. Importantly, in all cases we relied on the identification strategy proposed by Yoon and Millsap[93] and compared the fit of the models using Cheung and Rensvold's[94] ΔCFI < −0.01 criterion. We used this identification strategy because it circumvents the problem of having to choose an arbitrary reference item which is otherwise required when using the standard marker method for identification[93]. Moreover, we rely on Cheung and Rensvold's (2002) ΔCFI < −0.01 criterion rather than the commonly used criterion of significant differences in $X^2$, because the significant differences in $X^2$ criterion is sample size dependent and overly sensitive for large samples[94]. As for the other confirmatory factor analyses conducted herein, we acknowledge that treating ordinal data as continuous may introduce bias[89] and therefore report results from corresponding analyses in which we treat the data as ordinal using robust diagonally weighted least squares estimation[88]

in the Supplementary Information (Supplementary Note 4). Notably, we find similar levels of measurement invariance irrespective of how we treat the data.

Testing for configural invariance, we first fitted a multi-group confirmatory factor analysis with no equality constraints across countries. This model fit the data well suggesting that the PFS is configurally invariant across Denmark and Germany (RMSEA = 0.06 SRMR = 0.02, TLI = 0.98, CFI = 0.99). Next, we tested for metric invariance by constraining the factor loadings across countries to equality and comparing the fit of this constrained model to the fit of the first model with no equality constraints. Comparing the fit of these two models, we find the PFS to be metrically non-invariant across Denmark and Germany (ΔCFI > −0.01). In light of these results, we turned to test for partial metric invariance by freeing the factor loadings of the fourth item of the PFS (i.e., "I feel strained from following all of the behavioral regulations and recommendations around COVID-19"). Freeing the factor loadings of the fourth item and comparing the fit of this third partially constrained model to the fit of the first model with no equality constraints, we find support for partial metric invariance of the PFS (ΔCFI < −0.001). As a final step, we proceeded to test for partial scalar invariance by additionally constraining the item intercepts across countries to equality−except the intercept of the fourth item−and comparing the fit of this additionally constrained fourth model to the fit of the less constrained third model. Comparing the fit of these two models, we find support for partial scalar invariance of the PFS (ΔCFI = −0.005). Taken together, these results indicate that the PFS measures pandemic fatigue in a similar manner across Denmark and Germany.

## The online experiment

**Procedure.** The online experiment was preregistered via aspredidcted.org on 2021-01-28 (see https://aspredicted.org/ua3ca.pdf) and set up and run in formr (https://formr.org)[60]. All confirmatory analyses correspond to the preregistered analysis plan. Ethical clearance was obtained from the Institutional Review Board at the Department of Psychology, University of Copenhagen (#IP-IRB/22012021). All participants provided informed consent prior to participation. The experiment took approximately seven minutes to complete, and participants were paid a flat fee of £0.75 for their participation. In the first part of the experiment, all participants were asked to provide information about their age, gender, and education, as well as to respond to two items assessing their cognitive risk perceptions regarding COVID-19 (i.e., "How likely do you think it is that you will be infected with the novel coronavirus (COVID-19)?" and "How serious would it be for you if you contracted the novel coronavirus (COVID-19)?"). Next, they were all randomized into one of three conditions−control, low, and high pandemic fatigue−and asked to complete a brief self-reflection task designed to manipulate their experience of pandemic fatigue (see Wildschut et al., for a similar self-reflection task)[95] by specifically targeting the (de)motivational aspect of pandemic fatigue (i.e., feeling demotivated towards following recommended health-protective behaviors, including keeping oneself informed about the pandemic). In particular, participants in the low/high pandemic fatigue condition were presented with the following instruction: "Using the space provided below, please spend the next few minutes to describe some of the things that, over the last two weeks, have motivated/demotivated you to follow recommended protective behaviors (e.g., physical distancing, mask wearing, hygienic practices) and keep yourself informed about the COVID-19 pandemic". In contrast, participants in the Control condition were given the following instruction: "Using the space provided below, please spend the next few minutes to describe some of the ordinary things that have happened over the last two weeks and affected your behavior in some way".

The decision to focus on the (de)motivational aspect of pandemic fatigue was made on the basis of both methodological and theoretical considerations. First and foremost, we decided to focus on the (de)

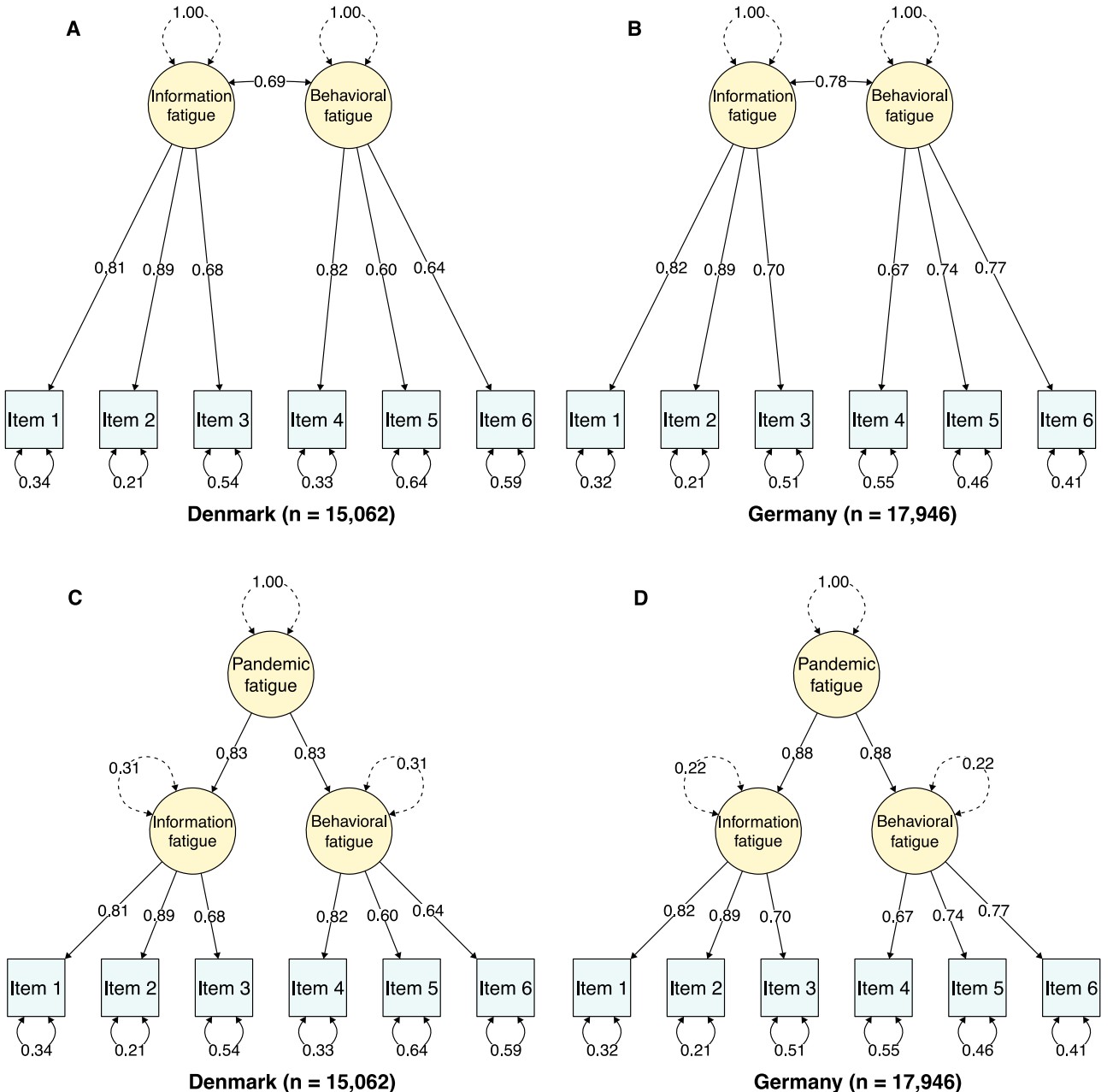

**Fig. 9 | Two-factor and second-order models of pandemic fatigue. A, B** The two-factor model of pandemic fatigue with fully standardized factor loadings and (residual) variances for Denmark and Germany, respectively. **C, D** The second-order model of pandemic fatigue with fully standardized factor loadings and (residual) variances for Denmark and Germany, respectively. All models were estimated using robust maximum likelihood estimation with robust standard errors and a Satorra-Bentler scaled test statistic[87]. Item 1 = 'I am tired of all the COVID-19 discussions in TV shows, newspapers, and radio programs, etc.'; Item 2 = 'I am sick of hearing about COVID-19'; Item 3 = 'When friends or family members talk about COVID-19, I try to change the subject because I do not want to talk about it anymore'; Item 4 = 'I feel strained from following all of the behavioral regulations and recommendations around COVID-19'; Item 5 = 'I am tired of restraining myself to save those who are most vulnerable to COVID-19'; Item 6 = 'I am losing my spirit to fight against COVID-19'. Response scale: 1 = strongly disagree, 2 = disagree, 3 = somewhat disagree, 4 = neutral/neither disagree nor agree, 5 = somewhat agree, 6 = agree, 7 = strongly agree.

motivational aspect of pandemic fatigue because it allowed us to straightforwardly manipulate the experience of pandemic fatigue in opposite directions by simply asking participants to reflect upon what motivated/demotivated them to adhere to recommended health-protective behaviors in the past few weeks. Second, we focused on this aspect because the feeling of weariness and exhaustion that also characterizes pandemic fatigue arguably is more perennial in nature and thus less susceptible to undergo rapid changes in response to time varying situational factors, including that of simple experimental manipulations.

Finally, after completing the brief self-reflection task, all participants were asked to complete the PFS and to respond to four items assessing their intention to adhere to recommendations regarding physical distancing (i.e., "Over the next two weeks I will avoid physical contacts and keep a safe distance to people outside my own household"), hygienic practices ("Over the next two weeks I will wash my hands very often and thoroughly and/or use hand disinfectant frequently"), and mask wearing ("Over the next two weeks I will wear a face mask whenever I am inside and cannot keep a safe physical distance to people outside my own household"), as well as to keep

themselves informed about the pandemic and current COVID-19 restrictions ("Over the next two weeks I will do everything I can to keep myself updated about the development of the pandemic, and stay informed about the current COVID-19 restrictions"). Both the PFS and the four items assessing participants' intentions to adhere to recommendations regarding physical distancing, hygienic practices, and mask wearing as well as to keep themselves informed about the pandemic and current COVID-19 restrictions were answered on a 7-point Likert scale ranging from 1 = "Strongly disagree" to 7 = "Strongly agree". Mean scores, standard deviations, and Cronbach's α for all measures obtained in the experiment are presented in Table S16. An overview of all items and scales used in the experiment is available in Table S17.

**Power analysis.** To determine an appropriate sample size for the experiment, we conducted an a priori power analysis based on results from a pilot study designed to test our experimental manipulation ($n = 299$) using G*Power[96]. Aiming to be able to detect a small effect size (Cohen's $d = 0.20$) in an independent samples $t$-test with a two-tailed alpha level of 0.05 and high statistical power (1- β = 0.90), the a priori power analysis revealed that a total of 1581 participants would be sufficient (i.e., 527 participants per condition). To compensate for potential exclusions, we decided to oversample by ~15% and thus aimed to recruit a total of 1850 participants.

**Participants.** In line with the results from the a priori power analysis, a total of 1854 participants from the U.S. were recruited via Prolific (https://www.prolific.co) to participate in the experiment. Of these, a total of 270 participants were excluded based on our a priori exclusion criteria (see https://aspredicted.org/ua3ca.pdf), resulting in a final sample of 1584 (50.32% female, 47.98% male, 1.70% other; $M_{age} = 35.58$, $SD_{age} = 11.87$ years). Sociodemographic information for each of the three conditions can be found in Table S18.

**Reporting summary**
Further information on research design is available in the Nature Portfolio Reporting Summary linked to this article.

## Data availability
The raw data from the online experiment and the Danish and German repeated cross-sectional surveys used herein have been deposited on the Open Science Framework at: (https://doi.org/10.17605/OSF.IO/XD463). Please note that we—in line with the European General Data Protection Regulation—are unable to publicly share the raw data of the Danish panel survey because it contains personal identifiers that were linked to sensitive personal information (even though the data are stored in a (pseudo)anonymized format now). Instead, we provide an exemplary synthetic version of this data created with the synthpop package in R[97] on the Open Science Framework: (https://doi.org/10.17605/OSF.IO/XD463). Raw data from the Danish panel survey is available upon request via llj@psy.ku.dk, but only after an appropriate data processing agreement can and has been signed. The data obtained from Our World in Data is available at: https://ourworldindata.org/coronavirus.

## Code availability
Code for replicating the results, tables, and figures presented herein are available via the Open Science Framework at: (https://doi.org/10.17605/OSF.IO/XD463).

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

## Acknowledgements

The Danish COVID-19 Snapshot Monitoring (COSMO) project was funded by grants from both the Lundbeck Foundation (R349-2020-592) and the Faculty of Social Sciences, University of Copenhagen (Denmark) to I.Z. and R.B. The German COVID-19 Snapshot Monitoring (COSMO) is a joint project of the University of Erfurt (Cornelia Betsch [PI], Lars Korn, Philipp Sprengholz, Philipp Schmid, Lisa Felgendreff, Sarah Eitze), the Robert Koch-Institute (RKI; Lothar H. Wieler, Patrick Schmich), the Federal Centre for Health Education (BzgA; Heidrun Thaiss, Freia De Bock), the Leibniz Centre for Psychological Information and Documentation (ZPID; Michael Bosnjak), the Science Media Center (SMC; Volker Stollorz), the Bernhard Nocht Institute for Tropical Medicine (BNITM; Michael Ramharter), and the Yale Institute for Global Health (Saad Omer). The study was funded by the German Research Foundation (BE3970/11-1, 12-1 to C.B.), University of Erfurt, Robert Koch-Institute, Leibniz Institute for Psychology Information, Federal Centre for Health Education.

## Author contributions

All: Conceptualization, Methodology, Writing—Review and Editing, Project administration, Investigation, Resources; L.L.: Writing—Original Draft, Visualization, Software, Validation, Formal analysis, Data Curation; Cornelia Betsch, Robert Böhm and Ingo Zettler: Supervision, Funding acquisition. The data from https://ourworldindata.org was accessed by L.L.

## Competing interests

The authors declare no competing interests.
