## [Peer Review File · Nature Communications]

Development and Validation of the Pandemic Fatigue Scale (PFS)Reviewer #1 (Remarks to the Author):

Pandemic fatigue has been a much-debated topic, and empirical work has been in many ways held back (or limited in what it dares to claim) by the absence of conceptual agreement. As such, I appreciate the task that this paper has set for itself in trying to move the field on.

Introduction:

In general, I think the paper's brief coverage of the different definitions of fatigue is good, and I am convinced that the authors justify their approach to define pandemic fatigue as a gradually emerging subjective state.

I would like to see a bit more probing of the idea baked into the WHO definition that fatigue also shows up as how much people keep themselves informed about the pandemic, in the sense that there will surely be a learning curve to general information at the start (answering questions like "How does one protect oneself?" "Who is most at risk?" etc) that is unlikely to require many updates over time, and so information seeking should reduce in the absence of fatigue. Put differently, it seems likely that an unfatigued person would not uniformly seek the same amount of information about the pandemic over time. This needs addressing theoretically, and picking up later in the manuscript when the metrics regarding information seeking are discussed.

More specific notes:

p4 , line 58 - I appreciate the use of "arguably" but I also think that this sentence would benefit from including the point that some of the studies cited at the end of this sentence are valuable precisely because they have sought to control for the "other factors" to which the authors are referring. And so while some of these studies do not claim to find direct evidence of pandemic fatigue, they show that other factors are unlikely explanations for changes in adherence. This part of the manuscript reads as though the authors are selling some of the previous contributions rather short, as though no one has made progress, and thus setting the stage for their own contribution to seem all the larger.

P4, line 69-70 - this reads as though the panel surveys were conducted in both countries.

Results

Overall, I think this section is reasonably strong. It is interesting that the repeated cross-sectional surveys in more than one country showed similar patterns, and that the models with quadratic fit had greatest model fit.

There are a few issues that need addressing, however:

Picking up on the point about unfatigued patterns in information seeking, I wasn't clear on reading whether initial levels of compliance (to the protective behaviours) and information seeking were taken into account in the analysis. For example, typically women initially adhere more than men to physically distancing, and so there is logically more likelihood of behavioural fatigue. Interpreting the authors' findings for those with less education (p9, line 161) would also benefit from a clarification here: how did the baseline level of information seeking differ between the highly educated and the less educated?

Similarly, the authors report a large negative coefficient for "keeping themselves informed about the pandemic" (e.g. p11, line 217). How does the size of this coefficient relate to my earlier point about an unfatigued person probably not uniformly seeking the same amount of information about the pandemic over time?

The other key issue for clarification relates to the authors' treatments of the 2 subfactors - information and behavioural fatigue - throughout the results section. There

is the 6-item PFS and the 4-items used, for example, in the experiment. In the experiment, the authors create a composite score of the 4 items (3 of which are protective behaviours, and 1 is information seeking). Are the 4 items given equal weight in this score? Or is information seeking is a quarter of this score, rather than treated as equally as important to behavioural fatigue - as implied by the definition provided in the introduction (and again in the discussion, p15, line 295 - "two distinct factors"). Does the weighting in the composite score between behavioral and information fatigue align with the PFS (as it should)? I found myself looking for this in the Methods section, without success.

More specific notes:

Nb p7, line 123 - "hearing about" is different from "seeking information about".

Elsewhere in the manuscript the authors discuss the seeking of information.

P8, lines 141, 142 - wording choice: "whereafter" and "wean off" sound odd ("after which" and "wear off"?).

P8, line 145-6 - again the sentence structure suggests there were both German and Danish panel surveys.

P8, line 155 - some of the emotions / perceptions are incompletely described, which makes the results hard to fully understand, and to fully make sense of how they fit into the literature at large. For example: are the authors referring to institutional or generalised trust? (This is important, noting that others have found different protective behaviour adherence patterns over time based on which of these kinds of trust is brought into focus). Similarly: optimism, or optimism bias?

P11, line 222-223 - I'd like to see some examples of the controls used. The case or death rate would be a key control, for example.

Causal impact study framing:

I find the description of the "low fatigue" treatment completely misleading. If anything it's the opposite of a fatigue treatment - i.e. it's a motivating treatment.

Similarly the control does not seem to be true control - a task that takes the same effort in an experiment but that is otherwise neutral with regard to the treatment. This is because asking people to write about what has "affected their everyday behaviour" is likely to have a motivating or (perhaps more likely) demotivating element to it. In any case, it's not neutral.

The above two points do not imply to me that the study isn't valid, but that its framing needs a lot of work. Adjusting this framing will also help make sense of findings such as "participants in the control condition reported higher levels of pandemic fatigue than participants in the low pandemic fatigue condition", which otherwise sound nonsensical.

Discussion

p15, line 312-314 - I would like to see the finding "other factors, such as people's affective risk perceptions regarding the pandemic, generally seem to matter as much or even more than pandemic fatigue..." given more room in the Results section. It may be worth a main figure.

More specific note:

p16, line 320-327 - I enjoyed this point of speculation, but I think the authors should make it a bit clearer that their findings don't provide much insight into rates of pandemic fatigue recovery, and that this is mere speculation.

Materials & Methods

I would also like to know more about the 6-item PFS here, and how it incorporates the two subfactors. This is crucial information to make sense and compare with the 4-item composite scores elsewhere.

As with all pandemic surveys, but especially anything that seeks to assess fatigue, I would like to know the rate at which people who were invited to participate (in the cross-sectional surveys) did so, as well as the non-completion rate of the survey, and

whether these two details changed over time. Certainly, the attrition of the panel survey (p19, line 407) suggests this information could be important.

Figures

Figure 1 - I recommend showing the waves of the pandemic on these charts as a second y axis. (Smoothed death rate and/or case rate)

Reviewer #2 (Remarks to the Author):

This is a very timely paper investigating the topic of pandemic fatigue, seeing how it waxes and wanes over the course of the pandemic, and what psychological factors vary with it in 3 very large datasets. In a final preregistered experiment, the authors find evidence that reminders of being fatigued of the pandemic decreased self-reported intentions to adhere to pandemic restrictions.

This is an excellent paper and I have very little to criticize. I think it is very well conducted, and also important for public policy. While I think this should be published, I have a few smaller concerns.

First, where is the pandemic fatigue scale? I can't find it anywhere in the manuscript. Maybe it's buried in the supplement, but I could not find it. The short final version of the scale should be in the manuscript itself so that others can freely use it.

Second, given the large samples, significance is less critical than effect sizes. The authors do not really talk about effect sizes though they occasionally include statements that the effect is large, pandemic fatigue thus seems to have risen quite dramatically" (line 140). Is the rise in pandemic fatigue actually dramatic? The effects seem small to me, albeit important, and the authors should explicitly talk about them.

Third, I found the experiment unconvincing. The authors essentially show that when they remind people of moments when they felt overwhelmed and demotivated by the pandemic, they report feeling more demotivated (i.e., the pandemic fatigue scale measures this) and less likely to adhere to pandemic measures (i.e., which might also be part of the pandemic fatigue scale; though I can't tell because I don't have access to the scale itself). There is a lot of circularity in this study. Basically, the authors manipulate pandemic fatigue and then measure fatigue in various ways. Related, how did the authors measure adherence to pandemic measures in the experiment? This too should be listed in the manuscript so that readers can see the content overlap between the pandemic fatigue scale and the outcome measures. In my eyes, this study is not particularly informative though I would not recommend removing it, but perhaps speaking about what it can and cannot say.

Fourth, I was surprised that the authors did not measure political orientation, as it seems to be a pretty large predictor of adherence to pandemic measures. Is such a measure present in the dataset? If so, I wonder if the authors would consider analyzing it.

In sum, I really like this paper and hope my comments can be of some help to the authors.

I sign all my reviews,

Michael Inzlicht

Reviewer #3 (Remarks to the Author):

The Results start with some type of modelling, with no outline of the statistical technique, software used or thresholds for model fit. The Methods must describe the reason for choosing such models, including the relevance of the distribution of the data, including whether the models accounted for the (likely) need to use polychoric correlations. There is no specific reference to the construction of the Pandemic Fatigue Scale (PFS) although other scales with the same name appear in the literature. This needs to be clarified.

Before a questionnaire can be used to make comparisons between groups, measurement invariance needs to be established. I don't see evidence that the 2 scales can be added. What is the correlation between these 2 scales, and was a 2-factor Confirmatory Factor Analysis undertaken?

What is the point of modelling the panel data over time with a quadratic term? What of different or concurrent secular trends in different/each country? Different sampling (demographics) at each panel? I am not sure how meaningful these findings are without a strong analysis of secular trends.

L140 – on what basis is the term 'dramatically' used? Figure 1 is not informative as nowhere is the scale described (response options, range of scores on the 2 scales), direction, and no sample sizes. The authors assume fatigue 'develops' – but the figures are not really showing 'development' to me – the title doesn't seem to represent the data.

L145 – Again, speculation that data from 2 countries being somewhat similar over time without careful examination of secular events (prevalence of infections, deaths, health policy, and the multitude of government announcements) is not convincing.

L159 – As noted above, before countries can be compared, configural invariance for factor structures, metric invariance for factor loadings, and scalar invariance for item intercepts need to be established for a scale.

L163 – The cross sectional surveys cannot be used to draw causal conclusions, i.e., a high or low score on the scale is only correlated with a score on another scale. The language throughout the paper needs to reflect the direction of causality is not known. Fig 2. With very large sample sizes it is relatively easy to find lots of statistically significant associations – but they are meaningless in magnitude from the public health, policy and clinical perspectives. The authors need to make clear what associations are of a magnitude that is important from the public health or clinical perspective.

The US study of 1854 which attempts to manipulate respondents through a writing experience seems incoherent with the rest of the manuscript. Seems odd that this US study was approved by a Danish ethics study – this needs to be clarified. The sampling and representativeness of the participants are unclear. Seems to be convenience sampling.

Reviewer #4 (Remarks to the Author):

The manuscript "Pandemic Fatigue: Measurement, Correlates, and Consequences" aims to conceptualize, operationalize, and understand the antecedents/consequences of the construct "pandemic fatigue" with respect to the COVID-19 global pandemic.

I thought the manuscript had many strengths. The authors do a nice job driving home why practically the above listed research objectives are important, given the common use of the term "pandemic fatigue" in the press and political discourse about the pandemic, as well the underdeveloped conceptualization/operationalization of the construct in the empirical literature thus far. I was convinced that empirically sound research speaking to the conceptualization, operationalization and antecedents/consequences of 'pandemic fatigue' is worthwhile, and will be of broad interest both to researchers studying pandemic response, as well as those in applied roles responding to the pandemic.

I also appreciated the manuscript in terms of its strong empirical/methodological strengths. The researchers embraced open science techniques (i.e., pre-registration, open methods, and data), used very transparent figures to show their data distributions and effect sizes, and used sound analytic strategies in their EFA, CFA and multilevel analyses. I also liked the general tone the authors used to describe and interpret their research findings – I did not feel like they had a “horse in the race” to make the concept of pandemic fatigue the most important construct to consider in the context of the pandemic, but rather tried to provide a sound measure to study the construct, and well-powered longitudinal studies to understand what it can (can’t predict) relative to other important psychological constructs in the context of the pandemic.

At the same time, I do have some comments, questions, and suggestions for the authors that I feel could further improve this work if considered. While I will dig into the specifics more below, a lot of my comments have to do with further expanding the conceptual section of the manuscript in terms of discussing what exactly “pandemic fatigue” is, and being more explicit and detailed in the main parts of the manuscript about how the authors’ conceptualization of the construct was translated to their operationalization of it, both in terms of their six item measure and their experimental manipulation. As I will describe, I really liked the 6-item measure and I feel it taps into a unique construct beyond just demotivation or amotivation to follow pandemic related measures – but I was less certain that the empirical manipulation taps into the same construct as the scale (and something beyond just demotivation). Relatedly, a lot of new ideas seemed to sprout up in the empiric sections without much theoretical setup in the lead in (e.g., the two distinct factors of pandemic fatigue and distinctions between seemingly similar constructs like affective/cognitive risk perception vs. pandemic worries were never really mentioned prior to their appearance in the methods/results). I think these concepts and their relevance to the focal research objectives need to be set up early on in the intro section for readers to understand these distinct yet similar concepts later, and how they apply to the authors’ overarching theoretical framework.

All this said, I also know that general science outlets like Nature Communications have less room for these types of details/nuance and I emphasize with the authors in the challenge of getting this nuance into this format. Hopefully, a happy medium can be found, as I think this focus on conceptualization to operationalization is especially important for measurement focused work. I also think that more conceptualization/positioning into the theoretical lit both in the intro and general discussion could help the paper make broader theoretical impacts beyond its clear applied impact.

I wish the authors all of the best with this important line of work.

Detailed Comments:

1. I think it will be important to further develop the “Conceptualizing and Defining Pandemic Fatigue” section of the introduction. I enjoyed how the authors began with past work on the concept of fatigue, and then also made important distinctions between physical depletion /objective lack of ability to continue to do something and subjective feelings of exhaustion (intertwined with demotivation). Where I would like to see more expansion is as follows:

A. The eventual measure of pandemic fatigue had two components – information fatigue and behavioral fatigue. Both of these components are inherent to the authors definition: “A gradually emerging subjective state of weariness and exhaustion from, and general demotivation towards, following recommended health-protective behaviors, including keeping oneself informed about the pandemic.” But I think the authors should make it more clear that they actually conceptualize the information seeking vs. behavioral adherence components as distinct aspects, and root this both in terms of qualitative descriptions of what people did during the pandemic as well as literature that might make distinctions between information seeking / vs. protective behavior in response to

major threats like a pandemic.

B. In reading the scale items of the construct I found it interesting how elements of exhaustion / fatigue were intertwined with the protective behaviors / information seeking people might do in response to the pandemic. This in a way seemed something distinct from just “general demotivation” that the authors describe in their definition beyond the fatigue elements. But on the other hand, the empirical manipulation the authors used really left the fatigue elements behind and centered on the motivated vs. demotivated distinction. So this made me still wonder about what exactly pandemic fatigue is – is it demotivation or feelings of exhaustion, or both? The authors hinted at this tension a bit in the GD when they wrote : “critics of pandemic fatigue might argue that pandemic fatigue, both from a theoretical and practical point of view, is a superfluous construct that represents nothing more than people’s motivation to adhere to different health-protective behaviors” (p17). I would say this critique is much more true of the experiment than the psychological measure presented in the work. All this said, I would like to see more explicit discussion and positioning of how pandemic fatigue is similar versus different from (de)motivation in the conceptual section of the intro to then set up both the measure and the experiment. It might be tough though – because both parts of the paper seem to take a different operational approach with the scale focusing on fatigue and the manipulation focusing on (de)motivation.

C. I wonder if the authors should incorporate other relevant work/constructs into their conceptualization of pandemic fatigue. Although I don’t know a lot about the concept, I think there has been a lot of research on “psychological burnout” within organizational psychology – and this idea of pandemic fatigue made me think of this.

Given the intersection of pandemic fatigue with motivation, I also wondered about the applicability/relevance of self-determination theory which talks about different types of motivation people may have (beyond amount). From SDT, motivation can range from amotivation, to controlled and introjected motivation (doing something to avoid guilt, punishment, or to look good), to autonomous forms of motivation where people value and see the importance of doing something (even if its tough/unpleasant). I thought of SDT especially because when people have more autonomous motivation, they tend to feel less exhausted and more vital when self-regulating (Milyavskaya, Inzlicht, Hope and Koestner, 2015), and SDT has also been applied to the COVID-19 pandemic response showing that autonomous versus controlled motivation lead to more policy adherence (Morbee et al., 2021). All this said, by building up the theoretical foundation of where the pandemic concept comes from it will both help readers understand what the concept is, but also increase the theoretical impact of the work by tying it to broader psychological theory/concepts outside the COVID-19 context.

2. Beyond clarifying the conceptualization of pandemic fatigue in the introduction, I think the authors also need to work on setting up a theoretical rationale / context for the other measures they related COVID-19 to. The authors briefly mentioned the HEXACO traits (when explaining the COSMO survey) but they did not really explain why its important to conceptually focus on these traits in terms of their research objectives. And it was also unclear how other concepts like cognitive and affective risk perception, trust, worries, ect., came in to play. I think it is very important for the authors to explain what and why these other concepts are related to the focal research objectives and COVID-19 fatigue, so they don’t come out of the blue when they come up in the empirics. For example, I did not really know how pandemic worries were conceptually different from cognitive/affective risk perception. While I know a lot of these measures were exploratory, I still feel there should be conceptual rationale for the inclusion of each variable analyzed (e.g., to look for convergent, divergent, or criterion validity for the focal fatigue construct) and this needs to be set up in the intro. Without this, its hard to draw meaning from the empirical results.

3. The authors’ rich data allowed them both to examine how pandemic fatigue changed/ developed over the pandemic, and also to examine between and within person effects in

terms of how fatigue related to pandemic response. This is really neat, and the analyses were well done. But I think the impact of these analyses could shine through even more if the authors discussed in the introduction section why it was worth looking at these temporal and within/between person effects in terms of understanding the concept of pandemic fatigue. Based on the authors' conceptualization and theory about the concept what would we have expected in terms of the temporal changes? Would we have expected the within or between effects to be stronger or weaker? It would help for all this to be laid out conceptually before the results, so that way the reader can use the findings to inform their understanding of the concept.

I should note here, that I was a bit surprised to see the between (vs. within) effects of pandemic fatigue be stronger in predicting COVID-19 measure adherence. If fatigue is some subjective state that changes over time as people become burdened with the pandemic (more than some trait-like lack of caring about the pandemic) I would have imagined the within versus between effects to take precedence. However, the stronger between versus within effects made me wonder if the authors are really capturing fatigue versus a more trait like willingness to care (or not) about the pandemic? On the other hand though, it is nice that the within effects are also all significant – so this does support the idea of fatigue somewhat.

4. I would like to see a little more discussion about the development of the fatigue scale since it is so relevant to the paper. For example, I am curious about how the authors conceptualization translated to the items used – and relatedly I think it is important for the measure items to be included in the main manuscript perhaps in a table (since it is so central to the paper). I really liked the items and felt they got at something more than demotivation (which the authors note in the general discussion has been a critique of the concept). So it might be helpful for the authors to flesh out these strengths by showcasing the measure more and their development of it, as this was part of the work that got me most excited. Similarly, a little more could be said in main text about the EFA process as this was very well done and integral to the scale development.

5. As I have noted above – I was a bit thrown by the experimental manipulation in terms of how it differs from both the authors' conceptualization of the construct, and the 6-item scale. Indeed, the one thing that changes from the two focal conditions is the word "motivated" to "demotivated". Nowhere are elements of feeling fatigued, or tired, drained brought up. I think this needs to be addressed by the authors head on both in the theory building section of the intro, the experiment lead in and the GD. If there was a place for more data collection, I could see an experiment in which the manipulation is more parallel to the measure being very helpful. This said, I am not sure if an experiment would be received in the same way now that we are in such a late stage of the pandemic and most measures have been lifted in many countries. I think this package is strong already, so I don't think more data collection in this regard would make or break the paper.

6. I know that the surveys used by the authors were quite large – if the authors had any measure of motivation/demotivation I think it could be quite impactful to show that (1) the Pandemic Fatigue scale factors onto a distinct construct and (2) predicts outcomes above and beyond more general (de)motivation.

7. I liked the general discussion overall – but I found the emphasis was really on the practical implications of this work with respect to the pandemic. While this is vital, I feel the authors could also try to discuss more how studying pandemic fatigue in a context like COVID-19 broadens are more basic understanding about how people self-regulate and mitigate threat during long-term global threats like a pandemic.

8. One last conceptual question that might be general discussion worthy – but I wondered if pandemic fatigue is something that people say/report post-hoc after they stray from the health guidelines as a means of rationalizing/justifying their behavior. This is of course tricky to test, and the experiment helps this a bit (although maybe

writing about fatigue could license future non-compliance).

Other minor comments

- A very minor comment – but on page 5, I found the information about the COSMO survey and present research context a little out of place. I feel this should come after the main theoretical conceptualization of fatigue section that follows.

- P.3 – line 38 – the authors talk about how adherence to the pandemic decreased over time – a little more specificity (e.g., days, months, periods) could help especially since time trends of fatigue are examined later on in the results.

- P4. – when talking about other factors that impact adherence to COVID-19 health protective behaviors it could also help to bring up how many of these measures became tied to political identity (Clinton et al., 2021 ; Gollwitzer et al., 2020)

- I would avoid using the term “highly significant” in reference to a p value <.001 in this work given the very large sample sizes. I think its ok to use that language with respect to an effect size though.

Response Letter

Comments from Reviewer # 1

Comment # 1

Pandemic fatigue has been a much-debated topic, and empirical work has been in many ways held back (or limited in what it dares to claim) by the absence of conceptual agreement. As such, I appreciate the task that this paper has set for itself in trying to move the field on.

Response # 1

Thank you very much for taking the time to review the manuscript and the positive evaluation overall. We also thank you for your thoughtful suggestions and comments which helped us improve the manuscript.

Comment # 2

Introduction:

In general, I think the paper's brief coverage of the different definitions of fatigue is good, and I am convinced that the authors justify their approach to define pandemic fatigue as a gradually emerging subjective state.

I would like to see a bit more probing of the idea baked into the WHO definition that fatigue also shows up as how much people keep themselves informed about the pandemic, in the sense that there will surely be a learning curve to general information at the start (answering questions like "How does one protect oneself?" "Who is most at risk?" etc) that is unlikely to require many updates over time, and so information seeking should reduce in the absence of fatigue. Put differently, it seems likely that an unfatigued person would not uniformly seek the same amount of information about the pandemic over time. This needs addressing theoretically, and picking up later in the manuscript when the metrics regarding information seeking are discussed.

Response # 2

This is an excellent point which we now highlight and discuss in relation to our definition of pandemic fatigue:

"With regard to the information-seeking aspect of PF, it is important to note, however, that people are likely to seek less information over the course of the pandemic irrespective of whether or not they are experiencing PF. People are, for instance, more likely to seek information about a spreading virus, who is at risk, and how to protect themselves in the beginning of a pandemic when the disease is new to them as compared to several months into a pandemic when they have become more familiar with the disease. Thus, the information-seeking aspect of PF specifically refers to a decline in people's tendency to seek information over and above what might be expected naturally." L.78 – L. 85

Comment # 3

More specific notes:

p4, line 58 - I appreciate the use of "arguably" but I also think that this sentence would

benefit from including the point that some of the studies cited at the end of this sentence are valuable precisely because they have sought to control for the “other factors” to which the authors are referring. And so while some of these studies do not claim to find direct evidence of pandemic fatigue, they show that other factors are unlikely explanations for changes in adherence. This part of the manuscript reads as though the authors are selling some of the previous contributions rather short, as though no one has made progress, and thus setting the stage for their own contribution to seem all the larger. P4, line 69-70 - this reads as though the panel surveys were conducted in both countries.

Response # 3

Tackling this issue we have adapted this section of the manuscript and now highlight that other studies provide indirect evidence for the existence of pandemic fatigue by controlling for other potential explanations for the observed decline in public adherence to recommended health protective behaviors:

“Arguably, much of this debate is fueled by the fact that PF so far largely has been derived from behavioral observations that could be explained by other factors^{10-12,19}. Indeed, there are several factors besides PF that could explain the observed decline in public adherence to health-protective behaviors, such as changes in the perceived risk of COVID-19 or plummeting trust in governments’ abilities to handle the pandemic^{14,15}. And while some studies aligning PF with behavioral observations have sought to control for such alternative explanations^{10,11}, they at best only provide indirect evidence for the existence of PF. In studies aiming to assess PF differently than via behavioral observations, by contrast, the construct has so far been defined rather vaguely, conceptualized rather roughly, and measured using single items or non-validated scales^{16,20-23}, making it difficult to draw firm conclusions about the existence and consequences of PF. Tackling these issues, we herein present a theoretically informed conceptualization of PF and provide empirical evidence for its existence, correlates, and consequences..” L.41 – L. 51

Also, thank you very much for making us aware of the misleading sentence on lines 69-70. We adapted it so that it is clear that the panel survey was conducted in Denmark only.

Comment # 4

Results

Overall, I think this section is reasonably strong. It is interesting that the repeated cross-sectional surveys in more than one country showed similar patterns, and that the models with quadratic fit had greatest model fit.

There are a few issues that need addressing, however:

Picking up on the point about unfatigued patterns in information seeking, I wasn’t clear on reading whether initial levels of compliance (to the protective behaviours) and information seeking were taken into account in the analysis. For example, typically women initially adhere more than men to physically distancing, and so there is logically more likelihood of behavioural fatigue. Interpreting the authors’ findings for those with less education (p9, line 161) would also benefit from a clarification here: how did the baseline level of information seeking differ between the highly educated and the less educated?

Response # 4

In our regression analyses we control both for the effect of time and gender (see Figures 2-7). Hence, our estimates regarding the link between pandemic fatigue and compliance do indeed control for the effect of time as well as for potential gender differences. The same goes for the results regarding education. To make this much clearer, we have updated the labelling of the time variable in Figures 2-7 from “Wave” to “Time (survey wave)”.

Comment # 5

Similarly, the authors report a large negative coefficient for “keeping themselves informed about the pandemic” (e.g., p11, line 217). How does the size of this coefficient relate to my earlier point about an unfatigued person probably not uniformly seeking the same amount of information about the pandemic over time?

Response # 5

As mentioned above, we control for the effect of time in our analyses and thus for the fact that people might not uniformly seek the same amount of information over the course of the pandemic. Notably, as suspected, we do indeed find that—independent of their level of pandemic fatigue—people seek less information over time in both the Danish and German repeated cross-sectional surveys (Figure 7) as well as in the Danish panel survey (Figure S7). More importantly, however, across all three surveys, we also observe a negative link between pandemic fatigue and people’s tendency to seek information that is independent of the effect of time (Figures 7 and S7).

Comment # 6

The other key issue for clarification relates to the authors’ treatments of the 2 subfactors - information and behavioural fatigue - throughout the results section. There is the 6-item PFS and the 4-items used, for example, in the experiment. In the experiment, the authors create a composite score of the 4 items (3 of which are protective behaviours, and 1 is information seeking). Are the 4 items given equal weight in this score? Or is information seeking is a quarter of this score, rather than treated as equally as important to behavioural fatigue - as implied by the definition provided in the introduction (and again in the discussion, p15, line 295 - “two distinct factors”). Does the weighting in the composite score between behavioral and information fatigue align with the PFS (as it should)? I found myself looking for this in the Methods section, without success.

Response # 6

The four items used to create the composite score in the experiment were, in line with our pre-registration, given equal weighting (see <https://aspredicted.org/blind.php?x=2cp7k9>). We now describe this more clearly in the manuscript:

“Assessing the impact of our experimental manipulation on people’s intentions to adhere to recommended health-protective behaviors, we relied on an equally weighted composite score of the four outcome items (Cronbach’s $\alpha = .76$).” L.301 – L.303

Whether it would have been more appropriate to use unequal weighting in the composite score from the experiment and put more weight on information seeking is an interesting question. While we do see reasons as to why it might be sensible to put more weight on information seeking, we do not think that it is fully justified to regard information seeking as

more important than other health-protective behaviors, such as physical distancing, mask wearing, and hygienic behavior. For this reason, we decided to stick with our pre-registered analysis plan and use equal weighting for the composite score in the experiment.

That being said, we further analyzed the link between pandemic fatigue and each of the four items used in the composite score from the experiment separately (Figure S22). Similarly, we also analyzed the link between behavioral and information fatigue and each of the four items used in the composite score from the experiment (Figure S23). Notably, in line with our conceptualization of pandemic fatigue, we find the effect of information fatigue to be more pronounced when predicting people's intentions to seek information (Figure S23).

Comment # 7

More specific notes:

Nb p7, line 123 - "hearing about" is different from "seeking information about". Elsewhere in the manuscript the authors discuss the seeking of information.

P8, lines 141, 142 - wording choice: "whereafter" and "wean off" sound odd ("after which" and "wear off"?).

P8, line 145-6 - again the sentence structure suggests there were both German and Danish panel surveys.

P8, line 155 - some of the emotions / perceptions are incompletely described, which makes the results hard to fully understand, and to fully make sense of how they fit into the literature at large. For example: are the authors referring to institutional or generalised trust? (This is important, noting that others have found different protective behaviour adherence patterns over time based on which of these kinds of trust is brought into focus). Similarly: optimism, or optimism bias? P11, line 222-223 - I'd like to see some examples of the controls used. The case or death rate would be a key control, for example.

Response # 7

Thank you very much for this helpful feedback. We adapted the wording of these sentences. Additionally, we now more clearly describe what the different measures of emotions and perceptions refer to in the Results:

"Based on the Danish and German repeated cross-sectional surveys (Figure 2), we further find that people who worried more about potential personal and societal consequences of the pandemic (e.g., losing a loved one or going through an economic recession) experienced higher levels of PF (Cohens $f^2_{\text{predictor} - \text{Denmark}} = .011$; Cohens $f^2_{\text{predictor} - \text{Germany}} = .039$). Conversely, people with heightened cognitive (i.e., the perceived probability and severity of getting infected with COVID-19; Cohens $f^2_{\text{predictor} - \text{Denmark}} = .001$; Cohens $f^2_{\text{predictor} - \text{Germany}} < .001$) and affective risk perception (i.e., the felt closeness, infectiousness, and affective response to the danger of COVID-19; Cohens $f^2_{\text{predictor} - \text{Denmark}} = .018$; Cohens $f^2_{\text{predictor} - \text{Germany}} = .050$), as well as those with higher levels of institutional trust (Cohens $f^2_{\text{predictor} - \text{Denmark}} = .128$; Cohens $f^2_{\text{predictor} - \text{Germany}} = .179$) experienced less PF. Concerning optimism, negative affect, and empathy, which were only assessed in the Danish repeated cross-sectional survey, we find that people who felt more negative emotions (e.g., boredom, loneliness, stress) experienced more PF (Cohens $f^2_{\text{predictor}} = .079$), whereas people who felt optimistic about the

future (Cohens $f^2_{predictor} = .002$) and had a strong sense of empathy towards those most vulnerable to COVID-19 (Cohens $f^2_{predictor} = .015$) experienced it less.” L.189 – L.202

Furthermore, as shown in Figures 2-7, we have added a number of additional control variables from Our World in Data (<https://ourworldindata.org/coronavirus>), including the number of new COVID-19 cases and deaths per million.

Comment # 8

Causal impact study framing:

I find the description of the “low fatigue” treatment completely misleading. If anything it’s the opposite of a fatigue treatment - i.e. it’s a motivating treatment.

Similarly the control does not seem to be true control - a task that takes the same effort in an experiment but that is otherwise neutral with regard to the treatment. This is because asking people to write about what has “affected their everyday behaviour” is likely to have a motivating or (perhaps more likely) demotivating element to it. In any case, it’s not neutral.

The above two points do not imply to me that the study isn’t valid, but that its framing needs a lot of work. Adjusting this framing will also help make sense of findings such as “participants in the control condition reported higher levels of pandemic fatigue than participants in the low pandemic fatigue condition”, which otherwise sound nonsensical.

Response # 8

Thank you for bringing this to our attention. To mitigate the issue highlighted above, we now clearly describe that the low pandemic fatigue condition indeed is akin to a motivating treatment:

“To manipulate participants’ experience of PF, we relied on a brief self-reflection task that specifically targeted the (de)motivational aspect of PF by asking participants in the low/high PF condition to write a few sentences about some of the things that over the last two weeks had motivated/demotivated them to adhere to the four aforementioned health-protective behaviors. In contrast, participants in the control condition were asked to write about some of the ordinary things that had happened and that somehow affected their behavior.” L.279 – L.284

Moreover, we now also discuss the fact that the control condition might not have been completely neutral as well as that our manipulation addresses only the (de)motivational aspect of pandemic fatigue as limitations of our study:

“Finally, while the experiment provides causal evidence for the link between PF and people’s intentions to adhere to various health-protective behaviors, it suffers from at least two limitations. First, the control condition, in which participants wrote about something ordinary, is likely to have elicited unintended feelings of (de)motivation and can therefore not be said to be perfectly neutral in terms of PF. Second, for various reasons (see Methods), the experimental manipulation only targeted the (de)motivational aspect of PF, while not directly addressing the weariness and exhaustion related to it. Overcoming these limitations, future research might develop more comprehensive manipulations of PF and test both their short- and long-term impact in realistic settings.” L.359 – L.366

We decided, however, to keep the same labeling of the experimental conditions (i.e., low pandemic fatigue, control, and high pandemic fatigue) so as to ensure that the framing of the experiment in the manuscript is consistent with the framing of the experiment in our pre-registration (see <https://aspredicted.org/blind.php?x=2cp7k9>). However, we do believe that the additional explanations of these experimental treatments as described above will help the readers to interpret the observed differences.

Comment # 9

Discussion

p15, line 312-314 - I would like to see the finding “other factors, such as people’s affective risk perceptions regarding the pandemic, generally seem to matter as much or even more than pandemic fatigue...” given more room in the Results section. It may be worth a main figure.

Response # 9

We agree and now devote a new paragraph to emphasize that pandemic fatigue appears to be a robust predictor, but not necessarily the most important one. We highlight some of the other predictors that appeared to be important:

“While we do find PF to be consistently related to people’s tendency to adhere to various health-protective behaviors, it is by no means the only nor necessarily the best predictor of this tendency. Among the many other factors considered herein, especially age (Cohens $f^2_{\text{predictor}} = .002$ to $.036$), gender (Cohens $f^2_{\text{predictor}} = .001$ to $.024$), institutional trust (Cohens $f^2_{\text{predictor}} < .001$ to $.042$), worries about potential personal and societal consequences of the pandemic (Cohens $f^2_{\text{predictor}} < .001$ to $.028$), and affective risk perceptions regarding COVID-19 (Cohens $f^2_{\text{predictor}} = .003$ to $.068$) predicted this tendency—in some cases even better than PF—in both the Danish and German repeated cross-sectional surveys. Similarly, for the Danish panel survey, age (marginal Cohens $f^2_{\text{predictor}} < .001$ to $.029$), gender (marginal Cohens $f^2_{\text{predictor}} < .001$ to $.020$), institutional trust (marginal Cohens $f^2_{\text{predictor} - \text{within/between}} < .001/.001$ to $.013/.038$), and affective risk perceptions regarding COVID-19 (marginal Cohens $f^2_{\text{predictor} - \text{within/between}} < .001/.001$ to $.004/.094$) were found to be good predictors together with empathy towards those most vulnerable to COVID-19 (marginal Cohens $f^2_{\text{predictor} - \text{within/between}} < .001/.001$ to $.010/.045$).” L.256 – L.267

Furthermore, we discuss this issue when summarizing our results:

“Overall, our findings suggest that PF is a real phenomenon that needs to be taken seriously. At the same time, our findings also indicate that PF is just one of many factors that may influence people’s tendency to adhere to recommended health-protective behaviors, making it crucial not to exaggerate its impact and lose sight of other and potentially more important factors.” L.321 – L.324

Comment # 10

More specific note:

p16, line 320-327 - I enjoyed this point of speculation, but I think the authors should make it a bit clearer that their findings don’t provide much insight into rates of pandemic fatigue recovery, and that this is mere speculation.

Response # 10

In line with your suggestion, we now highlight the speculative nature of these thoughts and emphasize that more research is needed to adequately answer this question:

“With regard to how to deal with PF, our findings further indicate that interventions aimed at reducing it might benefit from paying particular attention to younger people and people high in extraversion who, on average, reported higher levels of PF. Yet, because PF swiftly began to decrease in both Denmark and Germany as soon as the pandemic slowed down, large-scale interventions might be largely unnecessary as long as each wave of the pandemic is short-lived and people have enough time to psychologically recover between them. The need for interventions remains unclear, however, as one might also speculate that people’s experience of PF builds up from one wave to the other, even when each wave of a pandemic is relatively short. In order to provide clarity to this issue, future research might investigate to what extent PF builds up between waves.” L.325 – L.333

Comment # 11

Materials & Methods

I would also like to know more about the 6-item PFS here, and how it incorporates the two subfactors. This is crucial information to make sense and compare with the 4-item composite scores elsewhere.

Response # 11

Much of the information about the PFS was previously described in the supplementary information. Based on your and the other reviewers’ comments, we have now moved most of this information to the Methods section of the manuscript (see pp. 22-27, Table 1, and Figure 9).

Comment # 12

As with all pandemic surveys, but especially anything that seeks to assess fatigue, I would like to know the rate at which people who were invited to participate (in the cross-sectional surveys) did so, as well as the non-completion rate of the survey, and whether these two details changed over time. Certainly, the attrition of the panel survey (p19, line 407) suggests this information could be important.

Response # 12

In line with your suggestion, we now provide information about the participation and completion rate over time for the Danish repeated cross-sectional survey in Table S2.

For the German cross-sectional survey, we are unfortunately not able to include this information, because this survey was collected by an external panel provider who only provided us with complete responses (i.e., participants who completed the survey in its entirety). In other words, for the German cross-sectional survey we simply do not know how many participants were invited or how many dropped out during the survey each week.

Comment # 13

Figures

Figure 1 - I recommend showing the waves of the pandemic on these charts as a second y axis. (Smoothed death rate and/or case rate)

Response # 13

Thank you very much for this suggestion. Because a dual y-axis can be rather confusing for some readers, we decided not to add a second y-axis. Instead, we have added more plots to Figure 1, so that it now also shows the number of new COVID-19 cases and deaths per million, the COVID-19 reproduction rate, and the policy stringency over time for both Denmark and Germany, based on data from Our World in Data (<https://ourworldindata.org/coronavirus>)

Comments from Reviewer # 2

Comment # 1

This is a very timely paper investigating the topic of pandemic fatigue, seeing how it waxes and wanes over the course of the pandemic, and what psychological factors vary with it in 3 very large datasets. In a final preregistered experiment, the authors find evidence that reminders of being fatigued of the pandemic decreased self-reported intentions to adhere to pandemic restrictions.

This is an excellent paper and I have very little to criticize. I think it is very well conducted, and also important for public policy. While I think this should be published, I have a few smaller concerns.

Response # 1

Thank you very much for reviewing the manuscript and for your positive evaluation overall. We greatly appreciate your insightful comments and suggestions which we found very helpful in revising and improving the manuscript.

Comment # 2

First, where is the pandemic fatigue scale? I can't find it anywhere in the manuscript. Maybe it's buried in the supplement, but I could not find it. The short final version of the scale should be in the manuscript itself so that others can freely use it.

Response # 2

The Pandemic Fatigue scale was indeed presented in the supplementary information. In line with your suggestion, we now present the final version of the scale in the manuscript itself (see Table 1 as well as Figure 9).

Comment # 3

Second, given the large samples, significance is less critical than effect sizes. The authors do not really talk about effect sizes though they occasionally include statements that the effect is large, pandemic fatigue thus seems to have risen quite dramatically" (line 140). Is the rise in pandemic fatigue actually dramatic? The effects seem small to me, albeit important, and the authors should explicitly talk about them.

Response # 3

Thank you for pointing this out. To tackle this issue, we did two things. First, we got rid of wording such as "pandemic fatigue thus seems to have risen quite dramatically". Second, we now also report standardized effect sizes in terms of Cohen's f^2 and critically discuss the magnitude of the observed effects in the Discussion:

"Third, even though we did find a fairly robust link between PF and people's self-reported intention and tendency to adhere to various health-protective behaviors, this link was not always particularly strong, typically yielding small effect sizes. While some might argue that this renders the dawn and rise of PF inconsequential, it should be noted that small effects can be cumulative in nature and can have big consequences in the long run⁵¹. Moreover, because all human behavior is driven by a multitude of factors, it is in most cases not only

unrealistic but also unjustified to expect anyone of these factors to have a big impact by themselves only⁵².” L.352 – L.358

Comment # 4

Third, I found the experiment unconvincing. The authors essentially show that when they remind people of moments when they felt overwhelmed and demotivated by the pandemic, they report feeling more demotivated (i.e., the pandemic fatigue scale measures this) and less likely to adhere to pandemic measures (i.e., which might also be part of the pandemic fatigue scale; though I can't tell because I don't have access to the scale itself). There is a lot of circularity in this study. Basically, the authors manipulate pandemic fatigue and then measure fatigue in various ways. Related, how did the authors measure adherence to pandemic measures in the experiment? This too should be listed in the manuscript so that readers can see the content overlap between the pandemic fatigue scale and the outcome measures. In my eyes, this study is not particularly informative though I would not recommend removing it, but perhaps speaking about what it can and cannot say.

Response # 4

While we do appreciate as well as agree with some of the criticism raised here, we still think that the experiment provides valid evidence regarding the causal impact of pandemic fatigue on people's intention to adhere to various health-protective behaviors for the following reasons:

First, since pandemic fatigue is defined as “a gradually emerging subjective state of weariness and exhaustion from, and general demotivation towards, following recommended health-protective behaviors, including keeping oneself informed about the pandemic” we believe that it is reasonable from a conceptual point of view to manipulate people's motivation in terms of adhering to recommended health-protective behaviors and keeping themselves informed about the pandemic. That being said, we do recognize that our experimental manipulation only manipulates part of what we define as pandemic fatigue, which we now openly discuss as a limitation of the experiment:

“Finally, while the experiment provides causal evidence for the link between PF and people's intentions to adhere to various health-protective behaviors, it suffers from at least two limitations. First, the control condition, in which participants wrote about something ordinary, is likely to have elicited unintended feelings of (de)motivation and can therefore not be said to be perfectly neutral in terms of PF. Second, for various reasons (see Methods), the experimental manipulation only targeted the (de)motivational aspect of PF, while not directly addressing the weariness and exhaustion related to it. Overcoming these limitations, future research might develop more comprehensive manipulations of PF and test both their short- and long-term impact in realistic settings.” L.359 – L.366

Second, although we do acknowledge this as a limitation of the experiment, we specifically decided to only manipulate the (de)motivational part of pandemic fatigue on the basis of both methodological and theoretical considerations—something which we now more clearly describe in the Methods section:

“The decision to focus on the (de)motivational aspect of PF was made on the basis of both methodological and theoretical considerations. First and foremost, we decided to focus on the (de)motivational aspect of PF because it allowed us to straightforwardly manipulate the

experience of PF in opposite directions by simply asking participants to reflect upon what motivated/demotivated them to adhere to recommended health-protective behaviors in the past few weeks. Second, we focused on this aspect because the feeling of weariness and exhaustion that also characterizes PF is more perennial in nature and thus less susceptible to undergo rapid changes in response to time varying situational factors, including that of simple experimental manipulations.” L.589 – L.595

Finally, we believe that the overlap between the manipulation, the pandemic fatigue scale, and the four outcome measures is not too large. To substantiate this, we have, as suggested, added the pandemic fatigue scale in Table 1 and Figure 9 as well as the four outcome measures to the main text:

“Finally, after completing the brief self-reflection task, all participants were asked to complete the PFS and to respond to four items assessing their intention to adhere to recommendations regarding physical distancing (i.e., “Over the next two weeks I will avoid physical contacts and keep a safe distance to people outside my own household”), hygienic practices (“Over the next two weeks I will wash my hands very often and thoroughly and/or use hand disinfectant frequently”), and mask wearing (“Over the next two weeks I will wear a face mask whenever I am inside and cannot keep a safe physical distance to people outside my own household”), as well as to keep themselves informed about the pandemic and current COVID-19 restrictions (“Over the next two weeks I will do everything I can to keep myself updated about the development of the pandemic, and stay informed about the current COVID-19 restrictions”).” L.597 – L.606

We hope that this to some extent alleviates your circularity concerns and makes the added value of the experiment much clearer.

Comment # 5

Fourth, I was surprised that the authors did not measure political orientation, as it seems to be a pretty large predictor of adherence to pandemic measures. Is such a measure present in the dataset? If so, I wonder if the authors would consider analyzing it.

Response # 5

Unfortunately, we did not measure people’s political orientation, and are thus unable to add it to our analyses, which we otherwise would have been happy to do. Please note, however, that previous research has shown that political orientation does not explain public support for governmental responses to COVID-19 in neither Denmark nor Germany¹ (in contrast to, for instance, the US²⁻⁴). Consequently, it might also not matter too much for people’s adherence to governmental recommended health-protective behaviors in these two countries.

Comment # 6

In sum, I really like this paper and hope my comments can be of some help to the authors.

I sign all my reviews,

Michael Inzlicht

Response # 6

Once again thank you very much for reviewing our manuscript. All of your comments were very helpful in revising the manuscript.

Comments from Reviewer # 3

Comment # 1

The Results start with some type of modelling, with no outline of the statistical technique, software used or thresholds for model fit. The Methods must describe the reason for choosing such models, including the relevance of the distribution of the data, including whether the models accounted for the (likely) need to use polychoric correlations. There is no specific reference to the construction of the Pandemic Fatigue Scale (PFS) although other scales with the same name appear in the literature. This needs to be clarified.

Response # 1

Thank you very much for your efforts in reviewing our manuscript, and for all of the insightful comments and suggestions you made. We have sought to address all of them and believe that this has substantially increased the quality of the manuscript.

Due to the word limitation, most information concerning the construction of the Pandemic Fatigue Scale (PFS) was previously presented in the supplementary information. Based on your comment, we have moved most of this information to the Methods section of the main manuscript.

To account for the ordinal structure of the data we did two things. First, we now conduct the exploratory factor analysis on the basis of both Pearson product-moment correlations and polychoric correlations. Importantly, the results from these two analyses yield qualitative similar results:

“In line with previous research suggesting that it is often reasonable to treat ordinal data as continuous^{66,67}, particularly when more than five response categories are used^{68,69}, we treated the data as continuous and conducted the exploratory factor analysis on the basis of Pearson product-moment correlations. For completeness and recognizing that treating ordinal data as continuous may introduce bias⁷⁰⁻⁷², we also report the results of an exploratory factor analysis based on polychoric correlations in the supplementary information. Notably, the exploratory factor analysis based on polychoric correlations yield qualitative similar results to that based on Pearson product-moment correlations.” L.467 – L.473

Second, for all confirmatory factor analyses, we now use robust maximum likelihood estimation with robust standard errors and a Satorra-Bentler scaled test statistic to account for the multivariate nonnormality of the data. Moreover, for completeness and recognizing that treating ordinal data as continuous may introduce bias even when using robust maximum likelihood estimation, we further fitted all models treating the data as ordinal, using robust diagonally weighted least squares estimation. Similar to the exploratory factor analysis, these two estimation methods yield similar results:

“Similar to the exploratory factor analysis, we treated the data as continuous, but estimated all models using robust maximum likelihood estimation with robust standard errors and a Satorra-Bentler scaled test statistic⁸³ to account for the multivariate nonnormality of the data⁸⁴. For completeness and recognizing that treating ordinal data as continuous may introduce bias even when using robust maximum likelihood estimation⁸⁵, we additionally fitted all models treating the data as ordinal, using robust diagonally weighted least squares

estimation⁸⁴ (see supplementary information). Notably, the two estimation methods yielded qualitative similar results.” L.511 – L.517

Following your suggestion, we now clearly state the criteria used to assess the fit of the confirmatory factor models:

“To evaluate the models, we relied on robust versions^{86,87} of the following fit indices and recommended cutoff values⁸⁸: $RMSEA \leq 0.06$, $SRMR \leq 0.08$, $TLI \geq 0.95$, and $CFI \geq 0.95$ ” L.517 – L.519

Moreover, we also describe the statistical software used:

“All analyses presented herein were conducted in R 4.1.3.⁴³” L.136 – L.137

For both the exploratory and confirmatory factor analyses, we now provide more information about our modelling choices and the underlying distribution of the data:

“For all items, no sign of severe univariate nonnormality was observed (i.e., skewness < 2.0 and kurtosis < 7.0)⁵⁹. On the other hand, Mardia’s multivariate tests⁶⁰ indicated that the items were multivariate nonnormal (multivariate skewness = 9.01, $p < .001$; multivariate kurtosis = 153.84, $p < .001$). To explore the factor structure of the initial 10-item PFS, we thus conducted an exploratory factor analysis using an ordinary least squares approach^{61,62} because this approach, in contrast to maximum likelihood estimation, makes no multivariate distributional assumptions about the data⁶³. Considering the fact that most factors are correlated⁶⁴, we opted for an oblique factor rotation, namely oblimin^{63,65}. In line with previous research suggesting that it is often reasonable to treat ordinal data as continuous^{66,67}, particularly when more than five response categories are used^{68,69}, we treated the data as continuous and conducted the exploratory factor analysis on the basis of Pearson product-moment correlations. For completeness and recognizing that treating ordinal data as continuous may introduce bias^{70–72}, we also report the results of an exploratory factor analysis based on polychoric correlations in the supplementary information. Notably, the exploratory factor analysis based on polychoric correlations yield qualitative similar results to that based on Pearson product-moment correlations.

The sampling adequacy of the data was verified using the Kaiser-Meyer-Olkin test⁷³ and found to be acceptable (overall KMO = .92; all KMO values for individual items are > .83). Bartlett’s test of sphericity⁷⁴ further indicated that the item correlations were sufficiently large for conducting an exploratory factor analysis ($X^2(45) = 4267.31$, $p < .001$).” L.460 – L.478

“To validate our findings from the exploratory factor analysis, we conducted a confirmatory factor analysis with data from waves 20 to 43 (16-11-2020–20-09-2021) of the Danish repeated cross-sectional survey ($n = 15,062$), and all 18 waves (27-10-2020–07-09-2021) of the German repeated cross-sectional survey considered herein ($n = 17,946$). The data showed no signs of severe univariate nonnormality (i.e., skewness < 2.0 and kurtosis < 7.0)⁵⁹, but was multivariate nonnormal in both Denmark (multivariate skewness = 3.04, $p < .001$; multivariate kurtosis = 55.46, $p < .001$) and Germany (multivariate skewness = 2.89, $p < .001$; multivariate kurtosis = 57.04, $p < .001$), as indicated by Mardia’s tests⁶⁰. Similar to the exploratory factor analysis, we treated the data as continuous, but estimated all models using robust maximum likelihood estimation with robust standard errors and a Satorra-

Bentler scaled test statistic⁸³ to account for the multivariate nonnormality of the data⁸⁴. For completeness and recognizing that treating ordinal data as continuous may introduce bias even when using robust maximum likelihood estimation⁸⁵, we additionally fitted all models treating the data as ordinal, using robust diagonally weighted least squares estimation⁸⁴ (see supplementary information). Notably, the two estimation methods yielded qualitative similar results.” L.504 – L.517

Although other measures of pandemic fatigue do exist in the literature, they typically consist of only one item or have not been validated more comprehensively. Please also note that we published an initial preprint of this manuscript one and a half years ago (<https://psyarxiv.com/2xvbr>) and that no other measure existed at that time (i.e., 17 Dec. 2020). Moreover, even to this date the PFS is, to the best of our knowledge, the only comprehensively validated measure of pandemic fatigue. This notwithstanding, we made sure to refer to several other measures of pandemic fatigue in the manuscript.

Comment # 2

Before a questionnaire can be used to make comparisons between groups, measurement invariance needs to be established. I don't see evidence that the 2 scales can be added. What is the correlation between these 2 scales, and was a 2-factor Confirmatory Factor Analysis undertaken?

Response # 2

Thank you for pointing this out. In line with your suggestion, we now test for measurement invariance between the two countries (i.e., Denmark and Germany) and find evidence for partial scalar invariance of the PFS:

“To ensure that the PFS measured PF similarly across Denmark and Germany, we tested for measurement invariance by fitting and comparing the fit of several multi-group confirmatory factor analyses with different levels of equality constraints using robust maximum likelihood estimation with robust standard errors and a Satorra-Bentler scaled test statistic⁸³. Importantly, in all cases we relied on the identification strategy proposed by Yoon and Millsap (2007)⁸⁹ and compared the fit of the models using Cheung and Rensvold's (2002)⁹⁰ $\Delta CFI < -.01$ criterion. We used this identification strategy because it circumvents the problem of having to choose an arbitrary reference item which is otherwise required when using the standard marker method for identification⁸⁹. Moreover, we rely on Cheung and Rensvold's (2002) $\Delta CFI < -.01$ criterion rather than the commonly used criterion of significant differences in X^2 , because the significant differences in X^2 criterion is sample size dependent and overly sensitive for large samples⁹⁰. As for the other confirmatory factor analyses conducted herein, we acknowledge that treating ordinal data as continuous may introduce bias⁸⁵ and therefore report results from corresponding analyses in which we treat the data as ordinal using robust diagonally weighted least squares estimation⁸⁴ in the supplementary information. Notably, we find similar levels of measurement invariance irrespective of how we treat the data.” L.537 – L.551

“Testing for configural invariance, we first fitted a multi-group confirmatory factor analysis with no equality constraints across countries. This model fit the data well suggesting that the PFS is configurally invariant across Denmark and Germany (RMSEA = .06 SRMR = .02, TLI = .98, CFI = .99). Next, we tested for metric invariance by constraining the factor loadings across countries to equality and comparing the fit of this constrained model to the fit of the

first model with no equality constraints. Comparing the fit of these two models, we find the PFS to be metrically non-invariant across Denmark and Germany ($\Delta CFI > -.01$). In light of these results, we thus turned to test for partial metric invariance by freeing the factor loadings of the fourth item of the PFS (i.e., “I feel strained from following all of the behavioral regulations and recommendations around COVID-19”). Freeing the factor loadings of the fourth item and comparing the fit of this third partially constrained model to the fit of the first model with no equality constraints, we find support for partial metric invariance of the PFS ($\Delta CFI < -.001$). As a final step, we therefore proceeded to test for partial scalar invariance by additionally constraining the item intercepts across countries to equality—except the intercept of the fourth item—and comparing the fit of this additionally constrained fourth model to the fit of the less constrained third model. Comparing the fit of these two models, we find support for partial scalar invariance of the PFS ($\Delta CFI = -.005$). Taken together, these results indicate that the PFS measures PF in a somewhat similar manner across Denmark and Germany.” L.552 – L.568

As previously described in the supplementary information, the correlation between the information and behavioral fatigue factors was substantial ($r_{Denmark} = .69, p < .001$; $r_{Germany} = .78, p < .001$), which is why we decided to model pandemic fatigue as a second-order latent construct with information and behavioral fatigue as first-order subfactors. Please note that this information now is provided in the Methods section of the main manuscript:

“Results indicated that a two-factor model fit the data well in both Denmark (RMSEA = .06, SRMR = .03, TLI = .97, CFI = .99) and Germany (RMSEA = .07, SRMR = .03, TLI = .98, CFI = .99). The two factors were found to be strongly correlated ($r_{Denmark} = .69, p < .001$; $r_{Germany} = .78, p < .001$), however, pointing to the possibility that a one-factor model would fit the data better. To explore this possibility, we fitted a one-factor model. In both Denmark (RMSEA = .18, SRMR = .08, TLI = .80, CFI = .88) and Germany (RMSEA = .16, SRMR = .06, TLI = .86, CFI = .92) a one-factor model did not fit the data well. Considering the bad fit of the one-factor model and the high factor intercorrelation of the two-factor model, we decided to model PF as a second-order latent construct with information and behavioral fatigue as first-order subfactors. Notably, while the second-order model is statistically equivalent to the two-factor model, and thus fits the data equally well, it has two important benefits: It allows for the combination of the information and behavioral fatigue factors into an overall and parsimonious measure of PF, while at the same time making it possible to explore the correlates and consequences of these two factors separately.” L.520 – L.532

In the supplementary information, we further report several regression analyses in which behavioral and information fatigue are treated as two independent factors, for those who disagree with our modelling approach and think that it is better to model these two factors separately (Figures S9 – S20).

Both a one-factor and a two-factor confirmatory factor analysis was, as described in the supplementary information, indeed undertaken. We now also report this in the main manuscript (see above L.520 – L.532).

Comment # 3

What is the point of modelling the panel data over time with a quadratic term? What of different or concurrent secular trends in different/each country? Different sampling (demographics) at each panel? I am not sure how meaningful these findings are without a strong analysis of secular trends.

Response # 3

The point of modelling the development of pandemic fatigue over time with a quadratic term is to account for the fact that pandemic fatigue, as shown in Figure 1, appears to have followed a concave developmental pattern in both Denmark and Germany. Importantly, we also provide clear evidence that a model with a quadratic term fits the data better than a model without a quadratic term:

“Indeed, including a quadratic term significantly improved the fit of the ordinary least square regression models for the Danish ($F(1, 15982) = 229.33, p < .001$; Cohen’s $f^2_{model} = .015$) and German repeated cross-sectional data ($F(1, 17943) = 66.55, p < .001$; Cohen’s $f^2_{model} = .029$), and of the mixed-model regression for the Danish panel data ($X^2(4) = 248.56, p < .001$; marginal Cohen’s $f^2_{model} = .024$, conditional Cohen’s $f^2_{model} = 4.285$).”
L.157 – L.161

Tackling the plausible issue of potential secular trends, we now also report models that control for the influence of the number of new COVID-19 cases and deaths per million, the COVID-19 reproduction rate, and the overall policy stringency (Figure S1) based on data from Our World in Data (<https://ourworldindata.org/coronavirus>):

“Controlling for time-dependent contextual factors in terms of new COVID-19 cases and deaths per million, the COVID-19 reproduction rate, and policy stringency (Figure 1B), we obtain a similar pattern of results with one exception: Only the quadratic term for time remained significant in the mixed-model regression for the Danish panel data (Figure S1).”
L.161 – L.164

Finally, we are now more cautious in comparing the development of pandemic fatigue in Denmark and Germany throughout the manuscript.

Comment # 4

L140 – on what basis is the term ‘dramatically’ used? Figure 1 is not informative as nowhere is the scale described (response options, range of scores on the 2 scales), direction, and no sample sizes. The authors assume fatigue ‘develops’ – but the figures are not really showing ‘development’ to me – the title doesn’t seem to represent the data.

Response # 4

We agree that the use of the term “dramatically” was not well-founded, and we have therefore removed it. The scale—including response options and range scores—was described in the supplementary information. This information has now been moved to the main manuscript (Table 1 and Figure 9). Moreover, we have adapted Figure 1 so that it now also includes sample sizes per measurement occasion.

Given that Figure 1 shows the mean level of pandemic fatigue over time in two repeated cross-sectional studies, as well as one longitudinal panel study, we believe that this figure illustrates the (non-linear) development of pandemic fatigue.

Comment # 5

L145 – Again, speculation that data from 2 countries being somewhat similar over time without careful examination of secular events (prevalence of infections, deaths, health policy, and the multitude of government announcements) is not convincing.

Response # 5

Thank you very much for this pointer. As described in Response # 3, we now also report models that control for the influence of the number of new COVID-19 cases and deaths per million, the COVID-19 reproduction rate, and the overall policy stringency based on data from Our World in Data (<https://ourworldindata.org/coronavirus>) to tackle the potential issue of secular trends.

Comment # 6

L159 – As noted above, before countries can be compared, configural invariance for factor structures, metric invariance for factor loadings, and scalar invariance for item intercepts need to be established for a scale.

Response # 6

As described in Response #2, we now test for measurement invariance between the two countries (i.e., Denmark and Germany) and find evidence for partial scalar invariance of the PFS.

Adding to this, we are now more cautious in comparing the results from the two countries (see also Response #3).

Comment # 7

L163 – The cross-sectional surveys cannot be used to draw causal conclusions, i.e., a high or low score on the scale is only correlated with a score on another scale. The language throughout the paper needs to reflect the direction of causality is not known.

Response # 7

We agree and in no way intended to suggest that any causal conclusions should be drawn on the basis of the repeated cross-sectional surveys, nor on the panel survey. To resolve this issue, we have carefully re-checked (and, if necessary, adapted) our language throughout the manuscript, so as to make it clear that the results from both the cross-sectional and the panel surveys are only correlational in nature.

Comment # 8

Fig 2. With very large sample sizes it is relatively easy to find lots of statistically significant associations – but they are meaningless in magnitude from the public health, policy and clinical perspectives. The authors need to make clear what associations are of a magnitude that is important from the public health or clinical perspective.

Response # 8

Thank you for pointing this out. To tackle this issue, we now also report standardized effect sizes in terms of *Cohen's f²* and critically discuss the magnitude of the observed effects in the Discussion:

*“Third, even though we did find a fairly robust link between PF and people’s self-reported intention and tendency to adhere to various health-protective behaviors, this link was not always particularly strong, typically yielding small effect sizes. While some might argue that this renders the dawn and rise of PF inconsequential, it should be noted that small effects can be cumulative in nature and can have big consequences in the long run⁵¹. Moreover, because all human behavior is driven by a multitude of factors, it is in most cases not only unrealistic but also unjustified to expect anyone of these factors to have a big impact by themselves only⁵².”*L.352 – L.358

Comment # 9

The US study of 1854 which attempts to manipulate respondents through a writing experience seems incoherent with the rest of the manuscript. Seems odd that this US study was approved by a Danish ethics study – this needs to be clarified. The sampling and representativeness of the participants are unclear. Seems to be convenience sampling.

Response # 9

While we do agree that the experiment most certainly has its limitations, we decided to keep it in the manuscript, in line with the other reviewers’ comments—while pointing out its limitations more clearly in the Discussion:

“Finally, while the experiment provides causal evidence for the link between PF and people’s intentions to adhere to various health-protective behaviors, it suffers from at least two limitations. First, the control condition, in which participants wrote about something ordinary, is likely to have elicited unintended feelings of (de)motivation and can therefore not be said to be perfectly neutral in terms of PF. Second, for various reasons (see Methods), the experimental manipulation only targeted the (de)motivational aspect of PF, while not directly addressing the weariness and exhaustion related to it. Overcoming these limitations, future research might develop more comprehensive manipulations of PF and test both their short- and long-term impact in realistic settings.” L.359 – L.366

The sample used in the experiment was indeed a convenience sample which we now clearly describe in the main manuscript:

“In particular, we recruited a convenience sample of 1,854 U.S. participants via Prolific⁴⁷ and randomized them into three conditions: control, low, and high PF.” L.277 – L.279

The experiment was conducted online by researchers employed at a Danish University. In line with common practices (see <https://www.apa.org/research/responsible/irbs-psych-science>), the experiment was thus approved by the internal review board of the respective (Danish) university.

Comments from Reviewer # 4

Comment # 1

The manuscript “Pandemic Fatigue: Measurement, Correlates, and Consequences” aims to conceptualize, operationalize, and understand the antecedents/consequences of the construct “pandemic fatigue” with respect to the COVID-19 global pandemic.

I thought the manuscript had many strengths. The authors do a nice job driving home why practically the above listed research objectives are important, given the common use of the term “pandemic fatigue” in the press and political discourse about the pandemic, as well the underdeveloped conceptualization/operationalization of the construct in the empirical literature thus far. I was convinced that empirically sound research speaking to the conceptualization, operationalization and antecedents/consequences of ‘pandemic fatigue’ is worthwhile, and will be of broad interest both to researchers studying pandemic response, as well as those in applied roles responding to the pandemic.

I also appreciated the manuscript in terms of its strong empirical/methodological strengths. The researchers embraced open science techniques (i.e., pre-registration, open methods, and data), used very transparent figures to show their data distributions and effect sizes, and used sound analytic strategies in their EFA, CFA and multilevel analyses. I also liked the general tone the authors used to describe and interpret their research findings – I did not feel like they had a “horse in the race” to make the concept of pandemic fatigue the most important construct to consider in the context of the pandemic, but rather tried to provide a sound measure to study the construct, and well-powered longitudinal studies to understand what it can (can’t predict) relative to other important psychological constructs in the context of the pandemic.

At the same time, I do have some comments, questions, and suggestions for the authors that I feel could further improve this work if considered. While I will dig into the specifics more below, a lot of my comments have to do with further expanding the conceptual section of the manuscript in terms of discussing what exactly “pandemic fatigue” is, and being more explicit and detailed in the main parts of the manuscript about how the authors’ conceptualization of the construct was translated to their operationalization of it, both in terms of their six item measure and their experimental manipulation. As I will describe, I really liked the 6-item measure and I feel it taps into a unique construct beyond just demotivation or amotivation to follow pandemic related measures – but I was less certain that the empirical manipulation taps into the same construct as the scale (and something beyond just demotivation). Relatedly, a lot of new ideas seemed to sprout up in the empiric sections without much theoretical setup in the lead in (e.g., the two distinct factors of pandemic fatigue and distinctions between seemingly similar constructs like affective/cognitive risk perception vs. pandemic worries were never really mentioned prior to their appearance in the methods/results). I think these concepts and their relevance to the focal research objectives need to be set up early on in the intro section for readers to understand these distinct yet similar concepts later, and how they apply to the authors’ overarching theoretical framework.

All this said, I also know that general science outlets like Nature Communications have less room for these types of details/nuance and I emphasize with the authors in the challenge of getting this nuance into this format. Hopefully, a happy medium can be found, as I think this focus on conceptualization to operationalization is especially important for measurement

focused work. I also think that more conceptualization/positioning into the theoretical lit both in the intro and general discussion could help the paper make broader theoretical impacts beyond its clear applied impact.

I wish the authors all of the best with this important line of work.

Response # 1

Thank you very much for taking the time to review the manuscript and the positive evaluation overall. We found your thoughtful suggestions and comments very helpful in revising the manuscript. Below, we outline how we have addressed each of your comments.

Comment # 2

Detailed Comments:

1. I think it will be important to further develop the “Conceptualizing and Defining Pandemic Fatigue” section of the introduction. I enjoyed how the authors began with past work on the concept of fatigue, and then also made important distinctions between physical depletion /objective lack of ability to continue to do something and subjective feelings of exhaustion (intertwined with demotivation). Where I would like to see more expansion is as follows:

A. The eventual measure of pandemic fatigue had two components – information fatigue and behavioral fatigue. Both of these components are inherent to the authors definition: “A gradually emerging subjective state of weariness and exhaustion from, and general demotivation towards, following recommended health-protective behaviors, including keeping oneself informed about the pandemic.” But I think the authors should make it more clear that they actually conceptualize the information seeking vs. behavioral adherence components as distinct aspects, and root this both in terms of qualitative descriptions of what people did during the pandemic as well as literature that might make distinctions between information seeking / vs. protective behavior in response to major threats like a pandemic.

Response # 2

As suggested, we extended the conceptualization of pandemic fatigue section and made it clearer that we conceptualize information and behavioral fatigue as two distinct factors that together capture the essence of pandemic fatigue. Following your suggestion, we root this distinction in qualitative descriptions of what was required of people during the pandemic:

“Correspondingly, PF is different from general fatigue, which may arise for various reasons and may affect people’s engagement in many different activities. Notably, the introduced definition of PF highlights information seeking as a health-protective behavior. This is crucial as it (i) acknowledges that for people to know about, and thus, to adhere to recommended health-protective behaviors, they need to keep themselves informed about the current situation and guidelines; and (ii) recognizes that feeling exhausted from and demotivated towards keeping oneself informed is as integral to the experience of PF as feeling exhausted from and demotivated towards adhering to other health-protective behaviors (e.g., physical distancing).” L.71 – L.78

After careful deliberation, we decided, due to space limitations, not to draw on literature that makes a distinction between information seeking and protective behavior in response to other major threats, although we think this would have been interesting.

Comment # 3

B. In reading the scale items of the construct I found it interesting how elements of exhaustion / fatigue were intertwined with the protective behaviors / information seeking people might do in response to the pandemic. This in a way seemed something distinct from just “general demotivation” that the authors describe in their definition beyond the fatigue elements. But on the other hand, the empirical manipulation the authors used really left the fatigue elements behind and centered on the motivated vs. demotivated distinction. So this made me still wonder about what exactly pandemic fatigue is – is it demotivation or feelings of exhaustion, or both? The authors hinted at this tension a bit in the GD when they wrote : “critics of pandemic fatigue might argue that pandemic fatigue, both from a theoretical and practical point of view, is a superfluous construct that represents nothing more than people’s motivation to adhere to different health-protective behaviors” (p17). I would say this critique is much more true of the experiment than the psychological measure presented in the work. All this said, I would like to see more explicit discussion and positioning of how pandemic fatigue is similar versus different from (de)motivation in the conceptual section of the intro to then set up both the measure and the experiment. It might be tough though – because both parts of the paper seem to take a different operational approach with the scale focusing on fatigue and the manipulation focusing on (de)motivation.

Response # 3

We agree that this is an important issue and now more explicitly discuss how pandemic fatigue is similar to and different from amotivation (or demotivation) in the conceptual section of the introduction:

“With regard to related constructs, it is important to dissociate PF from both amotivation and burnout. Broadly speaking, amotivation (or demotivation) may be defined as “a state in which one either is not motivated to behave, or one behaves in a way that is not mediated by intentionality” (p. 190)³⁴. According to Self-Determination Theory, amotivation can take two distinct forms³⁴. First, people may feel amotivated if they believe that their actions will not yield a desired outcome or if they perceive themselves as incapable of attaining a desired outcome³⁴. As an example, people’s motivation for adhering to physical distancing measures may be thwarted if they believe that physical distancing will not help contain the spread of a virus, or if they find it virtually impossible to keep a safe distance to others. Second, people may feel amotivated when the behavior in question has no meaning or overall value for them³⁴. That is, people may feel amotivated when the perceived intrinsic and/or extrinsic utility of engaging in a specific behavior is low. As an example, people’s motivation for wearing a mask may be undermined if the perceived cost of wearing a mask is so high that it outweighs its perceived intrinsic and/or extrinsic benefits. While these two forms of amotivation are likely to play a role in shaping people’s experience of PF, and thus, their sustained effort to adhere to recommended health-protective behaviors, there is more to PF than simply feeling amotivated. In particular, for people to experience PF they should not only feel amotivated, but also worn out and exhausted from having adhered to various health-protective behaviors for a prolonged period of time. It is thus possible to clearly differentiate between PF and simply being amotivated. Someone who doubts the effectiveness of physical distancing measures and who for this reason does not adhere to them is not experiencing PF, for instance, but rather feeling amotivated. In contrast, someone who, after several weeks of adhering to physical distancing measures, is feeling exhausted and due to a

lack of motivation no longer adheres to the measures is not just amotivated, but rather experiencing PF.” L.86 – L.107

We fully agree that the narrow focus on (de)motivation in the experiment is a limitation. However, as we now describe in the Method section, we specifically decided to only manipulate the (de)motivational aspect of pandemic fatigue on the basis of both methodological and theoretical considerations:

“The decision to focus on the (de)motivational aspect of PF was made on the basis of both methodological and theoretical considerations. First and foremost, we decided to focus on the (de)motivational aspect of PF because it allowed us to straightforwardly manipulate the experience of PF in opposite directions by simply asking participants to reflect upon what motivated/demotivated them to adhere to recommended health-protective behaviors in the past few weeks. Second, we focused on this aspect because the feeling of weariness and exhaustion that also characterizes PF is more perennial in nature and thus less susceptible to undergo rapid changes in response to time varying situational factors, including that of simple experimental manipulations.” L.590 – L.597

To openly address the limitations of the experiment, we added the following paragraph to the Discussion:

“Finally, while the experiment provides causal evidence for the link between PF and people’s intentions to adhere to various health-protective behaviors, it suffers from at least two limitations. First, the control condition, in which participants wrote about something ordinary, is likely to have elicited unintended feelings of (de)motivation and can therefore not be said to be perfectly neutral in terms of PF. Second, for various reasons (see Methods), the experimental manipulation only targeted the (de)motivational aspect of PF, while not directly addressing the weariness and exhaustion related to it. Overcoming these limitations, future research might develop more comprehensive manipulations of PF and test both their short- and long-term impact in realistic settings.” L.359 – L.366

Comment # 4

C. I wonder if the authors should incorporate other relevant work/constructs into their conceptualization of pandemic fatigue. Although I don’t know a lot about the concept, I think there has been a lot of research on “psychological burnout” within organizational psychology – and this idea of pandemic fatigue made me think of this.

Response # 4

Thank you for pointing us in this direction. As suggested, we now also briefly touch upon the concept of burnout and how it is related to, but distinct from pandemic fatigue:

“Another construct conceptually related to PF is burnout. Broadly defined as a prolonged psychological response to chronic emotional and interpersonal stressors on the job, burnout is characterized by feelings of cynicism, exhaustion, and inefficacy^{35,36}. What differentiates burnout from PF is not only (some of) the symptoms, but also, and perhaps more importantly, the source of the symptoms. Whereas burnout develops as a consequence of a persistent imbalance between one’s job resources and demands and/or diverging personal and organizational values and visions³⁵, PF emerges as a consequence of continuously having to adhere to various health-protective behaviors that restrict one’s personal freedom and that can be at odds with some of one’s basic needs, such as the need for autonomy (i.e., self-

*regulating one's experiences and actions) and relatedness (i.e., feeling socially connected)³⁴.
..” L.108 – L.117*

Comment # 5

Given the intersection of pandemic fatigue with motivation, I also wondered about the applicability/relevance of self-determination theory which talks about different types of motivation people may have (beyond amount). From SDT, motivation can range from amotivation, to controlled and introjected motivation (doing something to avoid guilt, punishment, or to look good), to autonomous forms of motivation where people value and see the importance of doing something (even if its tough/unpleasant). I thought of SDT especially because when people have more autonomous motivation, they tend to feel less exhausted and more vital when self-regulating (Milyavskaya, Inzlicht, Hope and Koestner, 2015), and SDT has also been applied to the COVID-19 pandemic response showing that autonomous versus controlled motivation lead to more policy adherence (Morbee et al., 2021). All this said, by building up the theoretical foundation of where the pandemic concept comes from it will both help readers understand what the concept is, but also increase the theoretical impact of the work by tying it to broader psychological theory/concepts outside the COVID-19 context.

Response # 5

Thank you very much for this suggestion. In conceptualizing pandemic fatigue, we now also draw on self-determination theory and clearly distinguish pandemic fatigue from simply being amotivated (see Response #3 above)

Comment # 6

2. Beyond clarifying the conceptualization of pandemic fatigue in the introduction, I think the authors also need to work on setting up a theoretical rationale / context for the other measures they related COVID-19 to. The authors briefly mentioned the HEXACO traits (when explaining the COSMO survey) but they did not really explain why its important to conceptually focus on these traits in terms of their research objectives. And it was also unclear how other concepts like cognitive and affective risk perception, trust, worries, ect., came in to play. I think it is very important for the authors to explain what and why these other concepts are related to the focal research objectives and COVID-19 fatigue, so they don't come out of the blue when they come up in the empirics. For example, I did not really know how pandemic worries were conceptually different from cognitive/affective risk perception. While I know a lot of these measures were exploratory, I still feel there should be conceptual rationale for the inclusion of each variable analyzed (e.g., to look for convergent, divergent, or criterion validity for the focal fatigue construct) and this needs to be set up in the intro. Without this, its hard to draw meaning from the empirical results.

Response # 6

In line with your suggestion, we now provide a theoretical/methodological rationale for including all the other measures considered:

“In order to test the existence and conceptualization of a new construct it is crucial to have a sound measurement tool,²³ and to subsequently demonstrate that the construct (via the measurement tool) behaves as expected and is meaningfully associated with other existing constructs, including links to relevant criteria³⁶. Given our conceptualization of PF, this entails (i) developing a measure that assesses people's experience of PF, (ii) demonstrating

that PF develops over time both within and between individuals, (iii) showing that it is meaningfully associated with other constructs, and (iv) providing evidence for its connection to people's tendency to adhere to recommended health-protective behaviors." L.119 – L.125

Moreover, in describing our results, we now more clearly describe what each measure specifically refers to:

"Based on the Danish and German repeated cross-sectional surveys (Figure 2), we further find that people who worried more about potential personal and societal consequences of the pandemic (e.g., losing a loved one or going through an economic recession) experienced higher levels of PF (Cohens $f^2_{\text{predictor} - \text{Denmark}} = .011$; Cohens $f^2_{\text{predictor} - \text{Germany}} = .039$). Conversely, people with heightened cognitive (i.e., the perceived probability and severity of getting infected with COVID-19; Cohens $f^2_{\text{predictor} - \text{Denmark}} = .001$; Cohens $f^2_{\text{predictor} - \text{Germany}} < .001$) and affective risk perception (i.e., the felt closeness, infectiousness, and affective response to the danger of COVID-19; Cohens $f^2_{\text{predictor} - \text{Denmark}} = .018$; Cohens $f^2_{\text{predictor} - \text{Germany}} = .050$), as well as those with higher levels of institutional trust (Cohens $f^2_{\text{predictor} - \text{Denmark}} = .128$; Cohens $f^2_{\text{predictor} - \text{Germany}} = .179$) experienced less PF. Concerning optimism, negative affect, and empathy, which were only assessed in the Danish repeated cross-sectional survey, we find that people who felt more negative emotions (e.g., boredom, loneliness, stress) experienced more PF (Cohens $f^2_{\text{predictor}} = .079$), whereas people who felt optimistic about the future (Cohens $f^2_{\text{predictor}} = .002$) and had a strong sense of empathy towards those most vulnerable to COVID-19 (Cohens $f^2_{\text{predictor}} = .015$) experienced it less." L.189 – L.202

Comment # 7

3. The authors' rich data allowed them both to examine how pandemic fatigue changed/developed over the pandemic, and also to examine between and within person effects in terms of how fatigue related to pandemic response. This is really neat, and the analyses were well done. But I think the impact of these analyses could shine through even more if the authors discussed in the introduction section why it was worth looking at these temporal and within/between person effects in terms of understanding the concept of pandemic fatigue. Based on the authors' conceptualization and theory about the concept what would we have expected in terms of the temporal changes? Would we have expected the within or between effects to be stronger or weaker? It would help for all this to be laid out conceptually before the results, so that way the reader can use the findings to inform their understanding of the concept.

I should note here, that I was a bit surprised to see the between (vs. within) effects of pandemic fatigue be stronger in predicting COVID-19 measure adherence. If fatigue is some subjective state that changes over time as people become burdened with the pandemic (more than some trait-like lack of caring about the pandemic) I would have imagined the within versus between effects to take precedence. However, the stronger between versus within effects made me wonder if the authors are really capturing fatigue versus a more trait like willingness to care (or not) about the pandemic? On the other hand though, it is nice that the within effects are also all significant – so this does support the idea of fatigue somewhat.

Response # 7

We agree and now more clearly describe why it is important to study the development of pandemic fatigue (both within- and between-subjects):

“In order to test the existence and conceptualization of a new construct it is crucial to have a sound measurement tool,²³ and to subsequently demonstrate that the construct (via the measurement tool) behaves as expected and is meaningfully associated with other existing constructs, including links to relevant criteria³⁶. Given our conceptualization of PF, this entails (i) developing a measure that assesses people’s experience of PF, (ii) demonstrating that PF develops over time both within and between individuals, (iii) showing that it is meaningfully associated with other constructs, and (iv) providing evidence for its connection to people’s tendency to adhere to recommended health-protective behaviors.” L.119 – L.125

Given that most of our analyses are exploratory in nature, we decided, after careful consideration, not to state any a priori expectations about the size of the within- and between-subjects effects.

When separating within- and between-subjects effects it is in our experience quite common to find stronger between-subjects effects. As a case in point, we also find stronger between-subjects effects for other state-like variables considered in our analysis (e.g., affective risk perceptions and negative affect; Figures S4 – S7). From a theoretical perspective, we suspect that this pattern of results typically emerges because most psychological states are highly dependent on more stable traits. This notwithstanding, we now provide one plausible explanation for the relatively small within-subjects effects in the Discussion:

“Second, because our assessment of PF began several months into the COVID-19 pandemic, it is not only plausible, but rather likely that people’s experience of PF had already markedly increased as compared to their initial baseline at the onset of COVID-19. Notably, this may have limited the additional rise of PF reported herein, resulting in both smaller within- and between-subjects effects than one would otherwise had observed.” L.347 – L.351

Comment # 8

4. I would like to see a little more discussion about the development of the fatigue scale since it is so relevant to the paper. For example, I am curious about how the authors conceptualization translated to the items used – and relatedly I think it is important for the measure items to be included in the main manuscript perhaps in a table (since it is so central to the paper). I really liked the items and felt they got at something more than demotivation (which the authors note in the general discussion has been a critique of the concept). So it might be helpful for the authors to flesh out these strengths by showcasing the measure more and their development of it, as this was part of the work that got me most excited. Similarly, a little more could be said in main text about the EFA process as this was very well done and integral to the scale development.

Response # 8

We agree and now more clearly describe the development of the pandemic fatigue scale in the Method section, as well as include the final items in Table 1 and Figure 9. Moreover, in line with your comment, we also extended and moved the information about the exploratory and confirmatory factor analysis from the supplementary information to the Method section of the main manuscript.

Comment # 9

5. As I have noted above – I was a bit thrown by the experimental manipulation in terms of how it differs from both the authors’ conceptualization of the construct, and the 6-item scale. Indeed, the one thing that changes from the two focal conditions is the word “motivated” to “demotivated”. Nowhere are elements of feeling fatigued, or tired, drained brought up. I think this needs to be addressed by the authors head on both in the theory building section of the intro, the experiment lead in and the GD. If there was a place for more data collection, I could see an experiment in which the manipulation is more parallel to the measure being very helpful. This said, I am not sure if an experiment would be received in the same way now that we are in such a late stage of the pandemic and most measures have been lifted in many countries. I think this package is strong already, so I don’t think more data collection in this regard would make or break the paper.

Response # 9

As noted in our previous Response # 3, we now discuss this as a limitation of our experimental manipulation. We also suggest future research to develop more comprehensive manipulations of pandemic fatigue (given the challenges that this might only be feasible in an earlier stage of a pandemic, which we will hopefully not experience again soon). We hope that this helps the readers to interpret the results of our experiment, including its limitations. Given the argument of the phase of the COVID-19 pandemic, we refrained (as mentioned by you) from conducting a new experiment.

Comment # 10

6. I know that the surveys used by the authors were quite large – if the authors had any measure of motivation/demotivation I think it could be quite impactful to show that (1) the Pandemic Fatigue scale factors onto a distinct construct and (2) predicts outcomes above and beyond more general (de)motivation.

Response # 10

We agree that this would have been very impactful, but unfortunately, we have no measure of general (de)motivation in our surveys and are thus unable to perform the suggested analyses.

Comment # 11

7. I liked the general discussion overall – but I found the emphasis was really on the practical implications of this work with respect to the pandemic. While this is vital, I feel the authors could also try to discuss more how studying pandemic fatigue in a context like COVID-19 broadens are more basic understanding about how people self-regulate and mitigate threat during long-term global threats like a pandemic.

Response # 11

Thank you for this suggestion. We have revised and extended the discussion and now also mention broader directions for future research to improve our understanding of pandemic fatigue as a construct, but also to improve pandemic management.

Comment # 12

8. One last conceptual question that might be general discussion worthy – but I wondered if pandemic fatigue is something that people say/report post-hoc after they stray from the health guidelines as a means of rationalizing/justifying their behavior. This is of course tricky

to test, and the experiment helps this a bit (although maybe writing about fatigue could license future non-compliance).

Response # 12

This is an interesting point. However, in light of the strict word limitation, and the fact that the experiment partially rules out this possibility, we ultimately decided not to include this in the discussion.

Comment # 13

Other minor comments

- A very minor comment – but on page 5, I found the information about the COSMO survey and present research context a little out of place. I feel this should come after the main theoretical conceptualization of fatigue section that follows.

- P.3 – line 38 – the authors talk about how adherence to the pandemic decreased over time – a little more specificity (e.g., days, months, periods) could help especially since time trends of fatigue are examined later on in the results.

- P4. – when talking about other factors that impact adherence to COVID-19 health protective behaviors it could also help to bring up how many of these measures became tied to political identity (Clinton et al., 2021 ; Gollwitzer et al., 2020)

- I would avoid using the term “highly significant” in reference to a p value <.001 in this work given the very large sample sizes. I think its ok to use that language with respect to an effect size though.

Response # 13

As suggested, we have moved the information about the COSMO survey to the Method section. Furthermore, throughout the manuscript and the supplementary information, we are now more specific when talking about changes over time. Finally, we have removed the term “highly significant” throughout the manuscript. We decided not to add a further discussion on the potential role of political identity in relation to pandemic fatigue, particularly as this would be largely speculative and we cannot speak to this with the current data.

References

1. Jørgensen, F., Bor, A., Lindholt, M. F. & Petersen, M. B. Public support for government responses against COVID-19: assessing levels and predictors in eight Western democracies during 2020. *West Eur Polit* **44**, 1129–1158 (2021).
2. Baxter-King, R., Brown, J. R., Enos, R. D., Naeim, A. & Vavreck, L. How local partisan context conditions prosocial behaviors: Mask wearing during COVID-19. *Proceedings of the National Academy of Sciences* **119**, (2022).
3. Clinton, J., Cohen, J., Lapinski, J. & Trussler, M. Partisan pandemic: How partisanship and public health concerns affect individuals' social mobility during COVID-19. *Sci Adv* **7**, (2021).
4. Grossman, G., Kim, S., Rexer, J. M. & Thirumurthy, H. Political partisanship influences behavioral responses to governors' recommendations for COVID-19 prevention in the United States. *Proceedings of the National Academy of Sciences* **117**, 24144–24153 (2020).

Reviewer #1 (Remarks to the Author):

I thank the authors for responding to my comments so thoroughly. Broadly, I am satisfied with the modifications that they have made to the paper. Please note the issue raised in the third bullet point below, however.

Specifically:

- **The response to my Comment 5 (noting the negative link between pandemic fatigue and people's tendency to seek information that is independent of the effect of time) is very helpful.**
- **I appreciate the clarity added with Response 6. I think this is very helpful for the paper.**
- **I think the changes to the wording of the paper explained in Response 8 suffice. These relate to framing of the low fatigue (motivating) and control conditions of the experiment.**
- **Response 10: Generally this is helpful, however, the final sentence confuses me. The idea is that PF builds up over time, but that there may be some recovery between waves. With a big gap between waves, recovery may even be full. Hence, the phrase "future research might investigate to what extent PF builds up between waves" doesn't make sense. Perhaps this is a typo? Surely they mean to say "recovers/abates between"?**

Reviewer #4 (Remarks to the Author):

It was my pleasure to again read and review the manuscript - Pandemic Fatigue: Measurement, Correlates, and Consequences. For transparency, I was Reviewer #4 in the original review.

Like I wrote in my original comments, I feel very positive about the work both in terms of its theoretical/practical contributions, and also, the quality of the work in terms of the data samples and analytic approaches. I was really impressed by the authors' revisions and responses, and I think they have further strengthened this already strong contribution.

I thank the authors for responding thoughtfully to each of my comments in their letter, incorporating some of my suggestions when possible, and also explaining why they chose not to incorporate some of my other suggestions. I think their conceptualization of pandemic fatigue on pages 4-6 is much improved, and does a great job differentiating the construct from other relevant literatures. I really like how the authors now explain how pandemic fatigue is separate from natural declines in people's tendency to seek information about a novel threat as it becomes less novel (p.5) and how they teased PF apart from general amotivation (p.6). I also commend the authors for all the work they added on p. 22-26 of the detailed methods section to speak to the factor structure and invariance of the measure - their justifications for the different choices they made in these analyses were clear and detailed, and something I will refer my students to in the future as examples of factor analysis and invariance analysis. Indeed, this is the type of paper you feel confident in reading throughout because of its methodological/analytical rigor, and one which I think will have utility as a template/guide that readers can refer to when implementing/learning some of the advanced statistical/data visualization techniques employed by the authors.

I was also asked to speak on behalf of whether the revision addressed the comments of Reviewer 2 who could not again review the work, and I do think they do. The authors do a wonderful job now detailing how they developed the PF scale (as I noted above) and list all the items in a table and also in their factor structure figure. The authors also now show effect sizes throughout and are open about the fact that some of them are small. Lastly, I like how the authors are open about some of the limitations of their

experimental study (both addressing my own and Reviewer 2's concerns about it) and also justify their experimental decisions in the detailed methods section.

Based on all this, I think this work meets the high bar set by Nature Communications, and I would be excited to see it published. Below I list just a last few comments/suggestions that came to mind when reading the work again. But I leave these to the authors' and editors' discretion.

I wish the authors all the best!

Last small comments:

1. One thing I still wondered a bit about at the end of the intro and then when reading the "Correlates of Pandemic Fatigue" section - is that in the intro the authors quickly write how they "investigate its [PANDEMIC FATIGUE] relation to other relevant constructs for people's adherence to various health-protective behaviors (namely, sociodemographics, personality, cognitive and affective risk perceptions regarding COVID-19, and institutional trust)".

But they don't really make it clear conceptually why they wanted to explore PF in relation to all these factors, or give some of their potential expectations they might have had for how it would relate to them. I understand that a lot of these potential correlations might be exploratory, but I think it's ok to say that and still explain a potential pattern we might expect conceptually, and/or say what looking at these relations gives us (e.g., convergent validity if we see PF relate to things we would expect it to). I found the relations the authors found between PF and variables like age, extraversion, emotionality, and agreeableness to correspond with my gut expectations, and so I wonder if the authors can say a little bit more about how these patterns make sense for how they conceptualize PF. In terms of PFs relations to variables like risk perception and institutional trust, these patterns also seemed to make sense, and I think there might even be a literature to support these associations that the authors could bring in more. They do this a little bit at the close of this section when they write: "Taken together, these results show that PF as measured by the PFS indeed is meaningfully associated with other relevant constructs for people's adherence to recommended health-protective behaviors. More precisely, PF was positively associated with constructs that seem to be negatively related to people's tendency to adhere to such measures (e.g., age, cognitive and affective risk perceptions regarding COVID-19, institutional trust), and negatively associated with constructs that seem to be positively related to people's adherence tendency (e.g., negative affect)." But I think all these analyses could be even more powerful, if the authors lead into the analysis with some of this literature and rationale, vs. just close with it.

2. On p. 10, the authors write Cohens $f^2 < .001$ - Is it normal to use this $< .001$ annotation for Cohens f^2 ? I just tend to associate that more with p values so was a little thrown. I was also a bit surprised that a Cohen's f could be so small for a significant effect, but I guess this is because of how large the samples were? This is probably fine, but just wanted to raise it in case.

3. On p. 10 - line 192 - I would write out "heightened cognitive risk perception" in full before the parentheses for clarity.

Reviewer #5 (Remarks to the Author):

I've been invited to comment on the responses of the authors of the paper "Pandemic Fatigue: Measurement, Correlates, and Consequences" to the review report of the original submission provided by Reviewer 3. I've organised my commentary below largely in the sequence of Reviewer 3's comments and revision suggestions. I've also

included a response of my own as a final comment.

Reviewer 3: Comment 1. This comment asked among other things for clarification of the authors' reasons for choosing the statistical techniques, software used and thresholds for model fit and also for information on the construction of the Pandemic Fatigue scale (PFS), the focus of the paper. Much of this requested detail has now been satisfactorily provided by the authors in the updated Methods section of the revised paper. In particular, the authors now provide the necessary detail on the complementary use of appropriate modelling procedures, based on both Pearson and polychoric correlations, for analysing the 7-point ordinal scaling chosen for the PFS items and the model fit statistics used. However, the revision still lacks the necessary detail about the way the initial PFS items were generated and selected. The authors simply state: "Based on our conceptualization of PF, we generated an item pool of 15 items, which we then adapted and reduced to 10 items." While the authors provide a quite detailed and valuable analysis of their conception of Pandemic Fatigue (PF) there is to my mind insufficient information about the item generation procedure. How many researchers were involved in generating the items? What was the source(s) of the items? Was any type of formalised item-writing process used (e.g. based on a clear statement of item and scale intent)? Were the items given any form of detailed testing with a sample of likely respondents (e.g. one-on-one or group cognitive interviews, 'think-aloud' interviews)?

Reviewer 3: Comments 2 and 6. These two comments requested information on measurement equivalence of the PFS across the Danish and German samples. The authors have now provided a detailed analysis of measurement invariance and provide clear evidence for partial scalar invariance across the two samples, thus enabling across-country comparison of the latent PSF variables.

Reviewer 3: Comment 3. This comment focussed on the authors' use of a quadratic term in their modelling of PFS over time and the possibility of confounding resulting from socio-demographic differences across panel cohorts. I believe these questions have now been adequately addressed in the revised submission.

Reviewer 3: Comment 4. Also appropriately addressed in the revision.

Reviewer 3: Comment 5. Appropriately addressed, see the authors' response to Comment 3.

Reviewer 3: Comment 7. This comment expressed Reviewer 3's concern about the authors' use of causal language, arguing that cross-sectional surveys can only provide information about correlation, not causation. The authors state in their rebuttal that: "... we have carefully re-checked (and, if necessary, adapted) our language throughout the manuscript, so as to make it clear that the results from both the cross-sectional and the panel surveys are only correlational in nature." I do not believe this claimed avoidance of causal language has been satisfactorily achieved. Frequently, for e.g., the authors refer in the revised MS to the 'consequences' of PF (including, notably, in the paper title). To me, this is a statement of causation. For example, on p. 4 they state, in relation to PF: "... we herein ... provide empirical evidence for its existence, correlates, and consequences." Given the concern expressed by Reviewer 3 and others with which I agree (see below) about the relevance of the section of the paper describing the attempted experimental manipulation of PF in relation to people's intentions to adhere to self-protective behaviours, I believe the authors should review their response to this Reviewer comment to make certain that the language used to comment on the results of the cross-sectional surveys does not contain causal language.

Reviewer 3: Comment 8. I strongly agree with the comment by Reviewer 3 on the need to provide estimates of the magnitude of the relationships described in addition to their statistical significance so that their public health relevance can be assessed. The authors have responded to this comment by providing estimates of the appropriate 'effect size' (ES - Cohen's f^2) for all relationships derived from their regression analyses. However, I cannot find any statement of the criteria the authors used to ascribe the descriptors 'marginal', 'small', 'medium', 'large' to these ES and some designations appear questionable. For example, on p. 10 of the revised MS, ES of 0.004, 0.19 and 0.21 are all designated as 'marginal', a criterion that is not typically used when researchers refer to Cohen ES. Using conventional criteria for f^2 an ES of 0.004 would likely be ignored or considered approx. zero whereas 0.19 and 0.21 would be designated 'medium'.

Reviewer 3: Comment 9. I agree with concerns about the relevance of the attempted

experimental manipulation of PF and its coherence with the rest of the study raised by Reviewer 3 and others. The stated aim of the experiment is to provide "... evidence for its (i.e. PFs) connection to people's tendency to adhere to recommended health-protective behaviors." But beyond offering evidence of a 'connection' between PF and health-protective behaviors, the authors see that connection as causal: "Finally, while the experiment provides causal evidence for the link between PF and people's intentions to adhere to various health-protective behaviors ...". Thus my reading is that the experiment was designed to manipulate participants' subjective experience of PF by influencing their immediate level of motivation to take self-protective measures (low, high or neutral) using a 'self-reflection' task. In turn, the induced experience of PF was anticipated to directly influence participants' anticipation that they would adhere to four aspects of self-protection from Covid (" ... physical distancing, hygienic practices, mask-wearing, and information seeking ..."), i.e. adhere to recommendations for self-protective behaviour. So, as I understand the design, PF seems to be hypothesised to mediate the anticipated relationship between motivation to undertake self-protective behaviour and the intention to do so in the future. But is this a theoretically justifiable causal model? To me, equally if not more plausible would be a model in which PF is the 'parent' causal factor that subsequently influences (de)motivation and then anticipated behaviours. Additionally, is the experimental manipulation simply determining the participant's expectations about subsequent tasks in the experiment and the frame of reference they should be answered from – simply a kind of 'halo' effect induced by the initial self-reflection task? The causal model of the implied relationships that underpins the experimental design along with an appropriate data analysis approach (e.g. a mediation model) should, I believe, be carefully elucidated, and the possible influence of an expectancy effect as a threat to the validity of the causal inferences drawn from the experiment should be addressed. To my mind, the account of the experiment should not be included in the present paper but reconsidered and re-analysed for a separate paper. An additional comment on sampling for the cross-sectional surveys. As I read the information on sampling, the exploratory factor analysis (EFA), carried out in the Danish sample, was conducted on a single wave of the cross-sectional survey (N = 923) while the confirmatory factor analysis and differential item functioning analyses (CFA, DIF) were conducted on the pooled data from multiple cross-sectional waves in the Danish (N = 15,062) and German (N = 17,946) surveys. While it is typical and desirable to use split samples to develop and validate a measurement model using EFA followed by CFA, also, typically, the split samples are either equal in size or somewhat unbalanced (say, if two samples are used, 40% / 60%). The sampling scheme used in the present study is very much more unbalanced with only a single wave used for the EFA. Additionally, statistical models for both continuous and ordinal data were applied to the same samples. Given the very large amount of data available, a more robust approach would have been to conduct the EFA on data pooled across more than one cross-sectional wave (with possibly a random split for EFA and CFA across all available waves) and also different random sub-samples for the continuous and ordinal methods used. Can the authors provide a supporting rationale for the sampling approach taken or, alternatively, consider re-analysing the cross-sectional data using a more evenly balanced sampling splitting approach?

Response Letter

Comments from Reviewer #1

Comments #1

I thank the authors for responding to my comments so thoroughly. Broadly, I am satisfied with the modifications that they have made to the paper. Please note the issue raised in the third bullet point below, however.

Response #1

Once again, thank you very much for taking the time to review the manuscript and the overall positive evaluation of its revised version.

Comment #2

Specifically:

- The response to my Comment 5 (noting the negative link between pandemic fatigue and people's tendency to seek information that is independent of the effect of time) is very helpful.
- I appreciate the clarity added with Response 6. I think this is very helpful for the paper.
- I think the changes to the wording of the paper explained in Response 8 suffice. These relate to framing of the low fatigue (motivating) and control conditions of the experiment.
- Response 10: Generally, this is helpful, however, the final sentence confuses me. The idea is that PF builds up over time, but that there may be some recovery between waves. With a big gap between waves, recovery may even be full. Hence, the phrase "future research might investigate to what extent PF builds up between waves" doesn't make sense. Perhaps this is a typo? Surely they mean to say "recovers/abates between"?

Response #2

We are glad that you found our responses to most of your previous comments helpful. With regard to our previous Response #10, we agree that our phrasing was imprecise. Accordingly, we have updated the sentences so that it now reads as follows:

"Conversely, because PF swiftly began to decrease in both Denmark and Germany as soon as the pandemic slowed down, such interventions could be largely unnecessary, especially if each wave of the pandemic is short-lived, and people have enough time to psychologically recover between them. The need for interventions remains unclear, however, as one might also speculate that people's experience of PF accumulates from one wave to the other, even when each wave of a pandemic is relatively short. In order to provide clarity to this issue, future research might critically investigate to what extent people recover from PF between waves." L.320 – L.326

Comments from Reviewer #4

Comment #1

It was my pleasure to again read and review the manuscript - Pandemic Fatigue: Measurement, Correlates, and Consequences. For transparency, I was Reviewer #4 in the original review. Like I wrote in my original comments, I feel very positive about the work both in terms of its theoretical/practical contributions, and also, the quality of the work in terms of the data samples and analytic approaches. I was really impressed by the authors' revisions and responses, and I think they have further strengthened this already strong contribution.

Response #1

Again, thank you very much for taking the time to review the manuscript and the overall positive evaluation of its revised version. As in the first round, we appreciate your comments and suggestions on how to further improve the manuscript. Below, we respond to each of your comments/suggestions.

Comment #2

I thank the authors for responding thoughtfully to each of my comments in their letter, incorporating some of my suggestions when possible, and also explaining why they chose not to incorporate some of my other suggestions. I think their conceptualization of pandemic fatigue on pages 4-6 is much improved, and does a great job differentiating the construct from other relevant literatures. I really like how the authors now explain how pandemic fatigue is separate from natural declines in people's tendency to seek information about a novel threat as it becomes less novel (p.5) and how they teased PF apart from general amotivation (p.6). I also commend the authors for all the work they added on p. 22-26 of the detailed methods section to speak to the factor structure and invariance of the measure - their justifications for the different choices they made in these analyses were clear and detailed, and something I will refer my students to in the future as examples of factor analysis and invariance analysis. Indeed, this is the type of paper you feel confident in reading throughout because of its methodological/analytical rigor, and one which I think will have utility as a template/guide that readers can refer to when implementing/learning some of the advanced statistical/data visualization techniques employed by the authors.

Response #2

We are very glad that you found our responses to your previous comments thoughtful, and we sincerely appreciate your assessment of our analytical/methodological approach (and the description/visualization thereof) very much.

Comment #3

I was also asked to speak on behalf of whether the revision addressed the comments of Reviewer 2 who could not again review the work, and I do think they do. The authors do a wonderful job now detailing how they developed the PF scale (as I noted above) and list all the items in a table and also in their factor structure figure. The authors also now show effect sizes throughout and are open about the fact that some of them are small. Lastly, I like how the authors are open about some of the limitations of their experimental study (both addressing my own and Reviewer 2's concerns about it) and also justify their experimental decisions in the detailed methods section.

Response #3

Thank you for taking on the responsibility of assessing our responses to Reviewer #2. We are happy to hear that you found our responses to their concerns positive.

Comment #4

Based on all this, I think this work meets the high bar set by Nature Communications, and I would be excited to see it published. Below I list just a last few comments/ suggestions that came to mind when reading the work again. But I leave these to the authors' and editors' discretion. I wish the authors all the best!

Response #4

Thank you for these additional comments and suggestions, to which we respond below.

Comment #5

Last small comments:

1. One thing I still wondered a bit about at the end of the intro and then when reading the "Correlates of Pandemic Fatigue" section - is that in the intro the authors quickly write how they "investigate its [PANDEMIC FATIGUE] relation to other relevant constructs for people's adherence to various health-protective behaviors (namely, sociodemographics, personality, cognitive and affective risk perceptions regarding COVID-19, and institutional trust)". But they don't really make it clear conceptually why they wanted to explore PF in relation to all these factors, or give some of their potential expectations they might have had for how it would relate to them. I understand that a lot of these potential correlations might be exploratory, but I think its ok to say that and still explain a potential pattern we might expect conceptually, and/or say what looking at these relations gives us (e.g., convergent validity if we see PF relate to things we would expect it to). I found the relations the authors found between PF and variables like age, extraversion, emotionality, and agreeableness to correspond with my gut expectations, and so I wonder if the authors can say a little bit more about how these patterns make sense for how they conceptualize PF. In terms of PFs relations to variables like risk perception and institutional trust, these patterns also seemed to make sense, and I think there might even be a literature to support these associations that the authors could bring in more. They do this a little bit at the close of this section when they write : "Taken together, these results show that PF as measured by the PFS indeed is meaningfully associated with other relevant constructs for people's adherence to recommended health-protective behaviors. More precisely, PF was positively associated with constructs that seem to be negatively related to people's tendency to adhere to such measures (e.g., age, cognitive and affective risk perceptions regarding COVID-19, institutional trust), and negatively associated with constructs that seem to be positively related to people's adherence tendency (e.g., negative affect)." But I think all these analyses could be even more powerful, if the authors lead into the analysis with some of this literature and rationale, vs. just close with it.

Response #5

Thank you for this suggestion which we followed. More specifically, in the introduction we now highlight how investigating the relations between pandemic fatigue and other constructs

that are relevant for people's adherence to various health-protective behaviors is important for the construct validity of the PFS:

"To test the existence and conceptualization of a new construct it is crucial to have a sound measurement tool²³, and to provide evidence for the construct validity of the proposed measurement tool, including its content, convergent, and criterion-oriented validity^{36,37}. Given our conceptualization of PF, this entails (i) developing a measure that assesses all relevant aspects of people's experience of PF (i.e., content validity), (ii) demonstrating that PF develops over time both within and between individuals, (iii) showing that it is meaningfully associated with other constructs (i.e., convergent validity), and (iv) providing evidence for its connection to people's tendency to adhere to recommended HPB (i.e., criterion-oriented validity)." L.104 – L.111

We further added a section explaining the kind of relations we would expect to see (as support for the convergent validity of the PFS) between pandemic fatigue and other constructs that are relevant for people's adherence to various health-protective behaviors in the lead-up to the respective analyses/results sections:

"Next, we investigated the relation between PF and other constructs relevant for people's adherence to recommended HPB. With respect to the convergent validity of the PFS one would expect PF to be negatively associated with factors that have been shown to correlate positively with people's tendency to adhere to recommended HPB (e.g., institutional trust³⁸) as well as positively associated with factors that have been shown to correlate negatively with people's tendency to adhere to recommended HPB (e.g., negative affect⁴³). Given our exploratory approach, we primarily focus on results that are stable across models and countries when presenting and interpreting our findings. Pairwise correlations for all variables considered in the Danish and German repeated cross-sectional surveys are presented in Figures S2–S3." L.160 – L.168.

Finally, we now highlight how the observed findings provide evidence for the convergent validity of the PFS:

"Taken together, PF was negatively associated with constructs that seem to be positively related to people's tendency to adhere to HPB (e.g., age⁴⁶, cognitive and affective risk perceptions regarding COVID-19⁴⁷, institutional trust³⁸), and positively associated with constructs that seem to be negatively related to people's tendency to adhere to HPB (e.g., negative affect⁴³). The PFS thus appears to have high convergent validity." L.206 – L. 210.

Comment #6

2. On p. 10, the authors write Cohens $f^2 < .001$ - Is it normal to use this $< .001$ annotation for Cohens f^2 ? I just tend to associate that more with p values so was a little thrown. I was also a bit surprised that a Cohen's f could be so small for a significant effect, but I guess this is because of how large the samples were? This is probably fine, but just wanted to raise it in case.

Response #6

We agree that it is probably not conventional to use this annotation for Cohens f^2 . We used it because we found it to be more informative than simply writing Cohens $f^2 = .000$. In line with the statistical reporting guidelines for Nature Communications, we have updated the annotation and now only report three-digit rounded values for Cohens f^2 (i.e., Cohens $f^2 < .001$ has been changed to Cohens $f^2 = .000$). Indeed, the large sample sizes allowed us to detect even minuscule associations between variables, which is why Cohens f^2 is so small for some of the significant associations observed. This underlines the importance of reporting effect sizes, particularly in studies with large sample sizes (such as the one reported in our manuscript).

Comment #7

3. On p. 10 - line 192 - I would write out "heightened cognitive risk perception" in full before the parentheses for clarity.

Response #7

We agree and have adapted the sentence accordingly:

“Conversely, people with heightened cognitive risk perception (i.e., the perceived probability and severity of getting infected with COVID-19; Cohens $f^2_{\text{predictor} - \text{Denmark/Germany}} = .001/.000$), heightened affective risk perception (i.e., the felt closeness, infectiousness, and affective response to the danger of COVID-19; Cohens $f^2_{\text{predictor} - \text{Denmark/Germany}} = .018/.050$), as well as those with higher levels of institutional trust (Cohens $f^2_{\text{predictor} - \text{Denmark/Germany}} = .128/.179$) experienced less PF.” L.187 – L.191

Comments from Reviewer #5

Comment #1

I've been invited to comment on the responses of the authors of the paper "Pandemic Fatigue: Measurement, Correlates, and Consequences" to the review report of the original submission provided by Reviewer 3. I've organised my commentary below largely in the sequence of Reviewer 3's comments and revision suggestions. I've also included a response of my own as a final comment.

Response # 1

Thank you very much for your efforts in reviewing our responses to Reviewer #3, and for your additional comments and suggestions. We have sought to address all of them (partly briefly, to adhere to the word limitation) and believe that this has substantially increased the quality of the manuscript.

Comment #2

Reviewer 3: Comment 1. This comment asked among other things for clarification of the authors' reasons for choosing the statistical techniques, software used and thresholds for model fit and also for information on the construction of the Pandemic Fatigue scale (PFS), the focus of the paper. Much of this requested detail has now been satisfactorily provided by the authors in the updated Methods section of the revised paper. In particular, the authors now provide the necessary detail on the complementary use of appropriate modelling procedures, based on both Pearson and polychoric correlations, for analysing the 7-point ordinal scaling chosen for the PFS items and the model fit statistics used. However, the revision still lacks the necessary detail about the way the initial PFS items were generated and selected. The authors simply state: "Based on our conceptualization of PF, we generated an item pool of 15 items, which we then adapted and reduced to 10 items." While the authors provide a quite detailed and valuable analysis of their conception of Pandemic Fatigue (PF) there is to my mind insufficient information about the item generation procedure. How many researchers were involved in generating the items? What was the source(s) of the items? Was any type of formalised item-writing process used (e.g. based on a clear statement of item and scale intent)? Were the items given any form of detailed testing with a sample of likely respondents (e.g. one-on-one or group cognitive interviews, 'think-aloud' interviews)?

Response #2

Thank you for pointing this out. We agree and have added additional information about the item generation and selection process:

"The item generation process consisted of five phases. At first, the first and last author each wrote seven or eight English items (15 items in total) that, in line with our conceptualization of PF, sought to capture a state of weariness and exhaustion from as well as a general demotivation towards following recommended HPB, including keeping oneself informed about the pandemic (Phase 1). Next, the second and third author commented on the items and made suggestions on how to maximize their content validity (Phase 2). The first and last author then subsequently adapted the items in accordance with the comments and suggestions made by the second and third author (Phase 3). In accordance with the recommendations put forward by DeVellis²³, we removed any item that we (i.e., all four authors) perceived as overly redundant, lengthy, and/or difficult to read, leaving us with a

final item pool of 10 items (Phase 4). Finally, the first and second author translated the items into Danish and German, respectively (Phase 5).” L.447 – L.457.

Comment #3

Reviewer 3: Comments 2 and 6. These two comments requested information on measurement equivalence of the PFS across the Danish and German samples. The authors have now provided a detailed analysis of measurement invariance and provide clear evidence for partial scalar invariance across the two samples, thus enabling across-country comparison of the latent PSF variables.

Reviewer 3: Comment 3. This comment focused on the authors’ use of a quadratic term in their modelling of PFS over time and the possibility of confounding resulting from socio-demographic differences across panel cohorts. I believe these questions have now been adequately addressed in the revised submission.

Reviewer 3: Comment 4. Also appropriately addressed in the revision.

Reviewer 3: Comment 5. Appropriately addressed, see the authors’ response to Comment 3.

Response #3

We are glad to hear that you found our revisions to appropriately address Reviewer #3’s comments #2, #3, #4, #5 and #6.

Comment #4

Reviewer 3: Comment 7. This comment expressed Reviewer 3’s concern about the authors’ use of causal language, arguing that cross-sectional surveys can only provide information about correlation, not causation. The authors state in their rebuttal that: “... we have carefully re-checked (and, if necessary, adapted) our language throughout the manuscript, so as to make it clear that the results from both the cross-sectional and the panel surveys are only correlational in nature.” I do not believe this claimed avoidance of causal language has been satisfactorily achieved. Frequently, for e.g., the authors refer in the revised MS to the ‘consequences’ of PF (including, notably, in the paper title). To me, this is a statement of causation. For example, on p. 4 they state, in relation to PF: “... we herein ... provide empirical evidence for its existence, correlates, and consequences.” Given the concern expressed by Reviewer 3 and others with which I agree (see below) about the relevance of the section of the paper describing the attempted experimental manipulation of PF in relation to people’s intentions to adhere to self-protective behaviours, I believe the authors should review their response to this Reviewer comment to make certain that the language used to comment on the results of the cross-sectional surveys does not contain causal language.

Response #4

Thank you very much for pointing this out. To tackle this issue, we changed the title of the manuscript to “Shedding Light on Pandemic Fatigue” and removed any statements about causality (including the ones highlighted above) that do not refer directly to the results of the experiment. In line with the editorial recommendation, we decided not to discard the experiment from the manuscript. We provide an explanation of our reasoning behind this decision in Response #6 below.

Comment #5

Reviewer 3: Comment 8. I strongly agree with the comment by Reviewer 3 on the need to provide estimates of the magnitude of the relationships described in addition to their statistical significance so that their public health relevance can be assessed. The authors have responded to this comment by providing estimates of the appropriate ‘effect size’ (ES - Cohen’s f^2) for all relationships derived from their regression analyses. However, I cannot find any statement of the criteria the authors used to ascribe the descriptors ‘marginal’, ‘small’, ‘medium’, ‘large’ to these ES and some designations appear questionable. For example, on p. 10 of the revised MS, ES of 0.004, 0.19 and 0.21 are all designated as ‘marginal’, a criterion that is not typically used when researchers refer to Cohen ES. Using conventional criteria for f^2 an ES of 0.004 would likely be ignored or considered approx. zero whereas 0.19 and 0.21 would be designated ‘medium’.

Response #5

We apologize for this omission and now clearly state, in line with the guidelines provided by Cohen (1988), that $f^2 \geq .02$, $f^2 \geq .15$, and $f^2 \geq .35$ can be interpreted as a small, medium, and large effect size, respectively. Please note that the “marginal” descriptor does not refer to the size of the effect sizes, but to the fact that f^2 was calculated based on the marginal R^2 (i.e., the proportion of the total variance attributable to the fixed effects portion of the model)—something which we now clearly explain in the manuscript as well:

“To help interpret our findings we report standardized effect sizes. For any comparison of group means we report Cohen’s d , for which values of $\geq .20$, $\geq .50$, and $\geq .80$ can be interpreted as small, medium, and large effect sizes, respectively⁴⁰. For all regression-based analyses we report Cohen’s f^2 , for which values of $\geq .02$, $\geq .15$, and $\geq .35$ can be interpreted as small, medium, and large effect sizes, respectively⁴⁰. For all mixed-model regression analyses we provide an estimate of Cohen’s f^2 based on either the marginal R^2 (i.e., the proportion of the total variance attributable to the fixed effects portion of the model) or the conditional R^2 (i.e., the proportion of the total variance attributable to both the fixed and random effects portion of the model)^{41,42}. For individual fixed effect predictors, we report marginal Cohen’s f^2 based on the marginal R^2 , whereas for full models we report both marginal and conditional Cohen’s f^2 based on the marginal and conditional R^2 , respectively.” L. 121 – L.130.

Comment #6

Reviewer 3: Comment 9. I agree with concerns about the relevance of the attempted experimental manipulation of PF and its coherence with the rest of the study raised by Reviewer 3 and others. The stated aim of the experiment is to provide "... evidence for its (i.e. PFs) connection to people's tendency to adhere to recommended health-protective behaviors." But beyond offering evidence of a ‘connection’ between PF and health-protective behaviors, the authors see that connection as causal: “Finally, while the experiment provides causal evidence for the link between PF and people’s intentions to adhere to various health-protective behaviors ...”. Thus my reading is that the experiment was designed to manipulate participants' subjective experience of PF by influencing their immediate level of motivation to take self-protective measures (low, high or neutral) using a ‘self-reflection’ task. In turn, the induced experience of PF was anticipated to directly influence participants’ anticipation that they would adhere to four aspects of self-protection from Covid (“... physical distancing, hygienic practices, mask-wearing, and information seeking ...”), i.e. adhere to

recommendations for self-protective behaviour. So, as I understand the design, PF seems to be hypothesized to mediate the anticipated relationship between motivation to undertake self-protective behaviour and the intention to do so in the future. But is this a theoretically justifiable causal model? To me, equally if not more plausible would be a model in which PF is the ‘parent’ causal factor that subsequently influences (de)motivation and then anticipated behaviours. Additionally, is the experimental manipulation simply determining the participant’s expectations about subsequent tasks in the experiment and the frame of reference they should be answered from – simply a kind of ‘halo’ effect induced by the initial self-reflection task? The causal model of the implied relationships that underpins the experimental design along with an appropriate data analysis approach (e.g. a mediation model) should, I believe, be carefully elucidated, and the possible influence of an expectancy effect as a threat to the validity of the causal inferences drawn from the experiment should be addressed. To my mind, the account of the experiment should not be included in the present paper but reconsidered and re-analysed for a separate paper.

Response #6

While we agree that the experiment has its limitations, we still believe that it provides causal evidence for the impact of pandemic fatigue on people’s intentions to adhere to various health-protective behaviors for the following reasons. First, as we state in our conceptualization of pandemic fatigue, pandemic fatigue partly consists of a “*general demotivation towards following recommended health-protective behaviors, including keeping oneself informed about the pandemic.*” Targeting the motivational aspect of pandemic fatigues, which our experimental manipulation was specifically designed to do, thus seems like a perfectly valid way to directly manipulate people’s experience of pandemic fatigue. With that said, we do of course recognize that it would have been better if our experimental manipulations also had targeted the other aspect of pandemic fatigue (i.e., “*a gradually emerging subjective state of weariness and exhaustion*”), but, as we describe in the method section, this is difficult to achieve for different theoretical and methodological reasons:

“The decision to focus on the (de)motivational aspect of PF was made on the basis of both methodological and theoretical considerations. First and foremost, we decided to focus on the (de)motivational aspect of PF because it allowed us to straightforwardly manipulate the experience of PF in opposite directions by simply asking participants to reflect upon what motivated/demotivated them to adhere to recommended HPB in the past few weeks. Second, we focused on this aspect because the feeling of weariness and exhaustion that also characterizes PF arguably is more perennial in nature and thus less susceptible to undergo rapid changes in response to time varying situational factors, including that of simple experimental manipulations.” L.602 – L.609.

Second, in support of the internal validity of the experiment, we found clear evidence that our experimental manipulations had a significant impact on the participants’ experience of pandemic fatigue, and further influenced their overall intention to adhere to various health-protective behaviors:

“As shown in Figure 8A, results from an independent samples t-test showed that participants in the low PF condition ($M = 3.08$, $SD = 1.36$) reported lower levels of PF than participants in the high PF condition ($M = 3.55$, $SD = 1.43$; difference = 0.47, $t(1,017.76) = 5.43$, $p_{Bonferroni-adjusted} < .001$, Cohen’s $d = .34$, 95% CI [0.30, 0.64]). Results from two additional independent samples t-tests further revealed that participants in the control condition ($M =$

3.29, $SD = 1.45$) reported higher levels of PF than participants in the low PF condition ($M = 3.08$, $SD = 1.36$; difference = -0.21 , $t(1,079.30) = -2.49$, $p_{\text{Bonferroni-adjusted}} = .039$, Cohen's $d = .15$, 95% CI [$-0.38, -0.05$]) as well as lower levels than participants in the high PF condition ($M = 3.55$, $SD = 1.43$; difference = 0.26 , $t(1,044.55) = 2.92$, $p_{\text{Bonferroni-adjusted}} = .011$, Cohen's $d = .18$, 95% CI [$0.09, 0.43$]). These results suggest that our experimental manipulation of PF was successful." L. 283 – L.293.

"As illustrated in Figure 8B, results from an independent samples t-test revealed that participants in the high PF condition ($M = 5.65$, $SD = 1.18$) expressed weaker intentions to adhere to the four HPB of interest as compared to participants in the low PF condition ($M = 5.94$, $SD = 1.13$; difference = 0.30 , $t(1,019.86) = 4.13$, $p_{\text{Bonferroni-adjusted}} < .001$, Cohen's $d = .26$, 95% CI [$0.16, 0.44$]). In addition, participants in the high PF condition ($M = 5.65$, $SD = 1.18$) expressed weaker adherence intentions than participants in the control condition ($M = 5.86$, $SD = 1.13$; difference = 0.21 , $t(1031.30) = 2.98$, $p_{\text{Bonferroni-adjusted}} = .009$, Cohen's $d = .18$, 95% CI [$0.07, 0.35$]). There was no significant difference between the control condition ($M = 5.86$, $SD = 1.13$) and the low pandemic PF condition ($M = 5.94$, $SD = 1.13$; difference = 0.09 , $t(1,078.10) = 1.24$, $p_{\text{Bonferroni-adjusted}} = .640$, Cohen's $d = .08$, 95% CI [$-0.05, 0.22$])." L. 296 – 305.

In other words, our findings clearly demonstrate that targeting the (de)motivational aspect of pandemic fatigue is a valid way of manipulating people's experience of this phenomenon.

Third, although we cannot rule out the possibility that our experimental manipulation created an experimenter demand (or "halo") effect—something which we now highlight as another potential limitation in the discussion—there is compelling evidence to suggest that experimenter demand effects only pose a very limited threat to the validity of online survey experiments, such as the one we used to test the impact of pandemic fatigue on people's intention to adhere to health-protective behaviors (see de Quidt et al., 2018; Mummolo & Peterson, 2019); a point which we now also highlight in the discussion:

"Second, given the specific nature of the experimental manipulation used, we cannot rule out the possibility that the observed impact of PF on people's intention to adhere to recommended HPB represents nothing more than an experimenter demand effect. Yet, it seems somewhat unlikely that this should be the case, given that experimenter demand effects tend to be fairly modest in size⁵⁴ and mostly non-existent in online survey experiments⁵⁵." L. 352 – L.357.

Although we find the proposed mediation model interesting, we decided not to conduct a corresponding analysis for two reasons. First, as described above, our experimental manipulation was specifically designed to directly manipulate pandemic fatigue by targeting its (de)motivational aspect. Thus, our causal model does not imply that pandemic fatigue mediates the link between (de)motivation and people's intentions to adhere to various health-protective behaviors. In fact, our causal model proposes that pandemic fatigue directly affects people's intentions to adhere to different health-protective behaviors. Second and perhaps even more importantly, our experimental design does not allow us to causally test a mediation model. The reason for this is that to causally test a mediation model both the independent variable and the mediator need to be experimentally manipulated such as in the double randomization design, concurrent double randomization design, or parallel design (see

Pirlott & MacKinnon, 2016). Given that only one of these factors was manipulated in our experimental design, there is no valid way for us to test the proposed mediation model.

Comment #7

An additional comment on sampling for the cross-sectional surveys. As I read the information on sampling, the exploratory factor analysis (EFA), carried out in the Danish sample, was conducted on a single wave of the cross-sectional survey (N = 923) while the confirmatory factor analysis and differential item functioning analyses (CFA, DIF) were conducted on the pooled data from multiple cross-sectional waves in the Danish (N = 15,062) and German (N = 17,946) surveys. While it is typical and desirable to use split samples to develop and validate a measurement model using EFA followed by CFA, also, typically, the split samples are either equal in size or somewhat unbalanced (say, if two samples are used, 40% / 60%). The sampling scheme used in the present study is very much more unbalanced with only a single wave used for the EFA. Additionally, statistical models for both continuous and ordinal data were applied to the same samples. Given the very large amount of data available, a more robust approach would have been to conduct the EFA on data pooled across more than one cross-sectional wave (with possibly a random split for EFA and CFA across all available waves) and also different random sub-samples for the continuous and ordinal methods used. Can the authors provide a supporting rationale for the sampling approach taken or, alternatively, consider re-analysing the cross-sectional data using a more evenly balanced sampling splitting approach?

Response #7

Thank you very much for pointing this out. Out of respect for the participants' time as well as to reduce the risk of low-quality responses and survey length-related dropout, we only included the 10-item version of the pandemic fatigue scale used for the EFA in one wave of the already quite extensive Danish cross-sectional survey. In turn, while we agree that it would have been good to conduct the EFA and the CFA with more balanced samples (as compared to each other), there is unfortunately not much we can do about this now. However, to partially address this issue, we did two things. First, as suggested, we now clearly explicate our rationale for the sampling approach used in method section:

“The final 10 items (Table S10) were administered in the 19th wave of the Danish repeated cross-sectional survey (2020-10-19–2020-10-25) in which 923 respondents participated. Notably, we only included this initial 10-item version of the PFS in one wave of the Danish repeated cross-sectional survey so as keep the length of this already extensive survey to a minimum and in turn reduce the risk of low-quality responses and survey length related dropout in any of the subsequent waves^{61,62}.” L. 458 – L.462.

Second, we now conducted CFAs for each wave of the Danish and German cross-sectional surveys independently, using both robust maximum likelihood estimation and robust diagonally weighted least squares estimation. Robust fit statistics for each of these additional CFAs (84 in total) are presented in Table S12-S15. Importantly, the fit of these CFAs does not qualitatively differ from the fit of the CFAs based on the pooled data:

“Finally, to test the robustness of the second-order model across different waves of the Danish and German repeated cross-sectional surveys, we re-fitted this model for each survey wave—except wave 19th of the Danish repeated cross-sectional which was used for the

exploratory factor analysis—using both robust maximum likelihood estimation and robust diagonally weighted least squares estimation. By and large, the results from this analysis (across 84 models) suggest that the proposed second-order model of PF is robust across waves in both the Danish (RMSEA = .03 to .15, SRMR = .02 to .05, TLI = .93 to 1, CFI = .96 to 1) and German (RMSEA = .03 to .11, SRMR = .02 to .04, TLI = .95 to 1, CFI = .98 to 1) repeated cross-sectional surveys (see Tables S12 – S15).” L.536 – L.544

References

- Cohen, J. (1988). *Statistical power analysis for the behavioral sciences* (Vol. 2). Erlbaum Associates.
- de Quidt, J., Haushofer, J., & Roth, C. (2018). Measuring and Bounding Experimenters Demand. *American Economic Review*, *108*(11), 3266–3302.
<https://doi.org/10.1257/aer.20171330>
- Mummolo, J., & Peterson, E. (2019). Demand Effects in Survey Experiments: An Empirical Assessment. *American Political Science Review*, *113*(2), 517–529.
<https://doi.org/10.1017/S0003055418000837>
- Pirlott, A. G., & MacKinnon, D. P. (2016). Design approaches to experimental mediation. *Journal of Experimental Social Psychology*, *66*, 29–38.
<https://doi.org/10.1016/j.jesp.2015.09.012>

Reviewer #5 (Remarks to the Author):

I've been asked to review the revised version of the paper 'Shedding Light on Pandemic Fatigue' by Dr Lilleholt and colleagues. I thank the authors for their thoughtful responses to my concerns regarding the original revisions they made in response to Reviewer 3's concerns. I was particularly pleased to see that they have now provided more detail on the item generation and selection process used in the development of the pandemic fatigue scale, their clear statement of the guidelines they used in interpreting Cohen's ES, and the revisions made to the rationale for and results of the experimental manipulation of performance fatigue. I am comfortable that the authors have provided satisfactory and complete responses to Reviewer 3's original concerns and I'm happy to recommend the paper for publication. I wish the authors much success with this important publication.

Response Letter

Comments from Reviewer #5

Comments #1

I've been asked to review the revised version of the paper 'Shedding Light on Pandemic Fatigue' by Dr Lilleholt and colleagues. I thank the authors for their thoughtful responses to my concerns regarding the original revisions they made in response to Reviewer 3's concerns. I was particularly pleased to see that they have now provided more detail on the item generation and selection process used in the development of the pandemic fatigue scale, their clear statement of the guidelines they used in interpreting Cohen's ES, and the revisions made to the rationale for and results of the experimental manipulation of performance fatigue. I am comfortable that the authors have provided satisfactory and complete responses to Reviewer 3's original concerns and I'm happy to recommend the paper for publication. I wish the authors much success with this important publication.

Response #1

Once again, thank you very much for taking the time to review the manuscript and the overall positive evaluation of its revised version. We are glad that you found our responses to your concerns thoughtful and satisfactory.